# REHEATFUNQ 2.0.1: A model for regional aggregate heat flow distributions and anomaly quantification

Malte Jörn Ziebarth[1,2] and Sebastian von Specht[3]

[1]GFZ German Research Centre for Geosciences, Telegrafenberg, 14473 Potsdam, Germany
[2]University of Potsdam, Institute of Geosciences, Karl-Liebknecht-Str. 24-25, 14476 Potsdam, Germany
[3]University of Potsdam, Institute of Mathematics, Karl-Liebknecht-Str. 24-25, 14476 Potsdam, Germany

**Correspondence:** Malte J. Ziebarth (ziebarth@gfz-potsdam.de)

**Abstract.** Surface heat flow is a geophysical variable that is affected by a complex combination of various heat generation and transport processes. The processes act on different lengths scales, from tens of meters to hundreds of kilometers. In general, it is not possible to resolve all processes for a lack of data or modeling resources, and hence the heat flow data within a region is subject to residual fluctuations.

We introduce the REgional HEAT-Flow Uncertainty and aNomaly Quantification (REHEATFUNQ) model, version 2.0.1. At its core, REHEATFUNQ uses a stochastic model for heat flow within a region, considering the aggregate heat flow to be generated by a gamma distributed random variable. Based on this assumption, REHEATFUNQ uses Bayesian inference to (i) quantify the regional aggregate heat flow distribution (RAHFD), and (ii) estimate the strength of given heat flow anomaly, for instance as generated by a tectonically active fault. The inference uses a prior conjugate to the gamma distribution for the

RAHFDs, and we compute parameters for a uninformed prior from the global heat flow data base by Lucazeau (2019). Through the Bayesian inference, our model is the first of its kind to consistently account for the variability of regional heat flow in the inference of spatial signals in heat flow data. Interpretation of these spatial signals and in particular their interpretation in terms of fault characteristics (particularly fault strength) is a longstanding debate within the geophysical community.

     We describe the components of REHEATFUNQ and perform a series of goodness-of-fit tests and synthetic resilience anal-

15 yses of the model. While our analysis reveals to some degree a misfit of our idealized empirical model with real-world heat flow, it simultaneously confirms the robustness of REHEATFUNQ to these model simplifications.

     We conclude with an application of REHEATFUNQ to the San Andreas fault in California. Our analysis finds heat flow data in the Mojave section to be sufficient for an analysis, and concludes that stochastic variability can allow for a surprisingly large fault-generated heat flow anomaly to be compatible with the data. This indicates that heat flow alone may not be a suitable

quantity to address fault strength of the San Andreas fault.

## 1 Introduction

Surface heat flow is an important geophysical parameter. It plays an important role in the global energy budget of the solid Earth (Davies and Davies, 2010) and has implications for the exploitability of geothermal energy as a renewable energy source (e.g. Moya et al., 2018). It is also intimately connected to the crustal temperature field which has the potential to control the

crustal elastic properties (Peña et al., 2020) and is hence vital for the understanding of seismic and aseismic crustal deformation. Furthermore, measurements of the surface heat flow haven been indicative of the frictional strength of the San Andreas fault (SAF) by constraining the heat production rate on the fault surface (Brune et al., 1969; Lachenbruch and Sass, 1980).

Global patterns of surface heat flow have been investigated in multiple works (e.g. Pollack et al., 1993; Goutorbe et al., 2011; Lucazeau, 2019). The models therein usually assign an average heat flow to each point of Earth's surface, for instance
by dividing the surface into a grid. We denote this as the "background heat flow" $q_b$ which might follow from the two main sources of crustal heat flow, mantle heat transmission and radiogenic heat generation (Gupta, 2011). As data accumulated, the additional information was used in later works to improve the spatial resolution of $q_b$ models.

Alas, even at the finer resolution of newer works, the models of global heat flow do not perfectly describe the heat flow measurements due to fluctuations. Goutorbe et al. (2011) observe that a residual variation of 10 to 20 mW m$^{-2}$ remains between
heat flow measurements not further than 50 km apart. One potential cause of this variation is the varying concentration of radiogenic elements within the upper crust, which has been observed to change by a factor of five within a couple of tens of meters (Landström et al., 1980; Jaupart and Mareschal, 2005).

Whatever the cause, the magnitude of the variation observed by Goutorbe et al. (2011) and its spatial extent are similar to some anomalous signals generated by processes that one might wish to investigate and distinguish from the background $q_b$.
The fault-generated heat flow anomaly discussed by Lachenbruch and Sass (1980) on the SAF, with peak heat flow less than about 27 mW m$^{-2}$, is an important example. The magnitude similarity between the residual variation and the queried signature implies that it is difficult to establish bounds on the latter.

In this article, we introduce the REHEATFUNQ model (REgional HEAT Flow Uncertainty and aNomaly Quantification) which aims to

1. quantify the variability within regional heat flow measurements, and to

2. identify how strong the surface heat flow signature of a deterministic process, e.g. fault-generated heat flow, can be given a set of heat flow measurements in the study area.

REHEATFUNQ approaches these goals by aggregating heat flow measurements in a region of interest (ROI) into a location-agnostic distribution of heat flow. It considers the heat flow within the region as the result of a stochastic process, and hence the
aggregate distribution as the probability distribution of a random variable. In a Bayesian workflow, this distribution is inferred from the regional heat flow data and from prior information. Processes such as the fault-generated surface heat flow can be quantified by supplying the impact of the process onto each data point and inferring the posterior distribution of a process strength parameter.

The REHEATFUNQ model is an empirical model. In this study, we have performed a number of analyses of the New
Global Heat Flow (NGHF) data base by Lucazeau (2019) to inform the model. Synthetic computer simulations based on the REHEATFUNQ model assumptions have been performed to test the model performance on ideal data. We also perform a resilience analysis based on a number of alternative to the model assumptions which are also compatible to the NGHF data base.

The manuscript starts with a description of the (heat flow) data fundament of the REHEATFUNQ model in section 2. The methodology section 3 continues with a physical motivation for the REHEATFUNQ model before it transitions to a technical description of the model's capabilities. Section 4 bundles statistical analyses of the performance of the REHEATFUNQ model and is rather technical. It starts out by assessing how well the model assumptions are reflected in real-world data, uses stochastic computer simulations to investigate whether known imperfections inhibit REHEATFUNQ's usefulness, and discusses physical limitations of the model. Section 5 then illustrates how to apply the REHEATFUNQ model by means of the San Andreas fault in Southern California before section 6 concludes this work. As a reference for the application of the REHEATFUNQ model, appendix A summarizes all analysis steps mentioned throughout the manuscript in a workflow cheat sheet.

## 2   Heat flow data

This work is fundamentally built on the analysis of surface heat flow measurements, that is, point measurements of the flow of thermal energy from Earth's interior through the outermost layer of the crust into the atmosphere. Heat flow has units of energy divided by time and area, and integrated over an area of Earth's surface, it gives the power at which thermal energy transfers from the inside to the atmosphere.

Heat flow is typically estimated from temperature measurements at varying depths within a borehole. From these measurements, the temperature gradient is estimated which, multiplied with the heat conductivity of the surrounding rock, leads to the heat flow estimate (e.g. Henyey and Wasserburg, 1971). For more details, we refer, for instance, to Henyey and Wasserburg (1971); Fulton et al. (2004); Sass and Beardsmore (2011).

Measuring heat flow is a difficult task. Each measurement requires a borehole and sufficient time to establish temperature equilibrium at the sensors (Henyey and Wasserburg, 1971). Furthermore, the temperature profile close to Earth's surface might not be linear with depth, as would be imposed by a constant heat flow. The causes for these perturbations can include topography, erosion, climate, and water circulation (Lucazeau, 2019), the latter as advection or convection. These perturbations have to be corrected for to estimate the crustal heat flow component of the measured temperature profile. Otherwise crustal heat flow estimates will be biased or uncertain.

Heat flow data is multidimensional, spanning most prominently the heat flow as well as the spatial dimensions. Within the REHEATFUNQ model, the heat flow data is aggregated within the region of interest (ROI), and investigated without regard for the further spatial distribution. We call this data set, reduced to the heat flow dimension only, the *regional aggregate heat flow distribution*. Formally, given a set $\{(\boldsymbol{x}_i, q_i) : i = 1, ..., N\}$ of heat flow measurements within a ROI, one may write the regional aggregate heat flow distribution as result of the mapping

$$\{(\boldsymbol{x}_i, q_i) : i = 1, ..., N\} \rightarrow \{q_i : i = 1, ..., N\}. \tag{1}$$

### 2.1   Data used and general filtering

In this work, we build upon a global data base of heat flow measurements compiled by Lucazeau (2019), the NGHF. This data set, a continuation of the effort of Pollack et al. (1993), is a heterogeneous collection of 69,730 heat flow measurements from

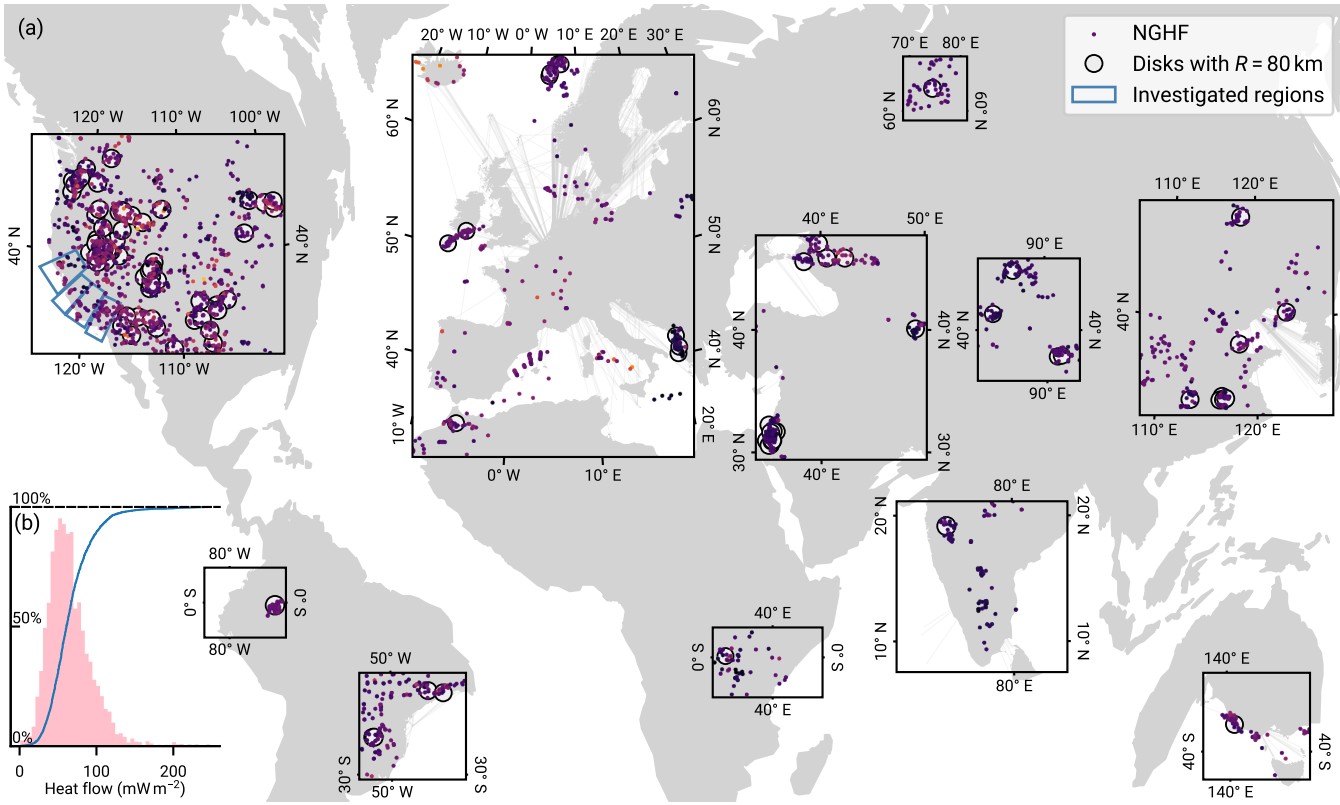

**Figure 1.** New Global Heat Flow (NGHF) database (Lucazeau, 2019) and random regional heat flow samples. The map shows data points from the NGHF used in this study, corresponding to positive continental heat flow values with quality ranking "A" to "B" and not exceeding 250 mW m$^{-2}$. The set of *random global R-disk coverings* (RGRDCs) used in section 3.3.2 to determine the estimates of the prior parameters is illustrated by thinly outlined disks. The algorithm to distribute the disks is described in appendix B. The analysis regions used in section 5 for the SAF are shown in thicker blue outline. Inset (b): empirical distribution function and histogram of the same global heat flow data.

a variety of studies. It covers the time period from 1939 to 2019 and covers the globe on multiple spatial scales from repeat measurements at the same location up to the largest nearest-neighbor distances of ~1200 km. We will use both the global coverage as well as the specific region of Southern California in this work. Due to the heterogeneous spatial coverage and quality of the data, we apply a number of data filters beforehand.

We use only a quality-filtered subset of the NGHF in all our following analyses. Lucazeau (2019) compiled heat flow data from a wide range of sources, spanning decades of technological improvements in instrumentation and combining different efforts of perturbation correction and uncertainty estimation. To obtain a more homogeneous data quality, we follow the quality assessment described by Lucazeau (2019) and discard data points of the lowest quality rankings C to F and those earlier than 1990 (marking an increase in data quality). We furthermore remove negative heat flow values.

Since we will consider a continental scenario, we furthermore remove data points not marked as continental crust (i.e. not key A to H in field "code1"). Finally, we discard data points categorized as possibly geothermal following Lucazeau (2019), that is, those exceeding $250\,\mathrm{mW\,m^{-2}}$. The remaining data set has 5,974 entries. The global aggregate heat flow distribution of this remaining data set is shown in Fig. 1.

The filtering steps described in this section are an example of what is subsumed as step 1 of the workflow listed in Appendix
105 A.

## 3 Methodology: a stochastic model of regional aggregate heat flow

### 3.1 Physical basis

Surface heat flow is the result of heat, generated in the Earth's interior, being transported to the surface by diffusive, advective, and convective processes. The main sources of heat within Earth are the thermal energy from its planetary genesis and the
110 decay of radioactive elements (Christensen, 2011; Mareschal and Jaupart, 2021). Within the crust, heat production due to the friction on faults can be large enough to cause measurable local disturbances of the temperature field (e.g. Kano et al., 2006) and could potentially also lead to significant disturbances in the surface heat flow field if the frictional strength of the fault were large (Brune et al., 1969; Lachenbruch and Sass, 1980).

After generation, three modes of steady transport can be available to bring the heat to the surface. Heat diffusion occurs
throughout the Earth's interior. Advection can occur with the tectonic movement of rock or by means of gravitationally driven pore water movement (Molnar and England, 1990; Fulton et al., 2004). Convective processes range from magma convection in the mantle through crustal pore water convection (Bercovici and Mulyukova, 2021; Hewitt, 2020).

Both generation and transport of heat within Earth are subject to a number of unknowns such as material composition in terms of heat generation and conduction, the geometry of convection cells, and the existence of groundwater flow (e.g. Morgan,
2011). Some of these parameters are difficult to determine, and typically residual fluctuations remain in thermal models even if those models take into account a multitude of available information (e.g. Cacace et al., 2013; Fulton et al., 2004). That is, even though the principles underlying the full surface heat flow field are known, the incomplete knowledge of the specific crustal processes and material properties defining a specific region's surface heat flow, in combination with measurement uncertainties, make it generally impossible to model the exact surface heat flow that is measured.
Our approach is to acknowledge that a model is unlikely to capture the full surface heat flow signal simply because the input data does not capture all relevant features of the subsurface, or because the measurement is uncertain. The concept of REHEATFUNQ is then to abstract these unknowns into a black box stochastic model of surface heat flow within a region. The stochastic model condenses the spatial distribution of heat flow into a single probability distribution of heat flow $q$, $\gamma(q)$, for the whole region, agnostic to where heat flow is queried within. This way, unknowns about the parameters that control surface heat
flow are captured by the amount and variance of the measurement data. If a region is characterized by uniform heat flow—that is, it is independent and identically distributed following a single distribution $\gamma(q)$—and sufficient data has been collected,

a statistical analysis will yield a precise result. Instead, variability in the heat flow controlling parameters reflects in a wider spread of the inferred distribution.

The arguments that motivate the REHEATFUNQ approach are related to the spatial variability of heat flow. Surface heat flow exhibits variability on a large range of scales. Long wavelength contributions follow from the diffusion from deep heat sources. The diffusion up to the crustal surface smoothes lateral pattern of heat flow from these sources with a characteristic length of 100 km (Jaupart and Mareschal, 2005, 2007). If the resulting signal does not vary significantly within the extent of the ROI, we label it the locally uniform background $q_b$. In our later analysis, the extent will be ~160 km but this is not a hard constraint on the region size.

The surface heat flow also contains signals of smaller spatial scale, say 50–100 km and below. We label the surface heat flow that varies spatially within the ROI with $q_s(\boldsymbol{x})$. For instance, the short-wavelength effects can be varying radiogenic heat production from the tens-of-meters to kilometers scale (Jaupart and Mareschal, 2005), or recent tectonic history through movement of heated mass or friction on faults (Morgan, 2011).

One type of short wavelength signal are topographic effects. Since they are more readily corrected for (e.g. Blackwell et al., 1980; Fulton et al., 2004), we list them seperately as $q_t(\boldsymbol{x})$. Topographic effects act on the scale of hundreds of meters to multiple kilometers (see e.g. the extent of the mountains listed by Blackwell et al., 1980) if the boreholes are not sufficiently deep (that is, shallower than 75–300 m depending on temperature gradient and topographic variability; Blackwell et al., 1980).

Finally, the heat flow might also be influenced by random measurement error $q_f$. This includes all kinds of difficulties inherent to the process of drilling, measuring temperature, and evaluating heat flow gradients. These effects are independent of location.

All these unknown contributions to the surface heat flow complicate the inference of a *known* constituent of the heat flow signal from the data. For instance, one might have good knowledge about the location of an underground heat source and its heat transport to the surface, and hence be able to accurately model the spatial surface heat flow signature $q_a(\boldsymbol{x})$ that the heat source generates, but might not know about the heat source's strength and hence the magnitude of the signature. The quantification of fault-generated heat flow anomalies on the San Andreas fault is a paragon of this problem (Brune et al., 1969; Lachenbruch and Sass, 1980; Fulton et al., 2004) and inspires the name $q_a$ ("anomaly"). Because the surface heat flow is influenced by many unknown effects—unknown but evident due to the variability that is not perfectly fit by the model's signature—it is not trivial to infer the magnitude of the model's heat source. This applies particularly if the magnitude of the signature generated by the actual heat source is in the order of or less than the spread due to the unknown constituents.

REHEATFUNQ aims to solve this issue through the stochastic model of the unknown constituents of surface heat flow, and consequently to help researchers calibrate models of specific surface heat flow constituents. The surface heat flow field $q(\boldsymbol{x})$ is separated into the modeled heat flow $q_a(\boldsymbol{x})$ and the unknown contributions $q_b$, $q_s(\boldsymbol{x})$, $q_t(\boldsymbol{x})$, and $q_f$. The magnitude of the modeled heat flow $q_a(\boldsymbol{x})$ is expressed in terms of the power $P_H$ of the heat source (an example is given later in section 3.4.1), and a Bayesian inference of this parameter is performed using heat flow measurements $q_i$, that is, samples of the unknown stochastic contituents transformed to $q(\boldsymbol{x})$ when combined with $q_a(\boldsymbol{x})$. Before the following sections discuss the

stochastic model and the inference of the magnitude of $q_a$, we discuss the separation of $q_a$ from the unknown constituents and, equivalently, how the stochastic model and the modeled heat flow relate to the heat flow measurements.

If heat transport in the crust is linear, which is the case for conduction and advection, the heat flow $q(\boldsymbol{x})$ is a superposition of the five constituents:

$$170 \quad q(\boldsymbol{x}) = \underbrace{q_b + q_s(\boldsymbol{x}) + q_t(\boldsymbol{x}) + q_f}_{q_u(\boldsymbol{x})} + q_a(\boldsymbol{x}). \tag{2}$$

Here, we have collected all these unknowns into $q_u(\boldsymbol{x})$. If heat transport is nonlinear, for instance in case of nonlinear convection, a superposition like this would not be possible. Instead, $q(\boldsymbol{x})$ would be a nonlinear function of the sources of $q_b$, $q_s$, $q_t$, $q_f$, and $q_a$. If the heat source that causes the anomaly $q_a$ is itself a driver of the convection, REHEATFUNQ as developed in this paper cannot be applied (with one technical exception mentioned further down in Appendix G whose applicability is
unclear). This might not be a significant restriction, however: if the heat source that generates the anomaly $q_a$ is strong enough to drive convection on significant length scales (that is, 1–10 km scale), the resulting surface heat flow signature is probably large enough (that is, more than 50–100 mW m$^{-2}$) that the separation from the 'background noise' (the undisturbed heat flow) is less challenging.

However, if the magnitude of $q_a$ is small (that is, less than about 50–100 mW m$^{-2}$), the need for a statistical method, such
as REHEATFUNQ, is essential. In case of a small $q_a$ with crustal heat source, the source will be similarly small and likely not be a driver of convection. Then, if some of the other heat sources $q_b$, $q_s$, $q_t$, and $q_f$ drive nonlinear convective transport, a linearization of the heat transport equation similar to the one performed by Bringedal et al. (2011) can be performed, which would again separate $q_a$ as a linear constituent of $q(\boldsymbol{x})$ from the unknown:

$$q(\boldsymbol{x}) = q_u(\boldsymbol{x}) + q_a(\boldsymbol{x}). \tag{3}$$

Illustratively, the nonlinear convection due to other sources would act as an advective term for the diffusion-advection of the anomalous heat source.

Equation (3) shows the extent of separation that is required for REHEATFUNQ to be applied. It enables the linear separation of the model output from the unknown heat flow which is treated by the stochastic approach. But what motivates the stochastic approach, describing $q_u$ by a probability distribution $\gamma(q_u)$?
For the error term $q_f$, the treatment as a random variable is straightforward. To treat the other terms stochastically is less evident since the surface heat flow field should in principle be deterministic and accessible to precise measurement given enough effort. Here we can consider the random location sampling of a deterministic $q_u$ landscape as a stochastic source of the $q_u$ random variable. Figure 2 illustrates this approach. The surface heat flow field acts like a random variable transform of the spatial random variable to the random variable $q$. The probability density of $q$ derives from the level set of the heat flow field.
The approach illustrated in Fig. 2 highlights why it can be important to prevent spatial clustering within the data. If $q_s(\boldsymbol{x})$ is indeed a significant source of randomness within $q(\boldsymbol{x})$, data independence depends significantly on the independence of sample locations, which is highly questionable if sample clusters heat flow measurements cluster e.g. around a geothermal field. What is more, clustered sampling point sets have high discrepancy so they could additionally lead to a less accurate deterministic

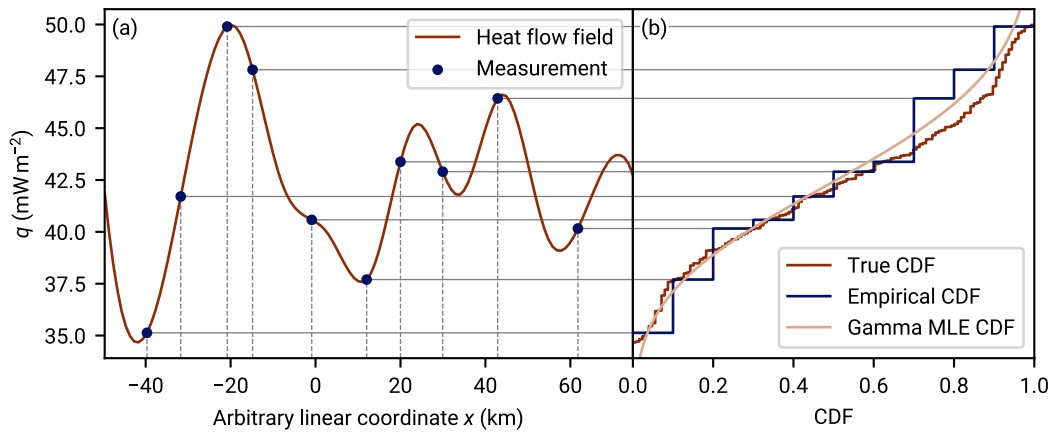

**Figure 2.** The stochastic model of regional aggregate heat flow from a deterministic surface heat flow field: an artificial illustration. Panel (a): an artificial one-dimensional surface heat flow field generated from artificial underground heat sources. The underground heat sources (200 km wide, 80 km deep grid with 201×151 cells, not shown) have been optimized from random initial values such that the surface heat flow they generate fits a *target heat flow distribution* whose aggregate distribution is close to a gamma distribution (details in Appendix F). The blue dots illustrate heat flow measurement $\{q_i\}$ at random locations $x_i$ (dashed gray lines). The set $\{q_i\}$ is a sample of the regional aggregate heat flow distribution (RAHFD), the projection of the measurements to the heat flow dimension $q$ (solid gray lines). Panel (b): this panel shows, in sideways view, the empirical cumulative distribution function (CDF) of the RAHFD. The aggregation process is illustrated by the horizontal gray lines connecting this panel to the (a). Furthermore, the target RAHFD (derived from the continous *target heat flow distribution* of the panel (a)) is shown as well as a maximum likelihood estimate from the sample data. Combined, the two panels show how random spatial sampling of a deterministic heat flow field yields a stochastic RAHFD.

integration properties of the underlying heat flow distribution (Proinov, 1988). The minimum distance criterion, effectively
creating a Poisson disk sampling, can potentially trade discrepancy (Torres et al., 2021) and bias for data set size.

Nevertheless, clusters may also contain variability due to the measurement error term $q_f$. This information would be lost if clusters would be reduced to single points through the minimum distance criterion. REHEATFUNQ mitigates clusters while preventing this data loss by considering data points which exclude each other due to the $d_{\mathrm{min}}$ criterion as alternative representations of the cluster. Each alternative is then considered in the likelihood. The following sections 3.2 and 3.3.1 detail this
process.

## 3.2 Mitigating spatial clustering of heat flow data

To date, the spatial distribution of heat flow data is inhomogeneous. In particular, spatial clusters exists around the points of interest of past or contemporary explorations in which the heat flow data were measured (e.g. The Geysers geothermal field in section 5, Fig. 18). This property can be problematic for our stochastic model of regional aggregate heat flow (section 3.1).
If a significant part of the stochastic nature of regional aggregate heat flow is due to the random sampling of an unknown

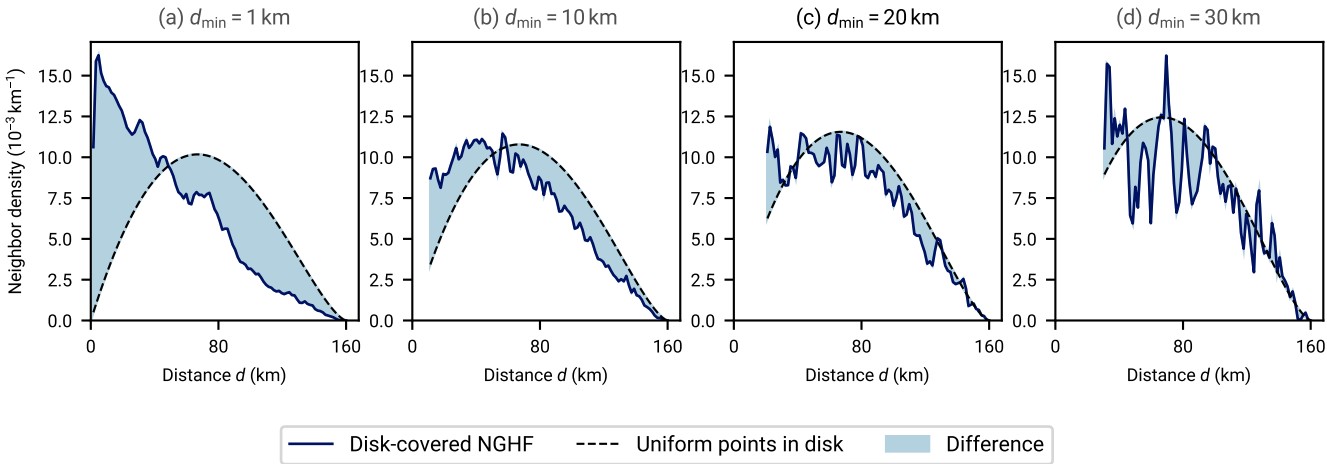

**Figure 3.** Spatial uniformity of heat flow measurements from NGHF within disks of radius 80 km when varying the minimum inter-point distance $d_{min}$. Each graph shows neighbor distributions within disks of radius 80 km as a function of inter-neighbor distance $d$, that is, the number of neighboring data points at distance $d$ from a data point within the disk, averaged over all points within the disks. The dashed lines show the expected distribution for a uniform distribution of points within the disks (derived in Appendix E). Deviations from the dashed lines indicate a non-uniform distribution of points within the disk. The blue solid lines show the empirical neighbor density obtained from disks of radius 80 km randomly distributed over Earth and selecting the NGHF data points within. The difference between the three panels lies in enforcing different minimum distances between NGHF data points. If two data points within a disk are closer together than the indicated $d_{min}$, a random one of them is removed. As $d_{min}$ is increased, the neighbor distributions approach uniformity but fluctuations due to small number of remaining points within the disks increase. In this work, we choose $d_{min} = 20$ km as a compromise between the two effects.

but smooth spatial heat flow field as described in the previous section, sampling in clusters that are too narrow might lead to correlated data. The statistical methods we develop in the following section, however, assume independence of the data.

To mitigate the potential bias of spatial clustering, we enforce a minimum distance $d_{min}$ between data points, using only one data point of pairs that violate this distance criterion. This realizes a more uniform spatial data distribution. In Fig. 3 we compare analytical expressions for the neighbor density under a uniform distribution (Appendix E) with the distance distribution between points of the filtered NGHF. The comparison leads us to choose a minimum distance of 20 km

$$d_{min} = 20\,\text{km} \tag{4}$$

between selected data points as a trade-off between uniform distribution and sufficient sampling.

Using only one data point of close pairs raises the question which data point to choose. Ignoring the other data point ensures that the dependency between the data points is avoided but it also results in loss of information about any spatially independent noise component. To retain the best of both worlds, we introduce a latent parameter that iterates all possible ways to select $d_{min}$-conforming subsets from the set of heat flow measurements in a region. Each value of the latent parameter therefore

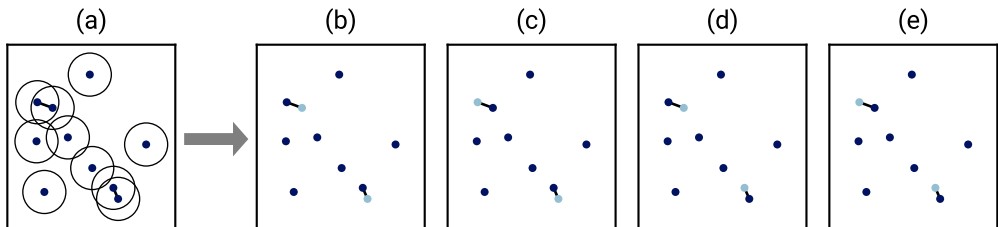

**Figure 4.** Selecting subsets of heat flow data when data point pairs violate the minimum distance criterion. The circles in panel (a) indicate the radius $d_{\min}$ which is violated by the two marked point pairs. Panels (b) to (e) show the data point subsets that would be used in the handling of spatial data clusters: of each conflicting pair, a maximum of one data point is retained (less if the violations occur in clusters). In this simple scenario, panels (b) to (e) list all possible permutations. The REHEATFUNQ code approximates this permutation procedure stochastically for large sample sizes.

corresponds to a data set that we consider independent data within our model assumption and we can evaluate posteriors as described in sections 3.3.1 and 3.4. Figure 4 illustrates the generated subsets for a simple example.

Choosing the parameter $d_{\min}$, whether according to our value of $20\,\mathrm{km}$ or based on the data density within the ROI, is step 2 of the workflow listed in Appendix A.

### 3.3 Model description

#### 3.3.1 A combined gamma model

The disaggregation of the heat flow measurements, Eq. (2), into different components is the basis for our model of regional

aggregate heat flow. In particular, we consider the unknown heat flow $q_u$ as a random variable. To yield useful results, this requires a model, that is, a probability distribution for $q_u$. In deriving a model for $q_u$, we make the following assumptions:

    I. The sum $q_b + q_f$ is i.i.d. gamma-distributed.

    II. The sum $q_s(\boldsymbol{x}) + q_t(\boldsymbol{x})$ is an i.i.d. gamma-distributed random variable if $\boldsymbol{x}$ is the random variable that is derived from the spatial distribution of the heat flow data after applying the minimum distance criterion (that is, successive point removal)

in the *right order*.

    III. The *right order* follows the uniform distribution of permutations of the ordering of the heat flow data.

When both $q_b + q_f$ and $q_s(\boldsymbol{x}) + q_t(\boldsymbol{x})$ are gamma-distributed, the resulting sum $q_u$ can be fairly well described by a gamma-distribution (Covo and Elalouf, 2014). The sum is exactly gamma distributed when both $q_b + q_f$ and $q_s(\boldsymbol{x}) + q_t(\boldsymbol{x})$ have the same scale parameter (Pitman, 1993). Hence, conditional on the *right order*, $q_u$ is assumed to be gamma distributed and the

likelihood of the remaining data points is the gamma likelihood. We can iterate the permutations of the ordering using a latent

parameter, the permutation index $j \in \{1, \ldots, N!\}$. The probability of $j$ is $P(j) = 1/N!$. The full likelihood is then

$$\mathcal{L}(j, \alpha, \beta \,|\, \{q_i\}) = \phi(\alpha, \beta) P(j) \prod_{i \in \mathcal{I}(j)} \gamma(q_i | \alpha, \beta) \tag{5}$$

where $\phi(\alpha, \beta)$ is the prior of the gamma distribution, $\mathcal{I}(j) = \{i\}_j$ is the set of indices of data points in permutation $j$ that are retained by the minimum distance selection algorithm (see section 3.2), and

$$\gamma(q_i | \alpha, \beta) = \frac{\beta^\alpha}{\Gamma(\alpha)} q_i^{\alpha-1} \exp(-\beta q_i), \tag{6}$$

with the gamma function $\Gamma(\alpha)$, is the gamma distribution for heat flow values $q_i > 0$. Here we have used the parameterzerization of the gamma distribution with shape parameter $\alpha$ and rate parameter $\beta$. An alternative parameterization uses the scale parameter $\theta = 1/\beta$ instead.

The likelihood, Eq. (5) warrants two comments on its structure: first note that $\mathcal{I}(j)$ always contains at least one index, the
250 start of the permutation, due to the iterative resolution of the minimum distance criterion. Secondly, if there is no conflicting pair, $\mathcal{I}(j)$ always contains all data indices and the $j$ dimension trivially collapses to a uniform distribution. Otherwise, $\mathcal{I}(j)$ can be a fairly complicated set.

Assumptions II. and III. are chosen peculiarly specific so as to yield a simple expression for the likelihood of the model. However, we can imagine a simple model of human data acquisition that is closely approximated by this likelihood. Imagine
that a set of heat flow measurements is generated by the following process: initial drilling operations are distributed uniformly randomly over an area. Given that the level set of the underlying heat flow field is gamma distributed (or can be closely approximated by a gamma distribution), these initial drillings are gamma distributed as laid out in section 3.1. Some of the initial wells turn out to be points of interest, for instance by identifying an oil or a geothermal field. Many of the following boreholes that lead to heat flow measurements would then cluster around these points of interest. This clustering, in turn, can
lead to bias in the regional aggregate heat flow distributions due the spatial correlation of $q_s(\boldsymbol{x})$. If we were to know the spatial extent of the clusters (say, disks of radius $d_{\min}$), and we assume that *a priori* each point within a cluster is equally likely the initial drilling, we could obtain the likelihood given in Eq. (5). In appendix C we confirm that this simple physically inspired sampling mechanism leads to estimation biases, and we have find that the minimum distance sampling used in REHEATFUNQ is an effective counter measure.

Assumption I, the use of the gamma distribution, is motivated by the general right-skewed shape of global heat flow (see panel (b) of Fig. 1), positivity of surface heat flow, and the existence of a conjugate prior (which greatly reduces the computational cost). Besides the aforementioned and rather subjective criteria, and to have an objective evaluation, we have performed goodness-of-fit tests (section 4.1.3) that show that the gamma distribution is at least as competitive as other simple probability distributions on the positive real line in terms of describing the regional aggregate heat flow distributions.

We restrict the parameter $\alpha$ to a minimum value $\alpha_{\min} = 1$ to prevent parameterizations with diverging density at $q \to 0$ (see panel (a) of Fig. 5). In this limit $\alpha = 1$ the gamma distribution is an exponential distribution. For smaller $\alpha$, the density has a singularity at $q = 0$. Illustratively, this causes the PDF to counter the effect of decreasing scale, and the mass decays only slowly on log scales in $q$.

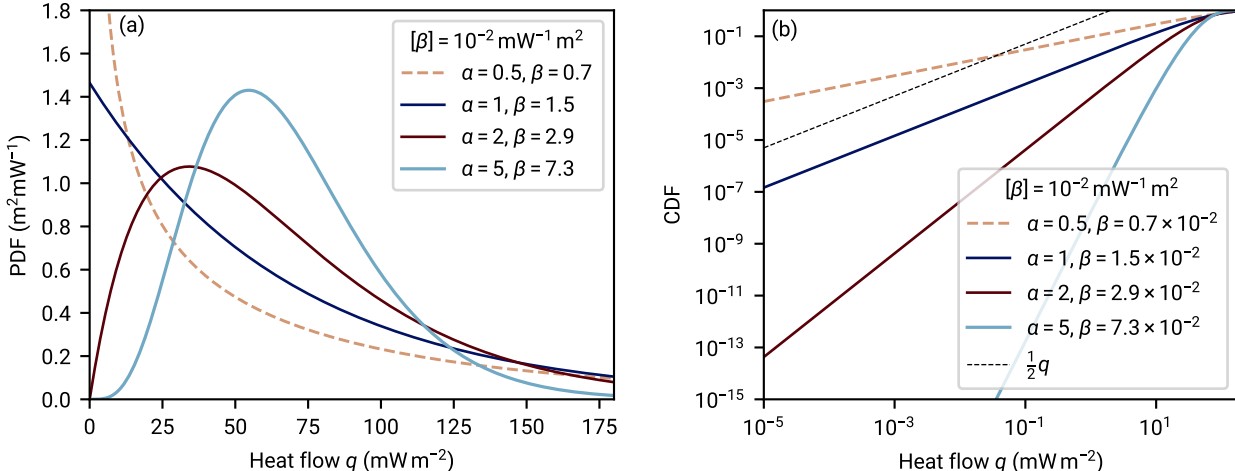

**Figure 5.** Gamma distribution as a model for regional aggregate heat flow. Panel (a) shows the probability density functions (PDFs) of four gamma distributions with varying shape parameter $\alpha$ that all have the same mean $\langle q \rangle = 68.3\,\mathrm{mW\,m^{-2}}$ (the mean of "A" quality data estimated by Lucazeau, 2019). For $\alpha = 1$, the PDF is finite for $q \to 0$. For $\alpha > 1$, this limit is zero, and for $\alpha < 1$, the PDF has a singularity for $q \to 0$. Panel (b) shows the corresponding cumulative distribution functions (CDFs) with an emphasis on the asymptotics for small $q$. A linear slope is plotted for comparison, corresponding to the growth of a uniform density with increasing integration interval. The increased mass located in small $q$ in case of $\alpha < 1$ becomes evident.

In the course of this manuscript, we will use the mean $\bar{q}$ and standard deviation $\sigma_q$ of the gamma distribution. Parameterized by $\alpha$ and $\beta$, they are given by (Thomopoulos, 2018)

$$\bar{q} = \frac{\alpha}{\beta} \qquad \text{and} \qquad \sigma_q = \frac{\sqrt{\alpha}}{\beta} . \tag{7}$$

For frequentist inference of $\alpha$ and $\beta$, the maximum likelihood estimator will be used a couple of times in this work. A Newton-Raphson iteration with starting values given by Minka (2002) is used.

For a Bayesian analysis of both the regional aggregate heat flow distributions and the fault-generated heat flow anomaly, a prior distribution of the parameters $\alpha$ and $\beta$ of the gamma distribution model is required—even if it is just the implicit improper uniform prior. Using an informative prior instead (see e.g. Zondervan-Zwijnenburg et al., 2017) opens up the potential to include information from outside sources in a regional analysis. Because the number of measurements in regional heat flow analysis is generally small, the $R = 80\,\mathrm{km}$ RGRDCs created from NGHF "A" quality data typically contain 31 disks with an average of 11 points per disk, additional information can be valuable.

Ideally, we would like the prior we use to be derived on physical grounds. We do not have any independent physical criteria for constructing the prior, but our empirical gamma distribution model aims to capture the predominant physics underlying regional aggregate heat flow (we will later investigate how much so). Hence, a physical basis that can guide our prior choice is the implied physics captured by our gamma distribution model. A prior that is constructed from the gamma distribution is the empirically best choice to reflect these underlying physics. This role is generally fulfilled by conjugate priors which arise from

the associated probability density functions and whose hyperparameters represent aspects of the data that can become evident in the Bayesian updating.

For the gamma distribution, a conjugate prior is given by Miller (1980), parameterized by the hyperparameters $p$, $s$, $n$, and $\nu$. Its probability density in gamma distribution parameters $(\alpha, \beta)$ is

$$\phi(\alpha, \beta \,|\, p, s, n, \nu) = \frac{\beta^{\nu\alpha - 1} p^{\alpha - 1} \exp(-s\beta)}{\Gamma(\alpha)^n \Phi(p, s, n, \nu)} \tag{8}$$

with gamma function $\Gamma(\alpha)$ and

$$\Phi(p, s, n, \nu) = \int\limits_{a_{\min}}^{\infty} \mathrm{d}\alpha \frac{p^{\alpha - 1} \Gamma(\nu\alpha)}{\Gamma(\alpha)^n s^{\nu\alpha}}. \tag{9}$$

As in the previous section, we restrict the range of $\alpha$ from $a_{\min} = 1$ to infinity to exclude probability densities that diverge at $q \to 0$ and place considerable weight into negligible heat flow (say $q < 10^{-2}\ \mathrm{mW\,m^{-2}}$).

The conjugate prior facilitates the computation of posteriors by means of Bayesian updating. The numerically expensive integrations over the parameter space of $\alpha$ and $\beta$ that are involved in computing the posterior are reduced to simple algebraic update rules of the conjugate prior's parameters. Since numerical quadrature is the leading computational cost in REHEAT-FUNQ and grows exponentially with the number of quadrature dimensions, reducing the set of quadratures to the computation of the normalization constant $\Phi$, Eq. (9), significantly benefits the performance.

The Bayesian updating of the prior Eq. (8) given a sample $Q = \{q_i : 1 \leq i \leq k\}$ of $k$ heat flow values is (Miller, 1980)

$$p^* = p \prod_i q_i, \qquad s^* = s + \sum_i q_i, \qquad n^* = n + k, \qquad \nu^* = \nu + k. \tag{10}$$

The posterior distribution of $\alpha$ and $\beta$ is hence Eq. (8) with the starred parameters given above.

Given a prior parameterization $(p, s, n, \nu)$ the probability density of heat flow within the region is the predictive distribution

$$\psi(q \,|\, p, s, n, \nu) = \int\limits_{\alpha_{\min}}^{\infty} \mathrm{d}\alpha \int\limits_{0}^{\infty} \mathrm{d}\beta\, \gamma(q \,|\, \alpha, \beta) \phi(\alpha, \beta \,|\, p, s, n, \nu) = \frac{\Phi(pq, s + q, n + 1, \nu + 1)}{\Phi(p, s, n, \nu)}. \tag{11}$$

Here, the final step utilizes the conjugate structure of the prior.

This expression can be translated to the likelihood Eq. (5) of the REHEATFUNQ model. The Bayesian update leads to the following proportionality:

$$\psi(q \,|\, Q, p, s, n, \nu) \sim \sum_{j=1}^{m} \int\limits_{\alpha_{\min}}^{\infty} \mathrm{d}\alpha \int\limits_{0}^{\infty} \mathrm{d}\beta\, \gamma(q \,|\, \alpha, \beta) \mathcal{L}(j, \alpha, \beta \,|\, \{q_i\}). \tag{12}$$

After some algebra used in Eq. (11), this resolves to

$$\psi(q \,|\, Q, p, s, n, \nu) = \frac{\sum\limits_{j=1}^{m} \Phi(p_j^* q,\ s_j^* + q,\ n_j^* + 1,\ \nu_j^* + 1)}{\sum\limits_{j=1}^{m} \Phi(p_j^*,\ s_j^*,\ n_j^*,\ \nu_j^*)} \tag{13}$$

where $p_j^*$, $s_j^*$, $n_j^*$, and $\nu_j^*$ are the parameters updated according to Eq. (10) with heat flow data set $\mathcal{I}(j)$.

### 3.3.2 Minimum surprise estimate

From a non-technical point of view, the purpose of the gamma conjugate prior in the heat flow analysis is to transport universal information about surface heat flow on Earth while at the same time not significantly favoring any particular heat flow regime above other existing regimes. In other words, the prior should penalize regions of the $(\alpha, \beta)$ parameter space that do not exist on Earth but should be rather uniform throughout the parts of $(\alpha, \beta)$ space that occur on Earth. The uniformity ensures that all regions on Earth are treated equally *a priori* in terms of heat flow while the penalty adds universal information that can augment the aggregate heat flow data of each region.

In practice, a compromise between the uniform weighting of existing aggregate heat flow distributions and the penalizing of non-existent parameterizations needs to be found. Our choice in REHEATFUNQ is to put more weight onto the *a priori* uniformity of regional characteristics, that is, less bias. In parlance, we want to be *minimally surprised* by any of the distributions of the RGRDCs if we start from the prior distribution. One notion of the *surprise* contained in observed data when starting from a prior model is the Kullback-Leibler divergence (KLD) $K$ from the prior distribution to the posterior distribution given the data (Baldi, 2002)

$$K(p(\boldsymbol{x}), \phi(\boldsymbol{x})) = \int\limits_{\mathcal{X}} \mathrm{d}^n x \, p(\boldsymbol{x}) \ln \frac{p(\boldsymbol{x})}{\phi(\boldsymbol{x})}, \tag{14}$$

where $\mathcal{X}$ is the support of the parameters $\boldsymbol{x}$. Baldi (2002) defines $p(\boldsymbol{x})$ to be the prior distribution and $\phi(\boldsymbol{x})$ the posterior distribution given a set of observations.

The KLD is asymmetric and Baldi and Itti (2010) note that the alternate order of probability distributions "[. . . ] may even be slightly preferable in settings where the 'true' or 'best' distribution is used as the first argument". Here, we follow the alternate order and assign the gamma conjugate prior to the role of $\phi(\alpha, \beta)$. Furthermore, for the purpose of estimating the gamma conjugate prior's parameters we consider the "uninformed" prior ($p = 1$, $s = n = \nu = 0$; Miller, 1980) updated to a regional heat flow data set using the update rules Eq. (10) to be the "true" distribution $p(\alpha, \beta)$ within that region. In this order, minimizing the KLD Eq. (14) is also known as the "principle of minimum discrimination information" (Kullback, 1959; Shore and Johnson, 1978, MDI hereafter), closely related to the "principle of maximum entropy" (Shore and Johnson, 1978).

Applying this estimator to a set of regional aggregate heat flow distributions leads us to the following cost function of *minimum surprise* which we aim to minimize. We enumerate the regional aggregate heat flow distributions of the RGRDC by index $i$ and the heat flow values within by index $j$ with $i$-dependent range ($Q_i = \{q_j\}_i$). We compute the updated parameters $p_i^*$, $s_i^*$, $n_i^*$, and $\nu_i^*$ starting from $p = 1$, $s = n = \nu = 0$ for each $Q_i$. Then, the cost function $\bigotimes$ reads

$$\bigotimes(p, s, n, \nu \,|\, \{Q_i\}) = \max_i \left\{ \int\limits_{\alpha_{\min}}^{\infty} \mathrm{d}\alpha \int\limits_{0}^{\infty} \mathrm{d}\beta \, \phi(\alpha, \beta \,|\, p_i^*, s_i^*, n_i^*, \nu_i^*) \ln\left( \frac{\phi(\alpha, \beta \,|\, p_i^*, s_i^*, n_i^*, \nu_i^*)}{\phi(\alpha, \beta \,|\, p, s, n, \nu)} \right) \right\}. \tag{15}$$

On an algebraic level, the $i$th KLD term emphasizes scale differences between the prior and the $i$th regional data-driven distribution in parts of the $(\alpha, \beta)$-space which the regional data favor, while other parts of the parameter space are less important. Taking the maximum over the distributions $\{i\}$ ensures that across distributions, the regions in which probability mass is

**Table 1.** Parameter bounds used in the minimum surprise estimate optimization.

| Parameter | Value |
|---|---|
| $p_{\min}$ | 1 |
| $p_{\max}$ | $10^5$ |
| $s_{\min}$ | 0 |
| $s_{\max}$ | $10^3$ |
| $\nu_{\min}$ | $2 \cdot 10^{-2}$ |
| $\nu_{\max}$ | 1 |
| `nv_surplus_min` | $10^{-8}$ |
| `nv_surplus_max` | 2 |

concentrated are equally accurately represented. Another advantageous property of the MDI estimator is that by taking into consideration the full probability mass, it can be better-suited for small sample sizes than point estimators (e.g. Ekström, 2008).

An explicit expression for the numerical quadrature of Eq. (15) is given in Appendix D1.1. For the purpose of optimization, we substituted the parameters

$$\boldsymbol{x} = \left( \ln p, s, \ln\left(\frac{n}{\nu} - 1\right), \nu \right) \tag{16}$$

with boundaries

$$\ln p_{\min} \leq x_0 \leq \ln p_{\max}, \quad s_{\min} \leq x_1 \leq s_{\max}, \quad \ln(\texttt{nv\_surplus\_min}) \leq x_2 \leq \ln(\texttt{nv\_surplus\_max}), \quad \nu_{\min} \leq x_3 \leq \nu_{\max}.$$

With adjustable parameters

$$0 < p_{\min} < p_{\max}, \quad 0 < s_{\min} < s_{\max}, \quad 0 < \nu_{\min} < \nu_{\max}, \quad \text{and} \quad 0 \leq \texttt{nv\_surplus\_min} \leq \texttt{nv\_surplus\_max},$$

this substitution ensures that the parameter bounds in $p$, $s$, $n$, and $\nu$ are adhered to. Choosing to optimize the logarithm of $\frac{n}{\nu} - 1$ has shown itself to lead to a gracious convergence to the $n = \nu$ limiting case, and $\ln p$ is the standard expression of the $p$ parameter in the numerical backend (see Appendix D).

Before the global optimization of $p$, $s$, $n$, and $\nu$, it is helpful to determine some *a priori* bounds on the parameters. One observation is that the MSE should not introduce a strong bias to the regional results. Miller (1980) noted that the parameterization in which the updated posterior parameters are dependent on the data only is the "uninformed" prior $p = 1$, $s = n = \nu = 0$. This line of thought leads to heuristic bounds on the parameters for the MSE. The posterior update rule for $n$ and $\nu$ is an increment by the data count. Hence, the prior $\hat{n}$ and $\hat{\nu}$ should be smaller than or close to 1 for our desired MSE if we expect less than one data point of "information". For $p$, the update rule is a product with each heat flow value $q_i$ and for $s$ it is the sum with $q_i$. Hence, $\hat{p}$ is expected not be larger than $250^k$, with, say, $k \sim 1$, and $\hat{s}$ not larger than $250k$. We have chosen conservative bounds based on these estimates and use the parameter bounds shown in Tab. 1.

To perform a global optimziation of Eq. (15), we employ the Simplicial Homology Global Optimization (SHGO) algorithm implemented in SciPy (Endres et al., 2018; Virtanen et al., 2020). This algorithm starts with a uniform sampling of a compact

multidimensional parameter space (we use the simplicial sampling strategy). The cost function is evaluated at the sample points
and a directed graph, approximating the cost function, is created by joining a Delaunay triangulation of the sample points with
directions of cost increase. The key step of SHGO is then to determine local minimizers of this graph as starting points for
further local optimization. The power of the algorithm is that under the condition that the cost function is Lipschitz continuous
and the parameter space has been sampled sufficiently (whereby the directed graph is a sufficient representation of the cost
function), the SHGO algorithm generates exactly one such starting point per local minimum. For the final iterative optimization
of SHGO, we use the Nelder-Mead simplex algorithm (Nelder and Mead, 1965; Virtanen et al., 2020).

From manual investigation, we have found that setting the iteration parameter of SHGO to three, and using the boundaries
previously defined, we obtain the optimum

$$\hat{p} = 2.52202, \quad \hat{s} = 15.3730, \quad \hat{n} = 0.218477, \quad \hat{\nu} = 0.218477 \tag{17}$$

with a final cost $\diamondsuit = 4.496$. Two-dimensional slices of the local neighborhood of this optimum are displayed in Fig. S1 of the
supplement.

The prior $\phi(\alpha, \beta)$ with our MSE parameters is shown in Fig. 6. There, we also show the maximum likelihood point estimates
$(\hat{\alpha}, \hat{\beta})$ for each of the regional aggregate heat flow distributions $\{Q_i\}$ from the RGRDC used in the prior parameter MSE. The
shape of the prior in Fig. 6 (a) does not follow the scatter of the $(\hat{\alpha}, \hat{\beta})$ estimates: while the $(\hat{\alpha}, \hat{\beta})$ are, on logarithmic scales,
within a constant range of a linear slope across scales, the prior widens on log scales with decreasing $\alpha$ and $\beta$. The picture
changes when considering the estimate uncertainties which also increase with respect to the scatter of estimates for decreasing
$\alpha$ and $\beta$ (Fig. 6 (b)). The prior thus captures the effects of the gamma distribution parameters and the parameters' sensitivities
for different $\alpha$ and $\beta$.

With respect to heat flow, this implies that the average heat flow, Eq. (7), is fairly constant for any heat flow distribution.
However, the sensitivity of the overall distribution relative to the distribution parameters—and consequently the uncertainty of
the distribution estimates—changes with the distribution parameters. This sensitivity is relatively lower at smaller parameter
values and vice versa. If a resulting distribution is less sensitive on the parameters, then in turn the uncertainties of estimating
the parameters of such a distribution will increase, as even a large change in parameters will result only in a minor change of
the resulting distribution. The prior reflects this behavior.

Equation (11) for the posterior predictive distribution of regional heat flow can also be evaluated for the non-updated prior.
Fig. 7 shows the PDF and the CDF for the prior parameters Eq. (17). The mode of the PDF is close to the average heat flow of
"A" quality data within the NGHF, $68.3 \, \mathrm{mW \, m^{-2}}$ (Lucazeau, 2019). The prior predictive CDF follows fairly closely the median
CDF of the RGRDC samples, with the exception of heat flow exceeding about $100 \, \mathrm{mW \, m^{-2}}$. The latter is linked to the heavy
tail of the PDF, which aggregates about 4.3 % probability, while the data are cut at $250 \, \mathrm{mW \, m^{-2}}$.

## 3.4  Bayesian inference of heat flow anomaly strength

We now turn to the quantification of the heat flow anomaly $q_a(\boldsymbol{x})$. This signal $q_a(\boldsymbol{x})$ is the surface heat flow signal due to a
specific heat source that a researcher would like to investigate. It is implied that the surface heat flow field due to the heat source

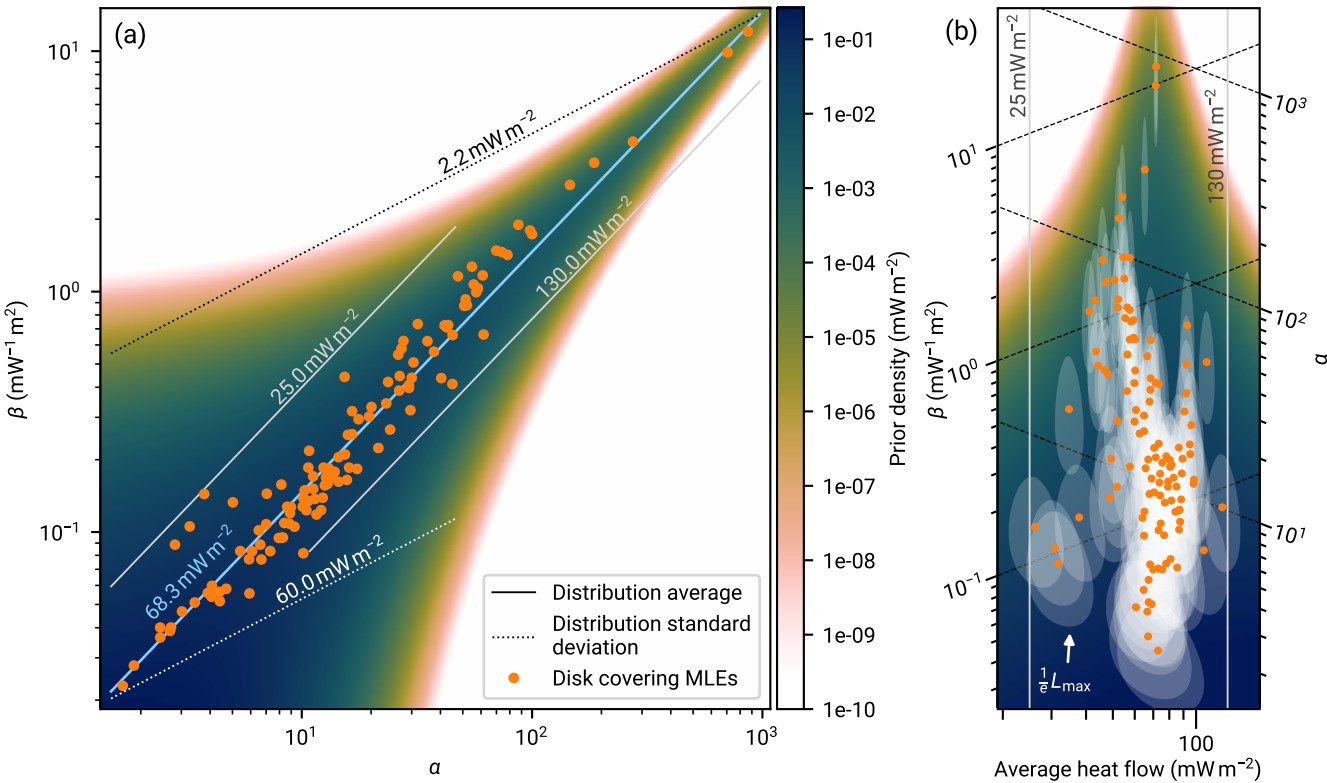

**Figure 6.** Analysis of the global heat flow database in $mW\,m^{-2}$ and parameter estimate of the gamma conjugate prior Eq. (8). Each dot marks the maximum likelihood estimate (MLE) of the gamma distribution parameters $(\alpha_i, \beta_i)$ for one of the randomly selected disk regions shown in Fig. 1 with selection criterion of section 2 applied. The solid lines mark parameter combinations of equal mean heat flow, and the dashed lines those with equal standard deviation (after Thomopoulos, 2018). If the gamma distribution is assumed, global heat flow split into 80 km radius disks can typically be described within a band of the parameter space given by distribution average between 25 and $120\,mW\,m^{-2}$ and standard deviation 3 to $60\,mW\,m^{-2}$. We capture this using the gamma conjugate prior Eq. (8) shown in background color. Its parameters $\hat{p}=2.52202$, $\hat{s}=15.3730$, $\hat{n}=0.218477$, $\hat{\nu}=0.218477$ stem from the minimum surprise estimate described in section 3.3.2. Panel (a): the global mean continental heat flow of $68.3\,mW\,m^{-2}$ is the estimate of Lucazeau (2019) from "A" quality data. Panel (b) shows a rotated and stretched section of the $(\alpha, \beta)$ parameter space such that the ordinate axis coincides with the average heat flow levels. The data are the same as in panel (a). Additionally we show, for each MLE, the region of the parameter space in which the corresponding likelihood is larger than $1/e$ its maximum. This illustrates the uncertainty of the parameter estimates.

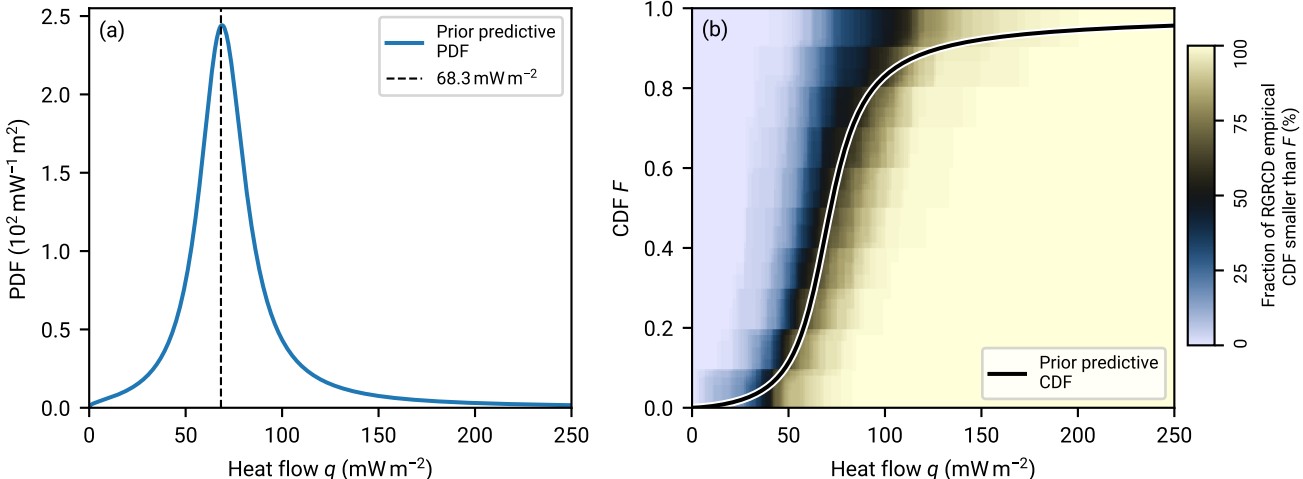

**Figure 7.** Prior predictive for regional aggregate heat flow. The gamma conjugate prior is parameterized as described in Eq. (17). Panel (a) shows the prior predictive PDF. The average value of "A" quality data from the NGHF (Lucazeau, 2019) is indicated. Panel (b) shows the prior predictive CDF. The background color shows, for each pixel in $(q, F)$ coordinates, the fraction of empirical cumulative distribution functions computed from the RGRDC heat flow samples at heat flow $q$ which exceed $F$.

can be computed. In this article, we will use the heat flow signature of a vertical strike-slip fault with linearly increasing heat production with depth (Lachenbruch and Sass, 1980), but in principle REHEATFUNQ is agnostic to the type of surface heat flow to separate from the regional scatter. As noted in section 3.1, the signal $q_a$ can be separated from the regional undisturbed
heat flow by means of Eq. (3) if the heat source is weak enough not to incite nonlinear convection.

In REHEATFUNQ, the heat flow anomaly signal $q_a(x)$ is expressed by the total heat power $P_H$ that characterizes the heat source, and a location-dependent heat transfer function $c(x)$ that models the surface heat flow per unit power that is caused by the heat source. This transfer function follows by solving the relevant heat transport equation. Given a power $P_H$ and a function $c(x)$, the heat flow anomaly contribution to the heat flow at measurement location $x_i$ is thereby

$$q_a(x) = P_H c(x_i) = P_H c_i \tag{18}$$

Providing the coefficients $c_i$ for each data point, by means of whichever solution technique to the heat transport equation available, is thereby the "application interface" of the REHEATFUNQ model for heat flow anomaly quantification. This is step 4 of the workflow listed in Appendix A. Note that while Eq. (18) requires the heat transport to be linear in $P_H$, in Appendix G we note a particular case of non-linearity in the heat transport with respect to $P_H$ that can still be addressed by REHEATFUNQ.
We can now combine the stochastic model for $q_u$ and the deterministic model for $q_a(P_H)$. Treating $P_H$ as a model parameter, we perform Bayesian inference using the gamma distribution model for $q_u$. First, we transform the heat flow measurements by removing the influence of the heat flow signature:

$$q_i' = q_i - P_H c_i. \tag{19}$$

The data $q_i'$ are now data of the "unknown" or "undisturbed" heat flow, for which we use the gamma model $\gamma(q_u)$ and its conjugate prior.

Assuming the heat flow anomaly to be generated by a heat source implies the lower bound $P_H \geq 0$ (zero if the heat flow data is not at all compatible with the anomaly). From Eq. (19) an upper bound on $P_H$ follows. Since we consider only positive heat flow,

$$P_H^{(j)} \leq \min_{i \in \mathcal{I}(j)} \left\{ \frac{q_i}{c_i} \right\} =: P_H^{\mathrm{m},j} \tag{20}$$

for any heat flow sample iterated by $j$. Outside of these bounds, we assume zero probability for this value of $j$. The global maximum $P_H$ that can be reached across all $j$ is

$$P_H^{\max} := \max_{1 \leq j \leq m} \min_{i \in \mathcal{I}(j)} \left\{ \frac{q_i}{c_i} \right\}. \tag{21}$$

Assuming a uniform prior in $P_H$ within these bounds, the full posterior of the REHEATFUNQ anomaly quantification reads

$$f\big(P_H, j, \alpha, \beta \,|\, p, s, n, \nu, \{(q_i, c_i)\}\big) \sim \phi\big(\alpha, \beta \,|\, p, s, n, \nu\big) \prod_{i \in \mathcal{I}_j} \gamma\big(q_i - P_H c_i \,|\, \alpha, \beta\big). \tag{22}$$

To quantify the heat power $P_H$, REHEATFUNQ uses the marginal posterior in $P_H$:

$$f\big(P_H \,|\, p, s, n, \nu, \{(q_i, c_i)\}\big)$$
$$= \begin{cases} \dfrac{1}{\mathcal{F}} \displaystyle\sum_{j=1}^{m} \int_{\alpha_{\min}}^{\infty} \mathrm{d}\alpha \int_0^{\infty} \mathrm{d}\beta \, \phi(\alpha, \beta) \prod_{i \in \mathcal{I}_j} \gamma\big(q_i' \,|\, \alpha, \beta\big) & : P_H \in [0, P_H^{\max}] \\ 0 & : \text{otherwise.} \end{cases} \tag{23}$$

In Appendix D2, we discuss how to compute the normalization constant $\mathcal{F}$.

If an upper bound on the heat power $P_H$ is the aim of the investigation, the tail distribution (or complementary cumulative distribution function)

$$\bar{F}(P_H) = \int_{P_H}^{\infty} \mathrm{d}P \, f(P) \tag{24}$$

can be used. It quantifies the probability with which the heat-generating power is $P_H$ or larger.

An illustration of the idea behind the approach in Eq. (23) is shown in Fig. 8. Panel (a) shows a sample of undisturbed heat flow $q_u$ drawn from a gamma distribution. This heat flow is superposed with the conductive heat flow anomaly from a vertical strike-slip fault (Lachenbruch and Sass, 1980). The result is the sample of "measured" heat flow $q$. Undisturbed data at the center of the heat flow anomaly are collectively shifted to higher heat flow values while those further away from the fault are barely influenced. Within the regional aggregate heat flow distribution, the most-affected data will be shifted towards the tail. This distortion of the aggregate heat flow distribution is picked up by the likelihood with the result that correcting for the heat flow anomaly of the right power $P_H = 140\,\mathrm{MW}$ (transforming the dots back to triangles in panel (a)) is more likely than no anomaly ($P_H = 0\,\mathrm{W}$) in the right panel.

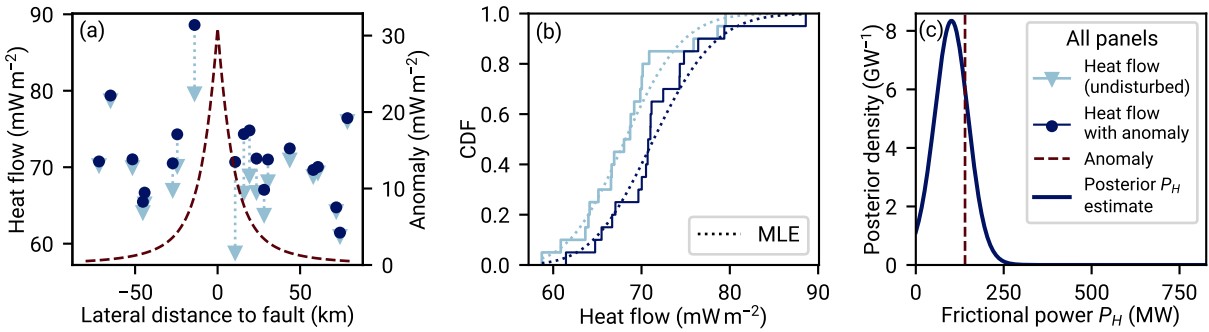

**Figure 8.** Sketch of the Bayesian analysis of the fault-generated heat flow anomaly strength. The analysis starts out in panel (a) with heat flow measurements (dots) in spatial relation to a known strike-slip fault. The heat flow measurements within the investigated region fluctuate, and they are distributed according to a probability distribution $p(q)$. Here we use a gamma distribution with $\alpha = 180$ and $\beta = 2.6354319\,\mathrm{mW^{-1}\,m^2}$. These undisturbed fluctuations (triangles) are superposed by the fault-generated conductive heat flow anomaly (dashed line) to yield the measurements. Both the undisturbed data and the anomaly's strength are unknown to the researcher but the anomaly can be modeled as a function of average frictional power $P_H$. Panel (b) shows the difference in aggregate cumulative distribution of the undisturbed heat flow and the data superposed by the anomaly. This is how REHEATFUNQ "sees" the data. Panel (c) shows the result of the REHEATFUNQ analysis. Our approach investigates the continuum of $P_H$. Each $P_H$ corresponds to a heat flow anomaly of different amplitude, which leads to different corrected data (from circles to triangles in panel (a)). The likelihood of the corrected data is evaluated against our proposed model of $p(q)$, a gamma distribution, which leads to the posterior of frictional power. In case of this synthetic gamma-distributed data, the actual anomaly strength (vertical dashed line) is well assessed.

Fig. 8 illustrates a core difficulty when identifying heat flow anomalies within noisy data of small sample sizes. The strength of the heat flow anomaly in this case is comparable to the intrinsic scatter of the regional aggregate heat flow distribution. This makes it difficult to identify the anomaly shape within the data. If the variance of the undisturbed heat flow is small compared to the actual magnitude of the anomaly, it becomes more and more feasible to visually identify the correct anomaly strength. Especially if the sample size is small, however, allowing for the occurrence of random fluctuations can significantly alter the interpretation of the data. The Bayesian analysis can capture all of this uncertainty in the posterior distribution of $P_H$, yielding a powerful analysis method.

### 3.4.1 Providing heat transport solutions

As outlined in section 3.1, steady crustal heat transport can be conductive, advective, or convective. The REHEATFUNQ model can be applied as long as the surface heat flow at the data locations is linear in the frictional power $P_H$ on the fault. The whole, potentially complicated model of heat conduction from the fault to the data points can then be abstracted to the coefficients $\{c_i\}$. At present, the task of computing these coefficients for use in REHEATFUNQ lies generally with the user (step 4 of the workflow listed in Appendix A). Numerical methods such as the finite element, finite difference, or finite volume method as well as analytical solutions to simplified problem geometries can be used to determine $\{c_i\}$ for a given problem by solving the

heat transport equation of heat generated on the fault plane and dividing the surface heat flow at the data locations by the total frictional power $P_H$ on the fault.

We illustrate this process using the single solution to the heat conduction equation that REHEATFUNQ presently implements: the surface heat flow anomaly generated by a vertical strike-slip fault. The solution stems from Lachenbruch and Sass (1980) and assumes a vertical fault in a homogeneous half-space medium. Furthermore, the fault is assumed to reach from depth $d$ to the surface and heat generation is assumed to increase linearly with depth up to a maximum $Q^*$ at depth $d$. In the limit of infinite time, the stationary limit, the anomaly then reads

$$q_a(x) = \frac{Q^*}{\pi}\left(1 - \frac{x}{d}\arctan\frac{d}{x}\right) \quad \text{for} \quad t \to \infty. \tag{25}$$

This shape of surface heat flow is shown in the sketch Fig. 8. In REHEATFUNQ, the anomaly is implemented based on the surface fault trace. For each data point, the distance to the closest point on this fault segment string is computed and inserted as $x$ into Eq. (25). For an infinite straight fault line in a homogeneous half space, this coincides exactly with the analytic solution. In real-world applications, the quality of this approximation depends on the straightness of the fault and its length compared to depth and data distance from the fault, as well as the dip of the fault—shallow-dipping faults lead to asymmetric heat flow instead.

The model (25) leads to a heat production $\bar{Q}d = Q^*d/2$ per unit length of the fault. We can balance this with the total heat dissipation power $P_H$ on a fault segment of length $L$ within a region:

$$Q^*d = 2\bar{Q}d = 2\frac{P_H}{L}. \tag{26}$$

This finally leads to the following expression of the coefficients $c_i$ as a function of distance to the surface fault trace:

$$c_i = \frac{q_a(x_i)}{P_H} = \frac{2}{\pi dL}\left(1 - \frac{x_i}{d}\arctan\frac{d}{x_i}\right) \quad \text{for} \quad t \to \infty. \tag{27}$$

To use the surface heat flow signature of other heat sources, or to include advection, one would perform similar steps. First, the heat transport equation needs to be solved. An analytical solution like (25) will not often be available, so that numerical techniques can be used to directly compute $q_a(x_i)$ at the data locations for a given $P_H$. Then $c_i$, the input values to the posterior (23), can be computed by dividing the $q_a(x_i)$ by $P_H$. The Python class `AnomalyNearestNeighbor` can then be used to specify the $c_i$ for use in the REHEATFUNQ Python module.

### 3.4.2 Heat transport uncertainty

The model for heat transport will in general be uncertain. For instance, in Eq. (26) one might be able to narrow down the depth $d$ only to within a certain range. Or, one might have an alternative model based on a different geometry, and perhaps another one that includes a small amount of ground water advection. Such uncertainties in parameter values and model selection can be accounted for in REHEATFUNQ.

The interface to do so is via the coefficients $c_i$. The user can provide a set

$$\mathfrak{C} = \left\{(w_k, \{c_i\}_k) : k = 1, \ldots, K\right\} \tag{28}$$

of $K$ solutions to the heat transport from source to heat flow data points. Each set $\{c_i\}_k$ should contain a number $N$ of coefficients $c_i$ equal to the total number of heat flow data points before applying the $d_{\min}$ sampling (effectively this is a $K \times N$ matrix $(c_{ki})$). The weights $w_k$ quantify the probability that the user assigns to the heat transport solution $k$. In this way, $k$ iterates a discretization of the $N$-dimensional probability distribution of the coefficients $c_i$.

Internally, REHEATFUNQ then uses a latent parameter $l \in \{1, \ldots, m \times K\}$ to iterate the combinations of the latent parameter $j$ with the index $k$ (another latent parameter). Then, in all previous equations, the index $j$ is replaced with $k$, the sets $\mathcal{I}_j$ with the set $\mathcal{I}_{j(k)}$ belonging to the index $j$ that $k$ iterates, and the coefficients $c_i$ are replaced by $c_{ki}$. This effectively adds the $k$-dimension to the REHEATFUNQ posterior.

Since $m \times K$ is a possibly very large number—even $j$ itself may be too large to iterate exhaustively and would hence be Monte Carlo sampled—only a user-provided maximum number of random indices $l$ will be used in the sums.

## 4 Model validation and limitations

The previous methodology section described the idea behind considering regional aggregate heat flow as a random variable, and set out straightforwardly to describe the REHEATFUNQ gamma model and its prior parameter estimation. Yet, no physical basis has been provided for the choice of a gamma distribution besides a number of general properties that the gamma distribution, among others, fulfills. In this section, we provide *a posteriori* support for this choice.

In the sections 4.1.1–4.1.3, the NGHF (Lucazeau, 2019) will be used to investigate whether the REHEATFUNQ gamma model is suitable for the description of real-world heat flow data. The analysis reveals a degree of misfit for which we investigate possible causes. Finally, we compare the gamma model to other two-parameter univariate probability distributions.

In section 4.2 and its subsections, we analyze synthetic data, which allows us to leverage large sample sizes. We investigate how well REHEATFUNQ can quantify heat flow anomalies both if the regional aggregate heat flow were gamma distributed, that is, according to the model assumptions, and if the regional heat flow were to follow some strongly gamma-deviating mixture distributions found in the NGHF in section 4.1.1. Furthermore, we investigate the impact of the prior parameters on the anomaly quantification.

In section 4.3 and its subsections, we discuss some physical limitations of the REHEATFUNQ model.

### 4.1 Validation using real-world data

#### 4.1.1 Goodness of fit: region size

Interpreting the regional heat flow as a stochastic, fluctuating background heat flow introduces a potential trade off in the region size. On one hand, considering heat flow data points across a larger area increases the number of heat flow measurements, which can increase the statistical significance of the analysis. In particular when investigating fault-generated heat flow anomalies, data points further away from the fault, say >20 km (see Fig. 8) are less influenced by the fault heat flow and can hence better quantify the background heat flow that is not disturbed by the heat flow anomaly. On the other hand, increasing the region

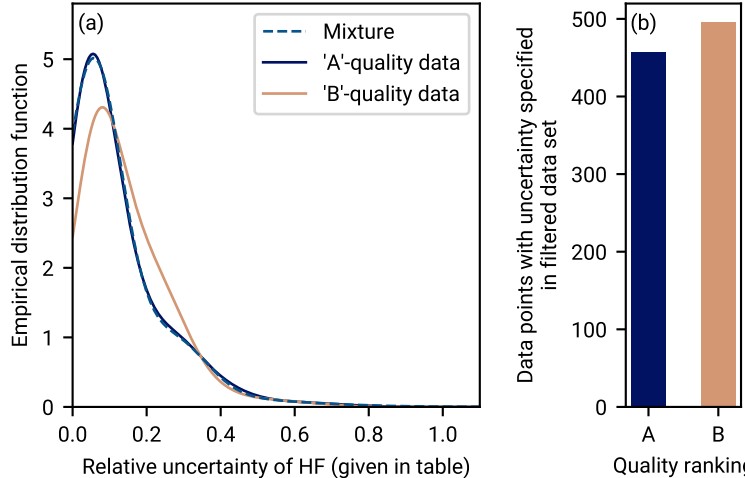

**Figure 9.** Distribution of relative error in the NGHF data base for "A" and "B" quality data. We show only "A" and "B" quality data according to our data filtering described in section 2.1. Panel (a) shows the distribution of relative uncertainty for data records of "A" and "B" quality, from the filtered NGHF data base, for which an uncertainty is specified. The dashed line shows a mixture distribution of three normal distributions that approximates the relative error distribution of the "A" quality data. The parameters of the mixture are means $\mu = (0.055, 0.272, 0.36)$, standard deviations $\sigma = (0.08, 0.09, 0.24)$, and weights $w = (0.79, 0.15, 0.06)$. Panel (b) shows a histogram of the number of data records of "A" and "B" for which uncertainty is specified and which pass our data selection criteria .

size makes the analysis more susceptible to capturing large scale spatial trends. These trends may introduce correlations or clustering between the data points which are not captured by the stochastic model. Conversely, using smaller region extents will improve the quality of approximating large-scale spatial trends as uniform. We will now set out to find a compromise
between these effects by finding a region size in which we can hope to apply the gamma model for regional aggregate heat flow distributions.

Our goodness-of-fit analysis by region size region is based on RGRDCs (see Appendix B). For each regional aggregate heat flow distribution, we investigate how well the sample can be described by the gamma distribution. Control over the radius $R$ allows us to investigate the fit over various spatial scales.

We performed tests based on the empirical distribution function (EDF tests, Stephens, 1986) to investigate the goodness-of-fit. We have used the Komogorov-Smirnov (KS) and Anderson-Darling (AD) test statistics, and we have applied them for the case that both parameters $\alpha$ and $\beta$ are unknown ("case 3" of Stephens, 1986). We calculated critical tables for the test statistics covering the sample sizes and maximum likelihood estimate shape parameters $\tilde{\alpha}$ that we encountered in the RGRDCs since the tables are independent of $\beta$ as a scale parameter (David and Johnson, 1948). The critical tables yield the values for both
test statistics that are exceeded at a certain rate if the data stem from a gamma distribution (we chose a 5% rejection rate). This rejection rate means that if $N$ samples of size $M$ are drawn from a gamma distribution with shape $\alpha$, the KS test statistic exceed the value read from the KS critical table for that $M$ and $\alpha$ in 5 % of the samples. The same holds for the AD statistic and the AD critical table. Hence if regional aggregate heat flow distributions were gamma distributed, we would expect 5% of

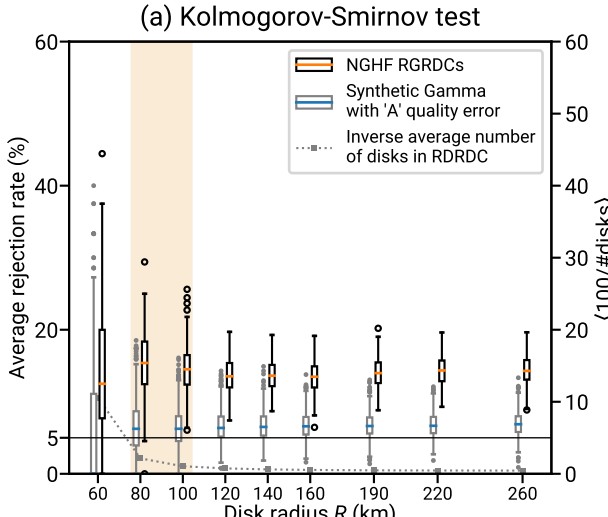 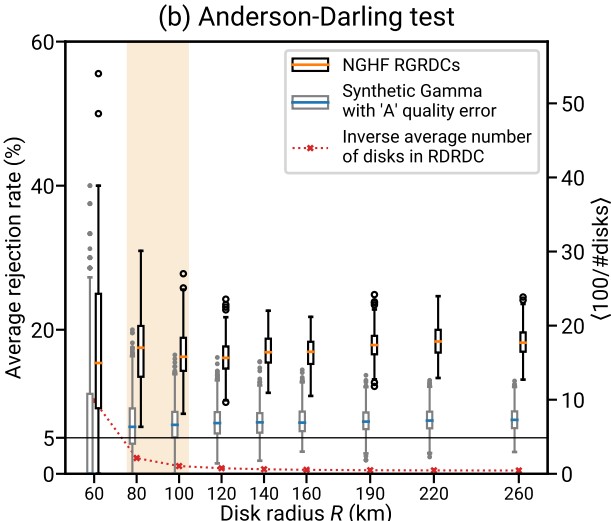

**Figure 10.** Investigating the fit of various probability distributions to the NGHF (Lucazeau, 2019) at different spatial scales. Both sides analyze the same data sets using goodness-of-fit (GOF) tests for the gamma distribution (e.g. Stephens, 1986). First, we analyze RGRDCs of the NGHF data set (defined in Appendix B). For each disk of a global covering, a GOF test is performed for the distribution of heat flow within. The average rejection rate is the fraction of disks within a covering for which the gamma hypothesis is rejected at the $\alpha = 5\%$ level. For sufficiently large samples from a gamma distribution, this rate would converge to $5\%$. The black box plots show, for each indicated $R$, the distribution of these rejection rates over 200 generated coverings. The gray box plots show the same distribution for synthetic gamma-distributed global coverings (details in Appendix B1). Known processes affecting the used part of the NGHF data set ($250\,\mathrm{mW\,m^{-2}}$ threshold, discretization, and typical uncertainty) have been simulated. The box plots show median (colored bar), quartiles (extent of the box), up to 1.5×interquartile range (whiskers), and outliers thereof. The box plot show a separation of the two rejection rate distributions, indicating that there are patterns in the real heat flow data that cannot be explained by a gamma distribution and uncertainty. As $R$ decreases, the discrepancy decreases as well until at $R \lesssim 80\,\mathrm{km}$, the $R$-disks contain too few data points to resolve the average rejection rate properly (illustrated by the dashed line showing the average of 100 divided by the number of disks in a covering).

disks to be rejected by the tests. Higher rejection rates (number of rejected samples / number of samples) indicate that they are
not gamma distributed.

Figure 10 shows the results of the goodness-of-fit analysis. Two $R$-dependent effects can be observed in Fig. 10: at small $R < 80\,\mathrm{km}$, the scatter of small sample sizes becomes dominant. There are just too few regions remaining. For increasing $R$, the rate of rejecting the gamma distribution hypothesis increases slightly.

A striking observation is that for all of the region sizes, the rejection rate is larger than $5\%$ (centered at about $15\%$) and
the fluctuations in rejection rates across different RGRDCs do not alter that conclusion. Regional aggregate heat flow is not generally gamma distributed.

This deviation is not due to known heat flow data uncertainty. To test whether the heat flow data uncertainty might be the cause of the elevated rate of rejections, we have performed a synthetic analysis using synthetic RGRDCs generated by the algorithm in Appendix B1. After generating gamma-distributed random values similar to RGRDC data, relative error following the uncertainty distribution of "A" and "B" data (shown in Fig. 9) is added to the data. The resulting rejection rates show similar spread as the NGHF data, but the bias is small (the median of the rejection rates across synthetic RGRDCs never exceeds ~8 %). Consequently, unbiased random error as specified for the heat flow data within the NGHF is not sufficient to describe the ~15% rejection rate of the gamma model.

The impact that this imperfect model of regional aggregate heat flow has on the accuracy of the results is not immediately clear. On one hand, using a wrong model to analyze the data suggests a detrimental impact on the accuracy. On the other hand, if the model is close enough the method might be accurate up to a desirable precision. Later in section 4.2.2 we investigate, using synthetic data, how well REHEATFUNQ can quantify heat flow anomalies when regional aggregate heat flow data is decidedly non-gamma distributed.

Before these synthetic investigations, the following sections investigate potential causes for the deviation from a gamma distribution in section 4.1.2, and test whether other parsimonious models for the regional aggregate heat flow distribution perform better than the gamma distribution on RGRDCs of the NGHF data set in section 4.1.3.

### 4.1.2 Goodness of fit: the level of misfit from mixture models

Following the observation that the gamma distribution is not a general description of regional aggregate heat flow distributions, we investigate potential causes for this misfit and how large the deviation from a gamma distribution has to be to produce the ~15% rejection rates of the previous section.

We find that the mismatch could be explained by mixture distributions. Figure 11 shows the same RGRDCs Anderson-Darling rejection rate as Fig. 10, and additionally the rejection rate computed for two mixtures of two gamma distributions each. The two mixture distributions are synthetic but cover the range of typical heat flow values and the samples drawn from them have the sample size distribution as the RGRDCs. One distribution ("mix 0") has fewer overlap between the two peaks than the other, and leads to large rejection rates ~80 %. The other, "mix 1", has larger overlap between the peaks and they are more equally weighted. This mixture model matches the observed rejection rates across the NGHF data RGRDCs very closely. Similar mixture models could hence be a possible cause for the observed rejection rates across the NGHF if the heat flow were indeed gamma distributed.

The mixture distribution can arise in the real heat flow data if the disk intersects a boundary between two regions of different heat flow characteristics. Since radiogenic heat production in the relevant upper crust can vary on the kilometer scale (Jaupart and Mareschal, 2005), such an occurrence seems plausible. The occurrence of a boundary intersection mixture might be frequent and with smaller difference between the modes (corresponding to "mix 1"), or it might be infrequent but with a larger inter-mode distance (dashed line in Fig. 11). Both cases are compatible with the statistics observed in the NGHF data RGRDCs.

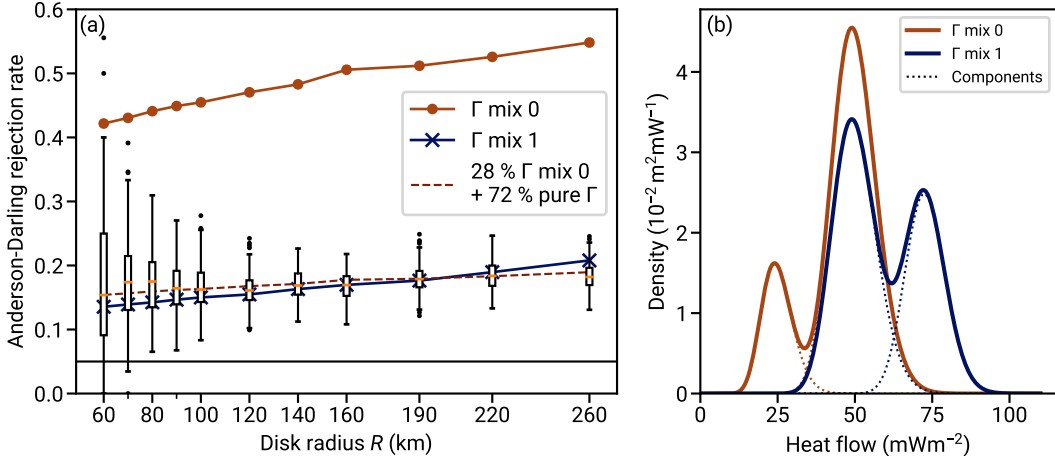

**Figure 11.** Exploring the misfit between the gamma distribution model of regional heat flow and the NGHF data. Panel (a): box plots show the fraction of heat flow distributions from RGRDCs of the NGHF data base for which the gamma distribution hypothesis is rejected at the 5 % level (same data as Fig. 10 (b)). The solid lines with dots show the same fraction of rejections computed for two sets of 10000 samples each, drawn from the two gamma mixture models shown in panel (b) (colors corresponding). Each sample from the mixture distributions has its size drawn from an NGHF RGRDCs of the corresponding $R$, replicating the sample size structure derived from the NGHF. The dashed line in the left plot shows the case if 72 % of the samples were gamma distributed (5 % rejection rate, black horizontal line) while 28 % of the samples were draft from the mixture model 0. Panel (b): the dotted lines indicate the two gamma distributions comprising each mixture. The parameters are $w_0 = 0.2$, $k_0 = 25$, $\theta_0 = 1\,\mathrm{mW\,m^{-2}}$, $k_1 = 50$, and $\theta_1 = 1\,\mathrm{mW\,m^{-2}}$ for $\Gamma_0$ (where $w_0$ is the weight of the zero-index component), and $w_0 = 0.4$, $k_0 = 128$, $\theta_0 = 0.57\,\mathrm{mW\,m^{-2}}$, $k_1 = 50$, $\theta_1 = 1\,\mathrm{mW\,m^{-2}}$ for $\Gamma_1$.

The match with the observed rejection rates is not conclusive evidence that the heat flow data within the regions follow gamma mixture distributions. It is likely that many different distributions could be constructed that lead to similar rejection rates. However the match is a good indication of how large the deviation between the underlying distribution and the simple gamma model is. Somewhere between "gamma mix 0" and "gamma mix 1" lies a critical point in terms of mode-separation beyond which the distribution would be further departed from a gamma distribution than what is observed in the NGHF data RGRDCs.

At this point, we can summarize that heat flow in disks of radius 60 to 260 km is not generally gamma distributed and this is not an artifact of data processing or uncertainty. Mixtures of gamma distributions within the disks, for instance representing variation on smaller scales below the smallest radius we can investigate, could explain the mismatch. Moreover, the mixtures indicate the level of mismatch that would lead to the statistics observed in the real heat flow data.

We proceed in the following section 4.1.3 by investigating whether other two-parameter univariate probability distributions perform better in describing the real-world regional aggregate heat flow distributions. Later in section 4.2.2, we will investigate the impact that the misfit of the gamma model has on the quantification of heat flow anomalies. This will motivate the model despite the mismatch to observed data until a physics-derived alternative becomes available.

### 4.1.3 Comparison with other distributions

We compare the performance of the gamma distribution with a number of common probability distributions. This aims to investigate whether the mismatch can be resolved by choosing another simple probability distribution, or whether the gamma distribution performs good in terms of a simple model.

The comparison is performed at the transition point between insufficient sample size and increasing misfit with radius, 80 km. Even though the global heat flow data base of Lucazeau (2019) is large, the number of samples within an $R = 80$ km-disk is rather small at 20 km minimum distance between the data points (typically 11). Therefore, we focus our analysis on two-parametric models.

We generate samples from the NGHF using the RGRDCs described in Appendix B. To each disk's sample, we fit probability distribution candidates using maximum likelihood estimators and compute the Bayesian information criterion (BIC, Kass and Raftery, 1995). The probability distribution with the smallest BIC is the most favorable to describe the subset and the absolute difference $\Delta$BIC to the BIC of the other distributions indicates how significant the improvement is. In our particular case, $\Delta$BIC depends only on differences in the likelihood since all investigated distributions are two-parametric. We repeat the process 1000 times to prevent a specific random regional heat flow sample selection skewing the results (see Figs. S9–S11 in the supplement for a convergence analysis).

Due to the right-skewed shape of the global distribution, see Fig. 1 (b), we test a range of right-skewed distributions on the positive real numbers: the Fréchet, gamma, inverse gamma, log-logistic, Nakagami's $m$-, shifted Gompertz, log-normal, and two-parameter Weibull distribution (Bemmaor, 1994; Leemis and McQueston, 2008; Kroese et al., 2011; Nakagami, 1960). The global distribution does not have to be representative of the regional distributions, however. Since the global distribution is a mixture of the regional distributions, only the weighted sum of the regional distributions needs to have the right-skewed shape. Therefore, we additionally test the normal distribution (e.g. used by Lucazeau, 2019).

In Fig. 12 (c), the results of the analysis are visualized using the rate of BIC selection, that is, the fraction of regional heat flow samples for which the hypothesized distribution has the lowest BIC. Furthermore in panel (a), the distribution of $\Delta$BIC to the second lowest scoring distribution is shown for the samples in which each distribution is selected, and in (b) the distribution of (negative) $\Delta$BIC to the selected distribution is shown for the samples in which each distribution is not selected. The Weibull distribution has the highest selection rate followed by the Fréchet distribution. Combined, they accumulate roughly 60 % of all selections. Together with the normal distribution and one outlier of the log-logistic distribution, they are the only distributions with occurrences of $\Delta$BIC > 2, which might be considered "positive evidence" (Kass and Raftery, 1995, p. 777). However, these $\Delta$BIC > 2 occur only in less than 4.4 % of the total subsets for each of the three distributions. Therefore, no distribution is unanimously best at describing the regional heat flow.

The $\Delta$BIC for regions in which a distribution is not selected leads to a different selection criterion: if a distribution is not the best-scoring distribution, how much worse than the best is it? These differences are generally more pronounced than the differences of the best to the second best fitting model. Especially the Fréchet and inverse gamma distribution perform strongly ($\Delta$BIC > 6, Kass and Raftery, 1995, p. 777) worse than the better fitting distributions in more than 50 % of the cases in which

they are not selected. The generally least badly performing models are the gamma, log-logistic, normal, Nakagami, and Weibull distribution. Their negative $\Delta$BIC distributions have only minor differences and one or the other performs better depending on the quantile of the negative $\Delta$BIC investigated.

To conclude, the gamma distribution is among the best-performing distributions in terms of a consistently good description of the data. There are no significant differences between the distributions in terms of fitting the data that would favor any of the other distributions over the gamma distribution. Up until the typical shape of the regional aggregate heat flow distribution is derived from physical principles, the choice among the set of best-performing distributions remains a modeling decision. Here, the gamma distribution is the only distribution of the best-performing set that fulfills all three of the following criteria: (1) it is defined on a positive support, (2) it has a conjugate prior for enabling costly computations, and (3) it is right-skewed, like the global heat flow distribution, for all parameter combinations. We hence choose the gamma distribution within REHEATFUNQ.

## 4.2 Validation using synthetic data

In this section we analyze the performance of the complete anomaly testing model described in section 3.4 using synthetic data, that is, computer-generated samples $\{(x, y, q)_i\}$ of surface heat flow. The purpose of this test is to investigate the impact of the conjugate prior, the model's+ correct identification of synthetic anomalies, and the impact that the deviation of real-world data from the assumed gamma distribution could have on the model's performance.

In the following sections, a number $N$ of artificial heat flow values within an 80 km radius disk are generated following a variety of distributions. To investigate the impact of the sample size and potential convergence, $N$ is varied in the synthetic experiments. We note that in these tests, the minimum distance of 20 km between data points is not enforced. As a consequence, we can also investigate data set sizes larger than 64 samples, which might be close to the densest point packing within a circle with a minimum-distance to radius ratio of 0.25 (Graham et al., 1998), that is, the maximum our minimum distance criterion allows.

The minimum data set size we investigate is 10, corresponding to a density of $5.0 \times 10^{-4}$ km$^{-2}$. The densest packing of 64 data points corresponds to a density of $3.2 \times 10^{-3}$ km$^{-2}$ and the maximum sample size we investigate, 100, to a density of $5 \times 10^{-3}$ km$^{-2}$ inside an 80 km disk. When enforcing the 20 km minimum distance, 100 samples would correspond to a densest-packed circle of radius $\simeq 100$ km (López and Beasley, 2011, Tab. 4, result minus 1) and again a density of $3.2 \times 10^{-3}$ km$^{-2}$.

### 4.2.1 Impact of the conjugate prior

In this first of two tests, we investigate how well the method is able to identify synthetic anomalies that have been superimposed on artificial gamma-distributed heat flow values. Specifically, this aims to investigate the impact of the prior for small data sets. To this end, we compare our choice of prior parameters, Eq. (17), with an (improper) uninformed prior $p = 1$, $s = n = \nu = 0$, whose posterior parameters are determined solely by the data.

In Fig. 14, the performance of the model is evaluated on synthetic data generated from gamma distributions and superposed with a synthetic anomaly of a vertical fault following Lachenbruch and Sass (1980). The anomaly, Eq. (25), is computed for a

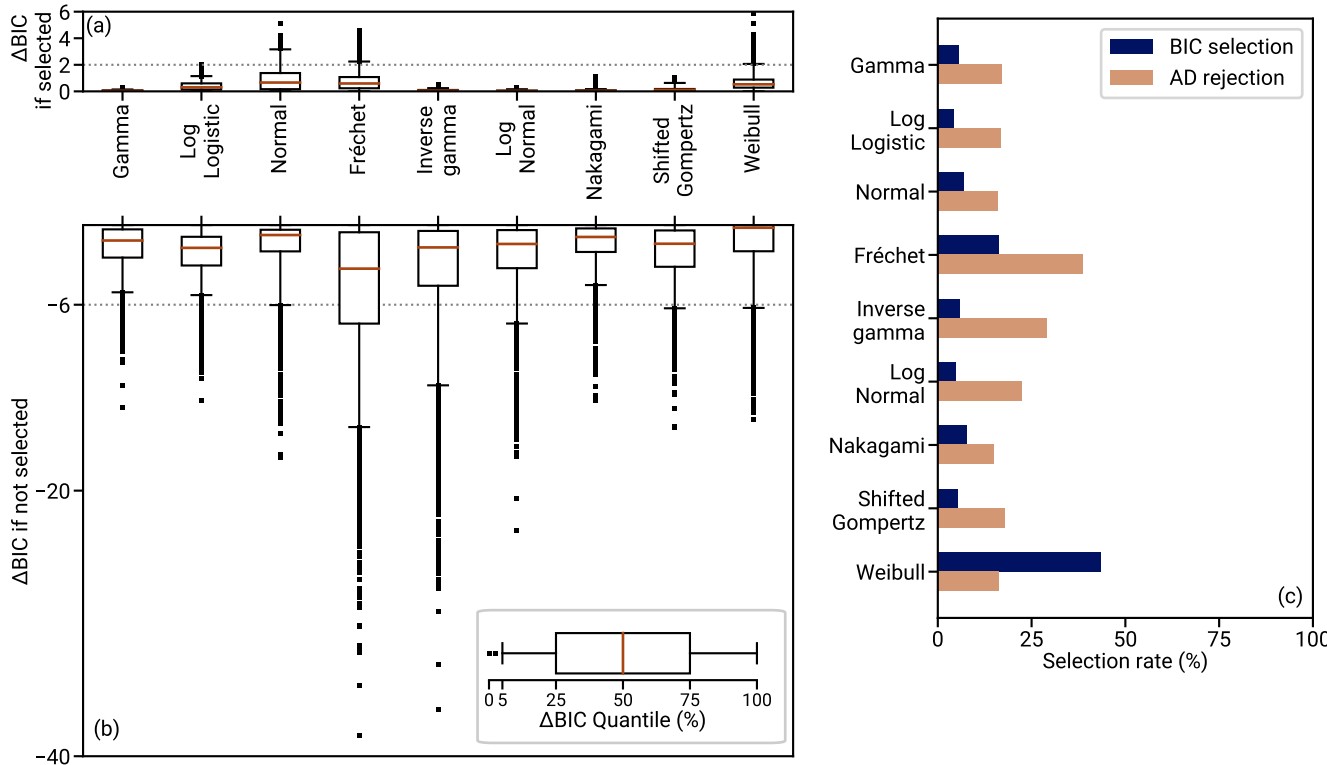

**Figure 12.** Selection rates, and their significance, of different two-parameter probability distributions for modeling regional heat flow distributions for global coverings of 80 km radius circles. Panel (c): given a RGRDC (described in Appendix B), the selection rate denotes the fraction of these regions for which the indicated distribution has the lowest BIC. Panel (a): the box plots show the distribution of $\Delta$BIC conditional to the respective distribution being selected (that is, each time it has the lowest BIC). $\Delta$BIC is then computed as the distance of this lowest BIC value to the second-lowest BIC value. In other words, it quantifies how much better than the best competition the distribution performs if selected. Panel (b) shows in similar box plots the distribution of $\Delta$BIC to the lowest scoring distribution among the regional heat flow samples in which the indicated distribution is not selected. In other words, the bottom panel shows how much worse than the best fitting distribution each distribution is if it is not the best (values closer to zero are better). The data of all three panels are aggregated from 100 random global coverings.

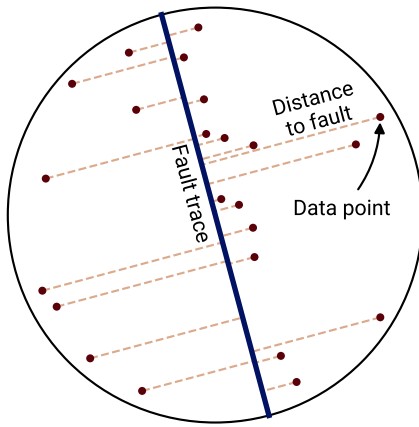

**Figure 13.** Configuration of the synthetic heat flow data and fault in the synthetic method tests, section 4.2 and subsections. The configuration is chosen to reflect both the *random global R-disk coverings* used in section 4.1.1 and the heat flow anomaly on the San Andreas fault used in section 5. The regions are disks of radius 80 km in which the locations of synthetic heat flow measurements (dots) are distributed according to a uniform probability density. A straight fault (thick line) intersects the disk through its center. The dashed lines show the distances $x$ of the heat flow data from the fault trace, which is used to determine the anomaly strength $c_i$ as a function of frictional power $P_H$ dissipated on the fault segment. The anomaly shape is according to Eq. (25), following Lachenbruch and Sass (1980), which is displayed in Fig. 8.

power of 10 MW and the spatial configuration shown in Fig. 13. Both using the prior with parameters Eq. (17) and a flat prior, the marginal posterior imposes correct bounds on the anomaly strength with increasing precision as the sample size increases.

The difference between the flat and the informed prior is most pronounced at small sample sizes and standard deviations. There, the flat prior places a considerably tighter constraint on the anomaly strength. At larger standard deviations, a region emerges in which this relation is reversed and the informed prior leads to stricter constraints albeit considerably less so (less
than 10 % improvement on the 1 % tail quantile bound). This region coincides in part with the isolines of the prior density (see Fig. 6). Since the region lies in a part of the $(\alpha, \beta)$ space with larger standard deviation, improvements in this area can be helpful.

All in all, the impact of the informed prior is ambiguous. In a part of the $(\alpha, \beta)$ space that is densely covered by modes of the RGRDC likelihoods and which is hence likely to cover the regional aggregate heat flow distribution of a heat flow analysis, the
prior influences the analysis positively. Yet, the improvement in these regions is small compared to the large overestimation of the anomaly—see panel (c). In regions of the parameter space in which the heat flow distributions have less scatter (large $\alpha$ and $\beta$), on the other hand, the upper bounds on the anomaly magnitude are significantly increased. This leads to a very conservative estimate of the anomaly bound and the uncertainty.

As a rule of thumb, the prior with optimized parameters, Eq. (17), is rather beneficial at small sample sizes (around ten
data points or less) for "typical" gamma distribution parameters, that is, for those parameters whose distributions resemble

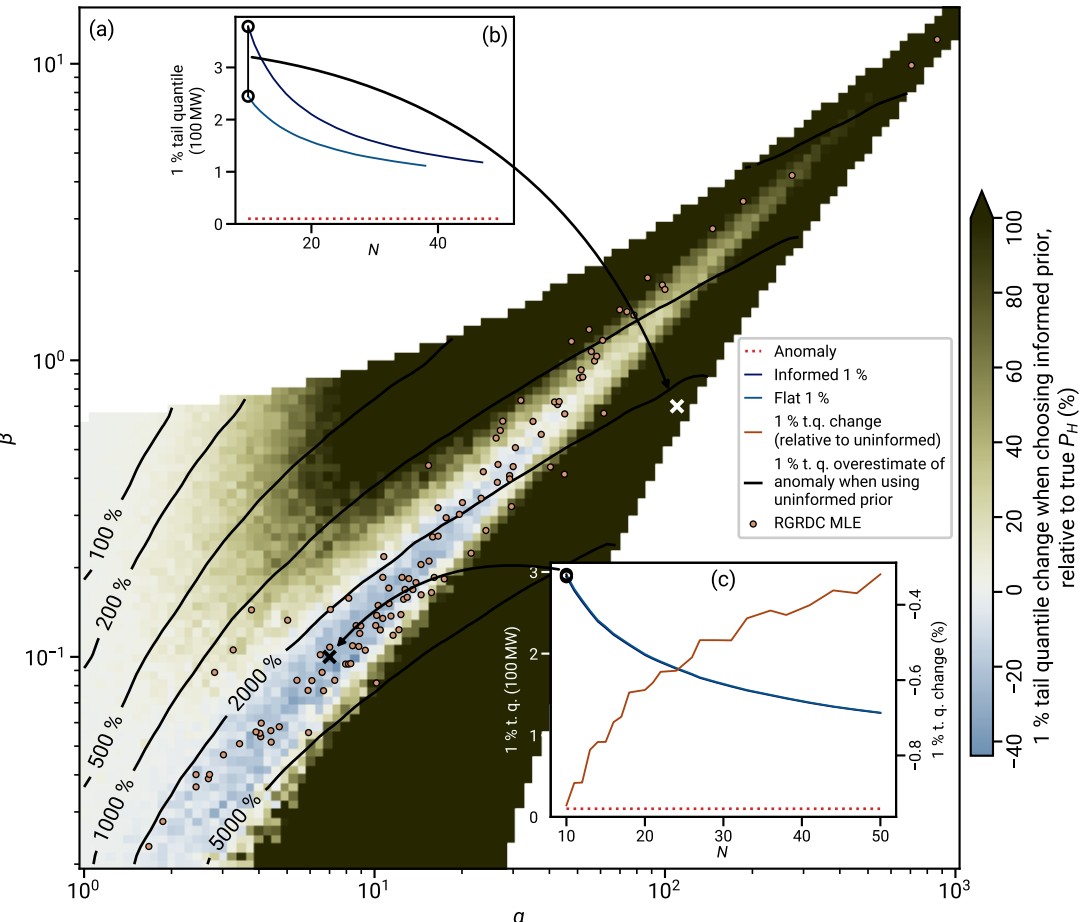

**Figure 14.** Performance of the anomaly testing with the posterior Eq. (23) for synthetic data. Heat flow data ($N = 10$) has been generated from gamma distributions with $\alpha$ and $\beta$ according to the position within the 2d parameter space and each data point has been assigned a lateral distance between -80 and 80 km from a vertical fault segment of 14 km depth and 160 km length. Afterwards, an anomaly with 10 MW according to the model Eq. (25) has been added to the data and quantiles of the posterior tail distribution (23) have been computed. This background color has been computed from 1000 such synthetic data sets. It shows the median of the relative difference between the 1 % tail quantiles computed from the informed (17) and a flat ($p = 1$, $s = n = \nu = 0$) prior. A value below zero (light blue color) indicates that the informed prior results in a lower 1 % tail quantile, that is, provides stronger constraints. Such a point in the $(\alpha, \beta)$ parameter space is shown in inlay (c) corresponding to the black cross, where we have performed the analysis for 100000 synthetic data sets and different $N$. A positive background color value in $(\alpha, \beta)$ space indicates that the flat prior imposes a tighter constraint on the anomaly. Such a data point is shown in inlay (b) corresponding to the white cross. In all points of the parameter space shown here, the median 1 % percentile of both priors is larger than the actual anomaly, i.e. no underprediction of the anomaly strength occurs. Top left and bottom right white parts of the parameter space are unlikely volumes of the prior (17) and have not been investigated.

real-world heat flow data sets. This prior is accessible as the default prior of the REHEATFUNQ model. If the variance of the data is particularly low (for instance after removing known spatial signals from the heat flow data), it might be preferable to use the uninformed prior. If sample sizes are large, say 50–100 data points or more, the results might not differ much due to the prior being forgotten.

We close this section with a remark on a potential lead for improved prior parameter estimates. A potential cause for the worse performance, compared to the flat prior, in parts of the $(\alpha, \beta)$ space could be the prior's favoring of small $\beta$ and $\alpha$ which correspond to larger variance. The prior would result in bias to larger variance. This prior shape is, in turn, likely a consequence of the higher concentration of likelihood modes of the random regional samples at low $\alpha$ and $\beta$ (see the location of MLE $(\alpha_i, \beta_i)$ in Fig. 14).

### 4.2.2    Impact of non-gamma heat flow distributions

Section 4.1.1 shows that the gamma distribution does not fully describe the RGRDCs. Thereafter, section 4.1.2 illustrates that an overlapping mixture of two gamma distributions would produce a similar mismatch and is hence one plausible explanation. Moreover, we were able to identify a limit of the separation of the two mixture components (Fig. 11) beyond which the mixture has departed further from the unimodal gamma distribution than the observed distributions in the RGRDCs. The question

remains how this moderate departure of $p(q_u)$ from our gamma model will affect the identification of the fault-generated heat flow anomaly power $P_H$ in Eq. (23).

To answer this question, we perform Monte-Carlo simulations. Regional aggregate heat flow distributions are drawn from mixture distributions that mimic general patterns of the RGRDC regional distributions from section 4.1.1 for which the gamma hypothesis is rejected. These general patterns are bimodal histograms and those which have a widespread base level and a sharp

peak.

We choose to model both patterns by a two component Gaussian mixture, the first by two well-separated normal distributions (NDs) of similar standard deviation, and the second by one ND with large and one with small standard deviation. Considering the small sample sizes (~10 data points) this is likely overfitting, but it suits the analysis of extreme deviations from the gamma distribution. Conceptually, these mixtures could represent a sharp separation of two heat flow regimes within an 80 km circle.

To ensure positivity, the mixture PDF is cut at zero using sample rejection.

In Fig. 15, we show three mixture distributions that have been inspired by Komogorov-Smirnov-rejected samples from the RGRDCs of section 4.1.1 (see the Figs. S2 to S4 in the electronic supplement for the histograms). The distributions have been selected to sample both previously described types and to sample a large range of heat flow levels from low (10–50 mW m$^{-2}$) to high (50–125 mW m$^{-2}$) values. For these three distributions the posterior quantiles of Eq. (23) are evaluated for different

samples sizes using 1000 Monte-Carlo runs.

To ensure that the method works for all fault powers, we have performed the analysis for fault powers 10 and 271 MW (62 kW km$^{-1}$ and 1.7 MW km$^{-1}$ power per fault length, respectively). The maximum power corresponds to a 15 km deep strike-slip fault segment (depth-averaged resisting stress $\bar{R} \approx 445$ bar according to Lachenbruch and Sass (1980) for Byerlee friction)

of 160 km length at a slip rate of 8 cm per year. This slip rate is an upper limit and corresponds to the fastest known continental shear zone, the Bird's Head region of eastern Indonesia (Stevens et al., 2002), if it were released on a single fault.

Figure 15 shows the median of the 1 %, 10 %, 50 %, and 90 % tail quantiles across 1000 samples from each distribution for a selection of sample sizes $N$ between 10 and 100. With increasing $N$, the posterior becomes increasingly concentrated. In case of the two bimodal distributions, the true anomaly strength $P_H$ is captured within the 10 % and 90 % tail quantile for $N > 50$. For small $N$, the 10 % tail quantile is an upper limit on the anomaly strength but the 90 % tail quantile (or equivalently 10 % quantile) is biased to larger values for $P_H = 10$ MW. In case of the unimodal distribution with longer tails, the true anomaly is larger than the range which 10 % to 90 % quantiles converge to at $N = 100$, that is, the power of the anomaly is underestimated. In all cases, the median 1 % tail quantile is an upper bound across the investigated $N \leq 100$.

If the anomaly is strong ($1.7$ MW km$^{-1}$), its strength can generally be well quantified. The median typically follows the true anomaly strength closer than the width of the 80 % symmetric quantiles. Even in the exceptional case of distribution D2 for $N \to 100$, the 80 % symmetric quantile is close to the true frictional power.

If the anomaly is weak ($62$ kW km$^{-1}$), the posterior overestimates $P_H$. Especially for $10 \leq N \leq 20$, the true anomaly $P_H$ is located at or below the 90 % tail quantile. In case of D3, the median overestimates $P_H$ by a factor of 10 at $N = 10$.

So far, the analysis has considered the median across the synthetic simulation (1000 samples) for each $N$. This ensemble-view is useful to investigate the bias, but it does not fully reflect the inference problem on a particular fault. When considering the median of a tail quantile $P_H(t)$ across the simulated set of samples, $P_H(t)$ will be smaller than this median in 50 % of the cases. For instance, if the 10 % tail quantile $P_H(10\%)$ straddles the strength of the anomaly at $P_H = 271$ MW for distribution D2, its use would underestimate the anomaly with 50 % chance if D2 were the true regional heat flow distribution.

This warrants investigating the relationship between a chosen tail quantile $t$ and the resulting rate $r$ of exceeding the corresponding power $P_H(t)$. The panels (d), (h), and (l) of Fig. 15 show the relation between $t$ and $r$ for the three distributions D1, D2, and D3, and for the two powers 10 MW and 271 MW.

Depending on the actual power of the anomaly, two different behaviors can be observed. If the anomaly is low, the rate of exceeding the tail quantile is lower than chosen, $r < t$. Especially for D1 and D3, this effect is pronounced when $t \lesssim 50\%$, where $r = 0$. This means that within the 1000 synthetic samples, none exceeded the tail quantile. A similar albeit not as pronounced effect can be seen for D2. Hence, the small tail quantiles are hence a cautious estimate for low $P_H$.

For large $P_H$, the opposite occurs. At small tail quantiles, $r > t$ and the size of the anomaly is underestimated. Depending on the distribution, the maximum excess of $r$ at the 10 % tail quantile ($t = 10\%$) is 4 % to 18 %. That is, there are 40–180 % more samples than desired in which the anomaly is underestimated.

The severity of this underestimation can be expressed by what we call the "bias" $B$. We define $B$ to be the relative under- or overestimate of the true power $P_H$ at the actual tail quantile $t$. Formally, if we express with $\hat{P}_h(t,i)$ the power of the $i$th sample at the posterior tail quantile $t$, and with $\hat{P}_h(t)$ the $t$th (smallest) quantile of $\hat{P}_h(t,i)$ among the generated samples, then we define $B$ by

$$B = \frac{\hat{P}_h(t)}{P_H} - 1. \tag{29}$$

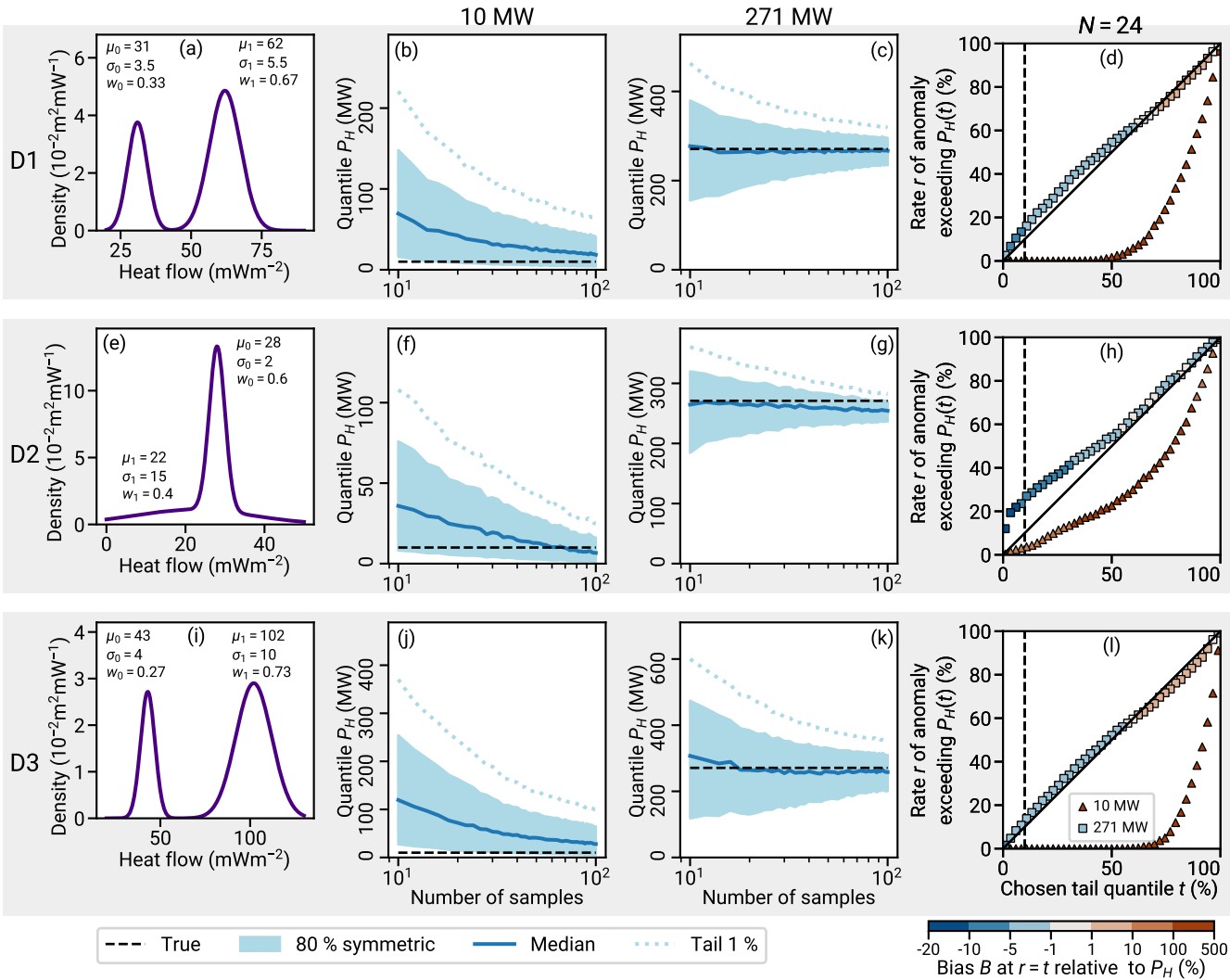

**Figure 15.** Resilience of the posterior Eq. (23) to violations of the gamma model hypothesis. We investigate how well REHEATFUNQ can quantify heat flow anomalies if the undisturbed regional aggregate heat flow follows a probability distributions that deviates from the gamma model. Panels (a), (e), (i): each graph shows the Gaussian mixture models D1 to D3 of that row ($\mu$ denotes mean, $\sigma$ standard deviation, and $w$ the weight of the mixture component). Panels (b), (f), (j): each graph shows quantiles of the posterior given the synthetic regional PDF of the same row and a synthetic anomaly as in Fig. 13 with 10 MW power over a fault length of 160 km (14 km depth). The anomaly profile follows Eq. (25). The quantiles are the median over a set of 1000 samples for each $N$. Panels (c), (g), (k): the same for a 271 MW anomaly. Panels (d), (h), (l): for $N = 24$, the chosen tail quantile $t$ is plotted against the rate $r$ at which $P_H(t)$ exceeds the true power $P_H$ of the anomaly within the 1000 samples. The $r = t$ correspondence is indicated as a solid line, the 10 % tail quantile as a dashed line. The marker fill color, corresponding to the color bar, indicates the difference of the $t$-quantile (across the 1000 samples) of $P_H(t)$ relative to the true power. That is, the bias at the actual exceedance rate $t$ when using $P_H(t)$. Negative shows that the anomaly is underestimated, positive that it is overestimated.

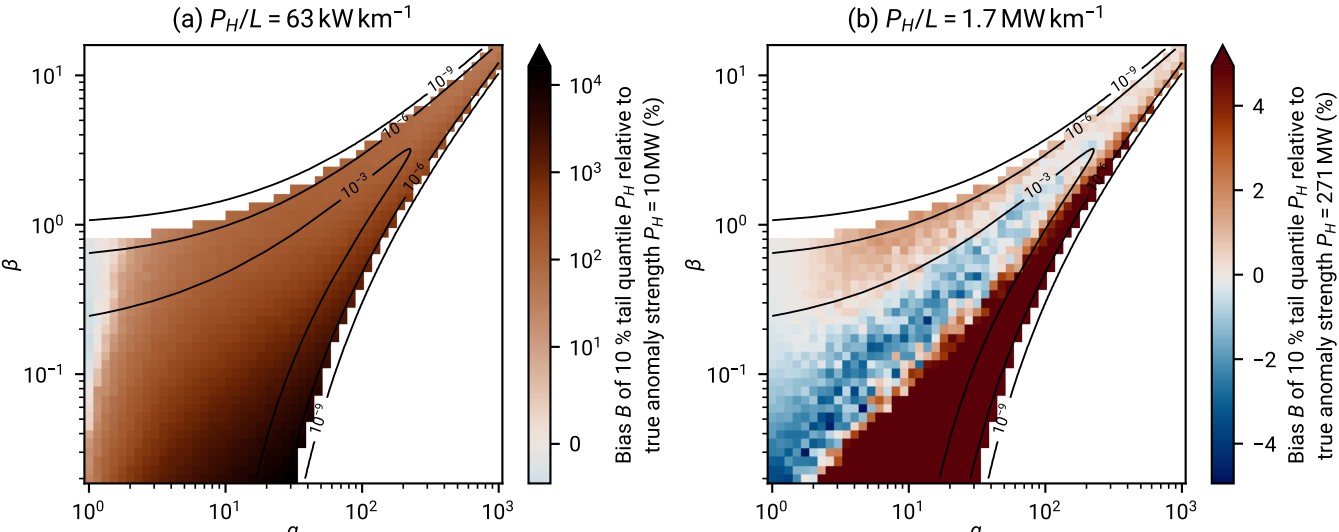

**Figure 16.** Bias $B$ of the $10\%$ tail quantile for different gamma-distributed regional aggregate heat flow distributions. The bias (Eq. 29) quantifies how much the location $\hat{P}_h(t)$ of the tail quantiles of the posterior $f(P_H)$, Eq. (23), interpreted as a frequentist exceedance interval, under- or overestimates the true anomaly power $P_H$ at the ensemble's $10\%$ quantile of $\hat{P}_h(t)$. A negative value indicates that the anomaly is underestimated by at least that amount in $10\%$ of 1000 generated samples. A positive value indicates that the anomaly is overestimated at the $10\%$ quantile of $\hat{P}_h(t)$, and that the smallest estimate of $P_H$ in that quantile is larger than $P_H$ by the given amount $B$. A value of $B=0$ indicates that the tail quantile coincides with the rate-$t$ exceedance interval of the ensemble.

The purpose of $B$ is to indicate the bias of $\hat{P}_h$ as an estimator at the true rate of exceedance $t$. If $B$ is negative, at a chance $t$ the anomaly power $P_H$ is underestimated by a fraction $B$ or more. If $B$ is positive, the rate at which $P_H$ exceeds $\hat{P}_h$ is actually smaller than $t$ and there is an overestimation of the power $P_H$ by a factor of at least $B$ among the $1-t$ largest $\hat{P}_h(t,i)$ (it would be zero for the "unbiased" $B=0$).

For small frictional power, $B$ is positive through nearly all of the range of $t$. At $t \approx 10\%$, for instance, we find $B = 40$–$610\%$ for $P_H = 10$ MW. At large $P_H$, $B$ is negative and ranges from -4 to -9 %. Conclusively, there is a substantial margin for the quantification of small anomalies and a small underestimation for large anomalies.

The two effects also occur for a purely gamma distributed heat flow sample, although they are less pronounced in underestimating large $P_H$ (in SI1.5, we find a maximum of 5 % underestimation at the 10 % tail quantile within the gamma distribution parameter space shown in Fig. 6).

To conclude, the tests yield posterior support for our gamma model choice. The results derived from the three normal mixture distributions indicate that the frictional power generating a heat flow anomaly can be constrained even if the regional aggregate heat flow is not gamma distributed. Small tail quantiles (e.g. 1 % or 10 % are typically larger than actual power $P_H$. If $P_H$ is small, the small tail quantiles have a large margin to the actual power. This behavior occurs also for gamma distributed aggregate heat flow, so that it is likely an effect of the large ratio of heat flow standard deviation to anomaly amplitude. If the

frictional power is small, the small tail quantiles might underestimate $P_H$ but the amount of this underestimation is relatively small.

Whether the posterior can be used to quantify the heat flow anomaly further than giving an upper bound on its generating power depends on its amplitude and the sample size. If the frictional power is large compared to the fluctuation of the aggregate heat flow distribution—where in our setting an example of "large" is the 271 MW anomaly—the REHEATFUNQ model can quantify the heat flow anomaly throughout the sample size range 10–100. The posterior's median is meaningful, and the posterior's 80 % centered quantile clearly separates from zero. If the frictional power is small compared to the fluctuations—
here 10 MW—the posterior's central quantiles lose their significance as best estimates and overestimate instead. This indicates that the bulk of the posterior can be used as a confident estimate of the frictional power only if the heat flow anomaly is large compared to the heat flow variability, or if the sample size is exceptionally large. Biases due to a differing distribution seem to have a relevant impact on this conclusion only at unrealistically large sample sizes (~100 and beyond).

## 4.3   Physical limitations

Besides the statistical limitations discussed above, we discuss two physical limitations of the REHEATFUNQ model. One concerns both the regional aggregate heat flow distribution and the anomaly quantification, while the second is relevant only to the quantification of heat flow anomalies.

### 4.3.1   Regional aggregate heat flow distributions

A first limitation that affects both the inference of regional aggregate heat flow distributions and the quantification of heat flow
anomalies is if the fluctuations are mostly due to random sampling of a spatially varying heat flow field (Fig. 2). The limitation is that the precision of the results cannot be arbitrarily increased by increasing the number of measurements (or at least this would not be obvious). As more and more measurements are taken, the full spatial variability will eventually be explored. In combination with the researcher's spatial measurement strategy, which will likely be systematic rather than random, all sources of "randomness" in the stochastic model of regional heat flow variability would eventually be exhausted. This limit
is particularly important for the anomaly quantifaction, where at some point—read number of measurements—the ability to quantify an anomaly will be limited by the ability to disentangle a fixed spatial signal and not by a limited number of data.

The solution to this limitations are straightforward: first, given that the fluctuations were mainly due to a randomly sampled spatial field and a large number of heat flow measurements were available, this field should be clearly distinguishable in the point cloud of heat flow measurements (imagine adding ten times more data points to Fig. 2). Hence the effect would not be
a hidden effect. To account for the spatially varying field, geophysical modeling of the heat generation and transport can then help to detrend the data. This latter part can of course also help reduce the uncertainty if there are only few data points (see e.g. Fulton et al., 2004).

### 4.3.2 Anomaly quantification

In section 3.1, we have listed a number of possible origins for the regional fluctuation of surface heat flow. While the distinction between these origins is not relevant for the derivation of a regional aggregate heat flow distribution, it is important for the quantification of heat flow anomalies. In particular, it is important whether the heat flow fluctuations are due to inhomogeneous heat sources or transport that differs from homogeneous conduction. In nature, both effects can be relevant (e.g. Norden et al., 2020).

If the origin of the heat flow fluctuations is the heat source distribution, heat flow anomalies are truly independent from the stochastic process. This is the setting in which the analytical solutions to the heat flow problem based on simplified assumptions (e.g. Brune et al., 1969; Lachenbruch and Sass, 1980), including Eq. (25), can be applied.

If the origin of the heat flow fluctuations is, even partly, due to the heat transport, the separation of the heat flow anomaly from the regional fluctuations is not possible without taking this transport into account. Figure 17 illustrates this with the synthetic example of heat conduction from a buried strike-slip fault. The same spatially fluctuating heat flow field, panel (c), is generated by varying conduction in panel (a) and by varying thermal power density in (b). While the undisturbed heat flow is the same in both configurations are equal, the surface heat flow anomaly that is generated by the buried strike-slip fault differs considerably in panel (d). While in case of homogeneous conductivity the resulting anomaly resembles the analytical solution by Lachenbruch and Sass (1980, Eq. (A22b)), the anomaly is asymmetrically distorted by the inhomogeneous heat conduction. To infer the fault-generating power in case of panel (a), the heat conductivity field $\kappa(\boldsymbol{x})$ needs to be known.

This is an application paradox: to separate a heat flow anomaly from fluctuations that are caused by the transport process, one would need detailed knowledge about this process—which in turn renders modeling the fluctuations by stochastic means unnecessary. The answer to this paradox is that the application of REHEATFUNQ requires that the heat transport equation is sufficiently known in the region of interest, and thereafter allows the separation of heat flow anomalies from the surface heat flow due to an unknown heat source distribution.

## 5   Example: San Andreas Fault

To demonstrate the REHEATFUNQ model, we apply it to the San Andreas fault (SAF) system in California. The SAF has a history of research regarding fault-generated heat flow in context of the discussion whether the fault is frictionally weak or strong (Brune et al., 1969; Lachenbruch and Sass, 1980; Scholz, 2006). The argument brought forward by Brune et al. (1969) and refined by Lachenbruch and Sass (1980) starts with a comparison of the fault-lateral plot of a fault-generated surface heat flow anomaly with heat flow measurements close to the fault (up to ~20–100 km distance). The lack of such an anomaly visible in the data is then used to derive an upper bound on the strength of the heat flow anomaly, that is, the frictional resistance at a given long-term fault slip rate. Visual inspection of the anomaly heat flow graphs are used to define a fuzzy upper bound on the frictional resistance above which the generated anomaly seems unlikely drawn against the data.

This discussion sparked the development of the REHEATFUNQ model, hence our work aims to contribute to the analysis. REHEATFUNQ aims to explore the fluctuation using a stochastic ansatz. The existence of a stochastic process might have a

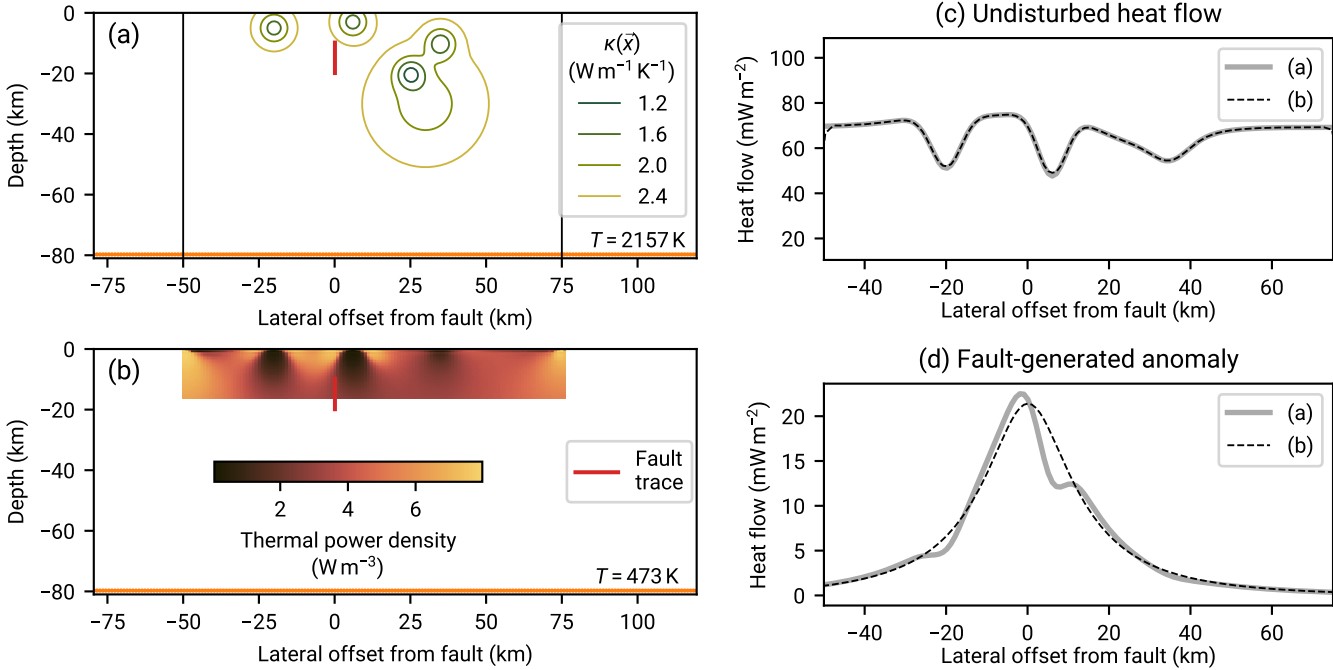

**Figure 17.** Two groups of heat flow fluctuation causes. Panel (a) and (b) illustrate heat flow fluctuations on the basis of inhomogeneous conductivity $\kappa$ and by varying volumetric heat production. Both (a) and (b) are driven by a line heat source of constant temperature at 80 km depth (leading to an average upward heat flow of 68.3 and 40 mW m$^{-2}$ at $\kappa = 2.5$ W m$^{-1}$ K$^{-1}$, respectively) and have a $T = 0$ boundary condition at the surface. The additional heat sources in (b) are fit to match the surface heat flow in (a) (panel (c) shows the surface heat flow of both models). While both the varying conductivity and production can lead to similar fluctuations in surface heat flow, only the heterogeneous conductivity $\kappa$, panel (a), influences the anomaly, panel (d), generated by a buried strike slip fault (frictional power of 0.98 MW km$^{-1}$). If fluctuations are due to inhomogeneous sources, the heat flow anomaly is independent of the fluctuations. The thermal conductivity in panel (a) is varied within bounds similar to what Harlé et al. (2019) found for the Upper Rhine Graben (1–4 W m$^{-1}$ K$^{-1}$), and the heat production in panel (b) is varied compatible to what Jaupart and Mareschal (2005) list for the Australian cratons (0–8 $\mu$W m$^{-3}$).

considerable impact on the analysis since the number of heat flow data in each of the regions investigated by Lachenbruch and Sass (1980) is rather small (6–19) and the fluctuations are in the order of the heat flow anomaly magnitudes. Furthermore, REHEATFUNQ aims to quantify the fuzziness of the assessment of which heat flow anomaly strengths are compatible with the data by means of the posterior $f(P_H)$, Eq. (23).

## 5.1 Regional aggregate heat flow

Figure 18 maps the regions investigated in this section, which we denote by Mojave, Carrizo, Creeping, and North Coast. The four regions can be understood as four distinct sections of the SAF. There are 49 (Mojave), 51 (Carrizo), eight (Creeping), and 36 (North Coast) data points within the four sections. Figure 19 shows close-ups of the four regions. Therein, heat flow data

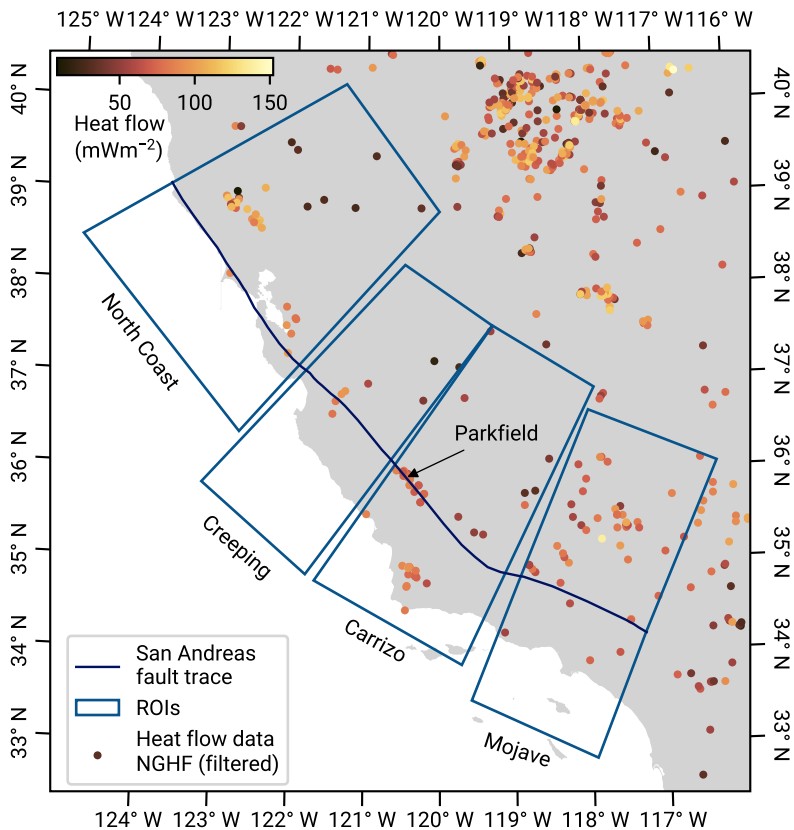

**Figure 18.** Map of the regions investigated in section 5. Heat flow data are from Lucazeau (2019), filtered as described in section 2.1. The San Andreas fault trace is from Milner (2014).

clusters can be seen, and a number of data point pairs that violate the minimum distance criterion can be identified. The range of heat flow varies across the four regions.

Figure 20 shows the posterior predictive distributions of regional aggregate heat flow for the four regions. The Mojave section, panels (a) and (b), is an example of a region that has a rather uniform data distribution and sufficient data to cover the regional aggregate distribution. The empirical CDF (eCDF) for all aggregate data within the region runs close to the center of the set of its random subsets that follow from enforcing the $d_{min} = 20$ km minimum distance criterion to prevent the pairwise clustering visible in Fig. 19 (a). Even though this reduces the density of points, the regional aggregate distribution is not significantly impact by the thinning. This shows in the posterior predictive CDF in Fig. 20 (a) and PDF in panel (b). In both cases, the posterior predictive follows closely the initial regional aggregate distribution.

In the Carrizo section, the effect of spatial clustering and conversely of the exclusive treatment of the clusters' members stands out. The region contains a slightly larger number of data points (51) but the $d_{min}$-enforced eCDFs in panel (c) are coarser-stepped than in case of the Mojave section, panel (a). The cause becomes evident in Fig. 19, panel (b), which shows

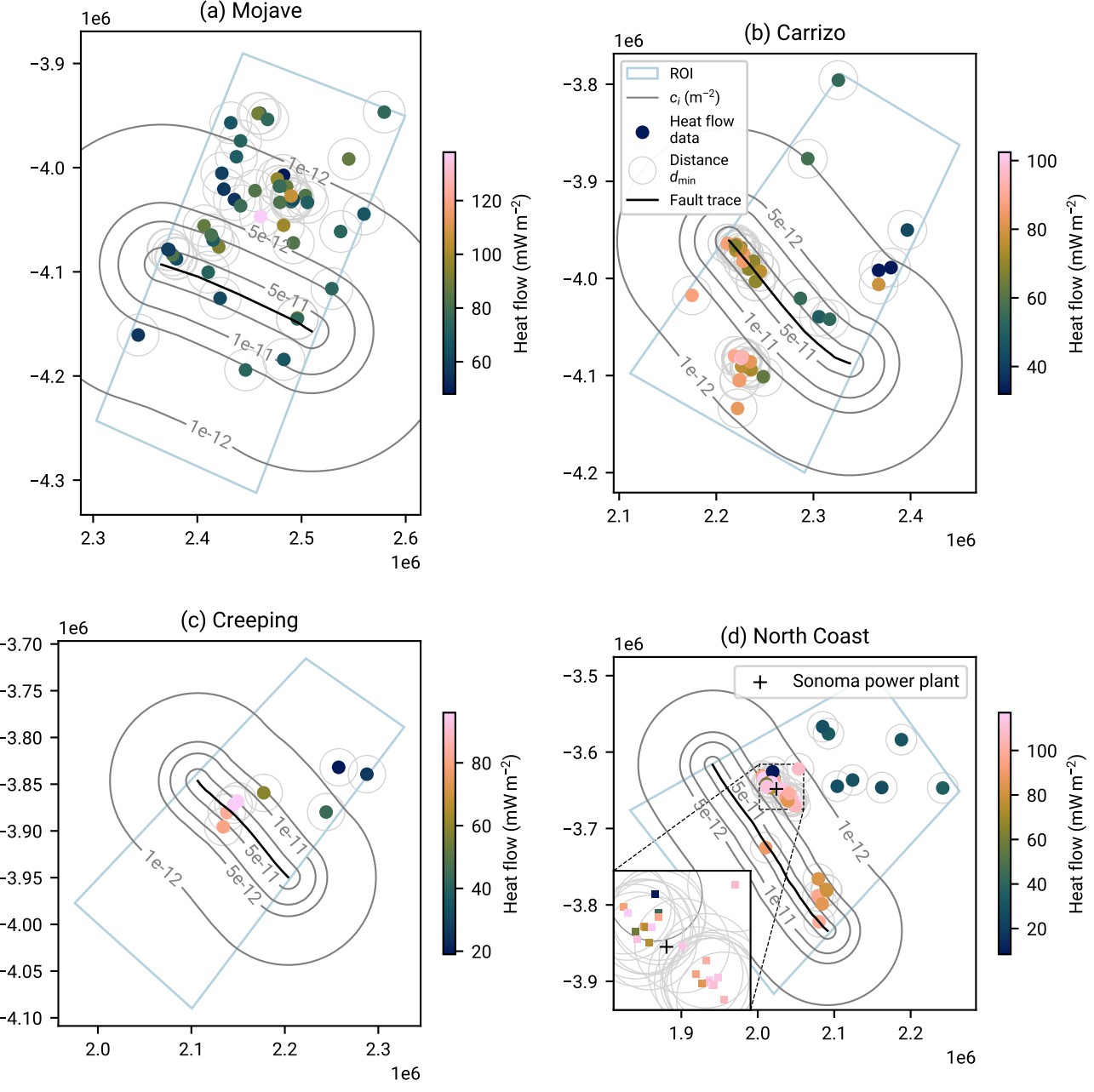

**Figure 19.** Close ups of the four regions of Fig. 18 and heat flow data within. The gray contour lines show the spatial heat flow field obtained by parameterizing the heat flow anomaly Eq. (25) by the distance to the closest point on the fault trace.

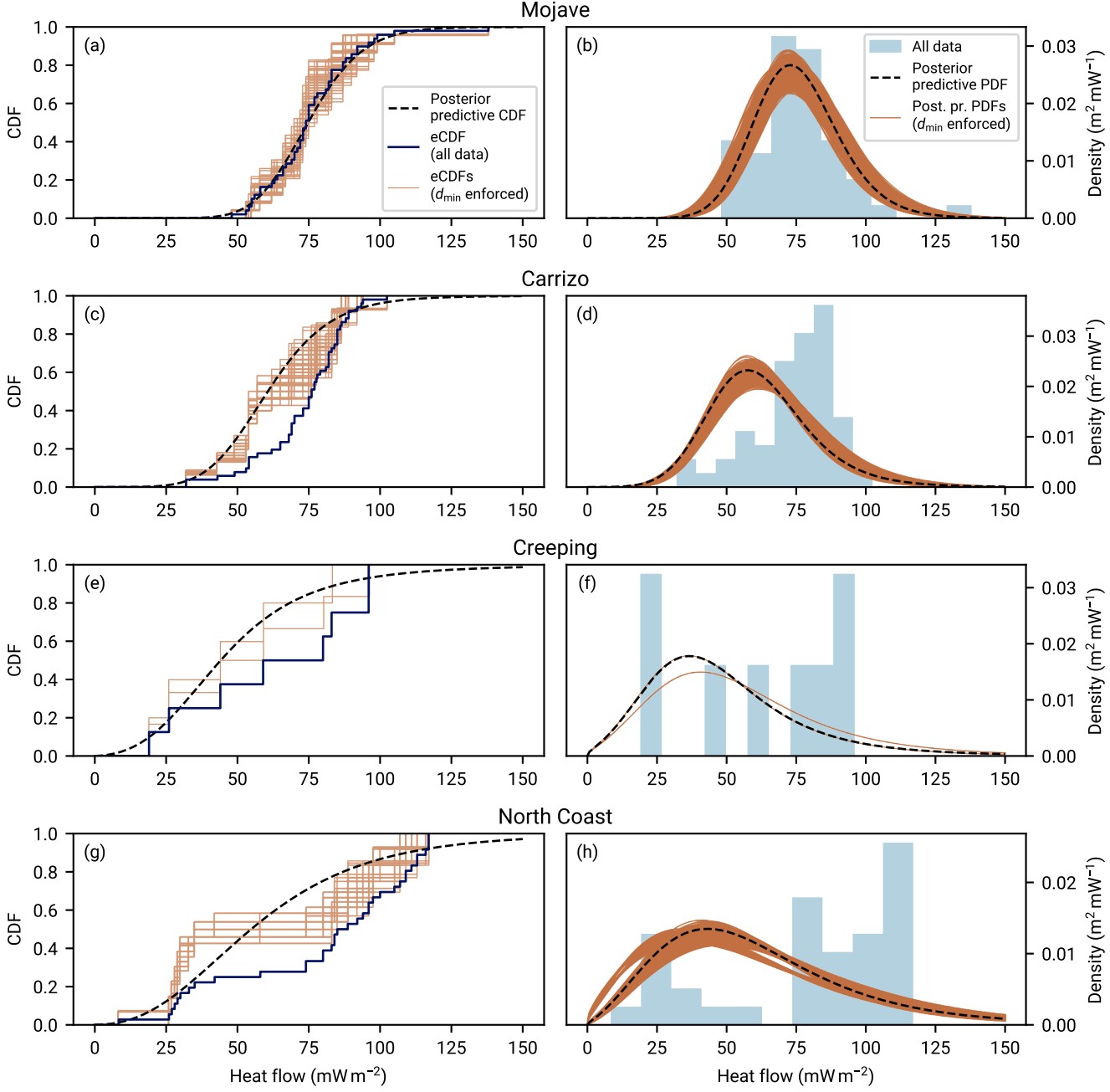

**Figure 20.** Posterior predictive of regional aggregate heat flow for the four regions surrounding the SAF in Fig. 19. The left column, panels (a), (c), (e), and (g), show the cumulative distribution of regional aggregate heat flow data from the NGHF within the region ("eCDF, all data"). The spatial distribution of these data is shown in Figs. 18 and 19. Since some data pairs are within 20 km of each other, the $d_{min}$ sampling approach leads to the set of curves denoted "eCDFs ($d_{min}$ enforced)". The resulting posterior predictive CDF is shown as a dashed line. The right column, panels (b), (d), (f), and (h), shows the densities corresponding to the cumulative distributions of the left column. For each of the eCDFs with $d_{min}$ enforced, the posterior predictive of that data subset is shown. The results, in particular the differences between the histograms and the posterior estimates due to the distance criterion, are discussed in section 5.1.

that the data is concentrated mostly in two clusters, one covering the fault trace and one south-west of it. Only few data points of each cluster are contained in each subset. Note that the two clusters are comprised mostly of high heat flow with respect to the remaining data points. As a result, the subsets' eCDFs and consequently the combined posterior predictive are concentrated at lower heat flow compared to the initial regional aggregate distribution. Besides this thinning effect, the posterior predictive PDF in Fig. 20 (d) seems to be the average of a number of subset predictive PDFs that cover a well-dispersed configuration space.

This is in contrast with the remaining two regions. The Creeping region, panels (e) and (f), contains only eight data points. Four of these data points, clustered across the fault trace, are pairwise within $d_{\min}$. In particular, all four points are within 20 km of the data point closest to the fault. If this data point is contained in the subset, the remaining three largest heat flow measurements are excluded. Since furthermore the largest (pairwise exclusive) heat flow measurements are equal, this leads to only two distinct eCDFs in panel (e) and posterior predictive PDFs in panel (f), both of which are controlled only by the selection of a single data point.

A similar effect occurs in the North Coast region, panels (g) and (h), with more data points (36). Many heat flow measurements are scattered around The Geysers geothermal field (see the location of the Sonoma power plant in Fig. 19). As a result, many of these points are pairwise exclusive under the $d_{\min}$ criterion. While most of the data in that cluster are at the upper end of the range within the region (>80 mW m$^{-2}$), the lowest data point of the region (8.4 mW m$^{-2}$) lies at the edge of the cluster (see panel (d) of Fig. 19). This particular data point has significant control over the aggregate heat flow distribution: the two distinct shapes around which the set of subset posterior predictive distributions scatter in Fig. 20 panel (h) correspond to the subsets in which the 8.4 mW m$^{-2}$ data point is included or not. Furthermore, two to three spatial clusters in panel (d) of Fig. 19 (The Geysers subset, the data close to the south end of the fault trace, and the data beyond the 10$^{-12}$ m$^{-2}$ contour) lead to distinct modes in the histogram in panel (h) of Fig. 20.

## 5.2 Heat flow anomaly

Given the heat flow data from the previous section and the REHEATFUNQ gamma model, we can now investigate the strength of fault-generated heat flow anomalies originating from the SAF segments within the four regions. We assume a conductive mode of heat transfer and use the analytical model of Lachenbruch and Sass (1980) given in Eq. (25), parameterized by the distance to the closest point on the fault segment. We then use the parameterization Eq. (26) to express the anomaly as a function of total frictional power $P_H$ within the four regions. The depths of the fault segments are taken from the UCERF3 model (Milner, 2014). The resulting unit scaled factors $c_i$ are shown in Fig. 19. This concludes step 4 of the workflow listed in Appendix A.

The resulting posterior distributions $f(P_H)$ of the frictional power $P_H$ are shown in Fig. 21 (step 5 of the workflow in Appendix A). They reflect features that we have discussed for the posterior predictive distributions $\psi(P_H)$ in the previous section. As before, the Mojave section, panel (a), features the clearest results. We have marked the frictional power of the "reference anomaly" of Lachenbruch and Sass (1980) (0.92 MW km$^{-2}$ leading to a total power of 146 MW over the fault segment) and we can confirm that this reference anomaly is unlikely. The REHEATFUNQ model places less than 10 % posterior

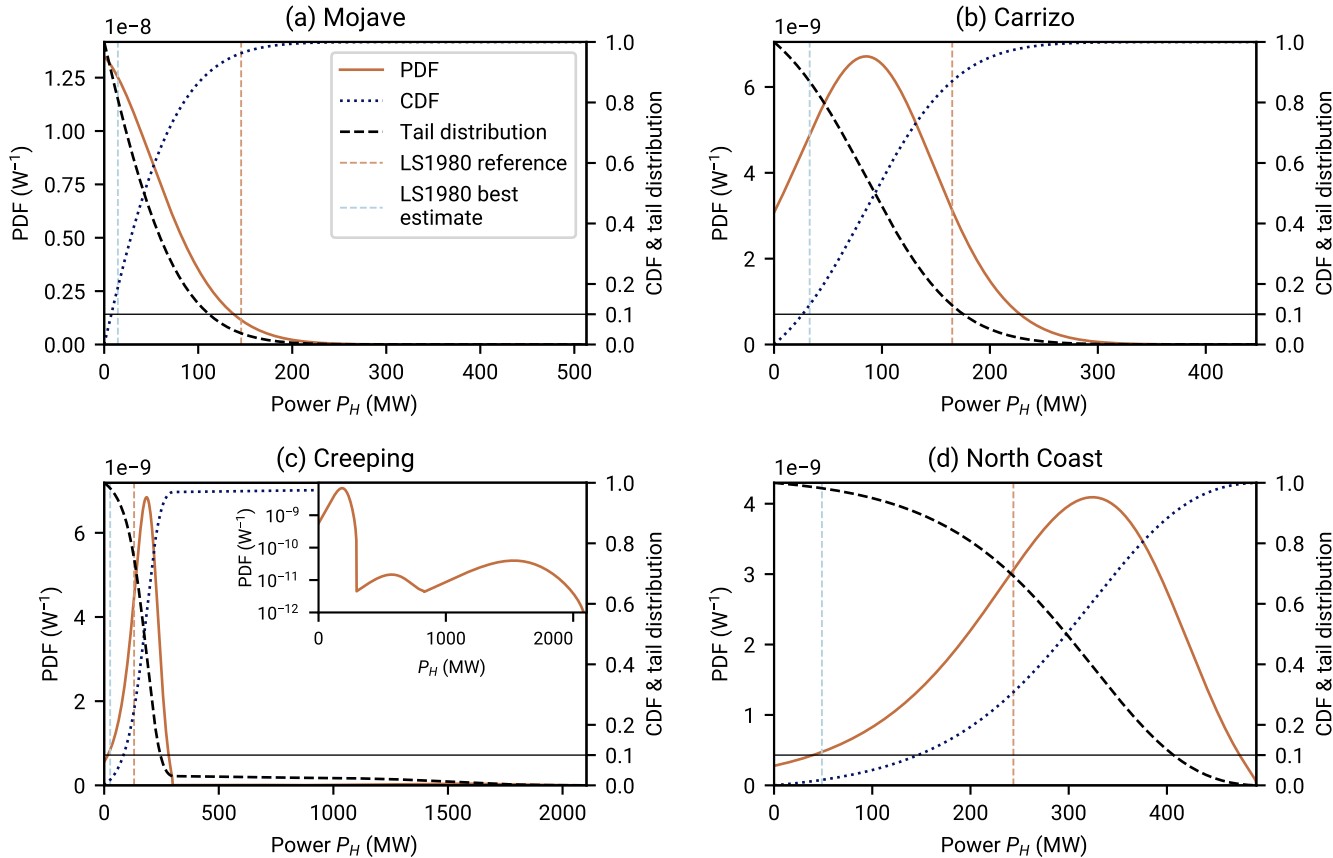

**Figure 21.** Posterior distributions of the frictional power $P_H$ from the analysis of heat flow anomalies surrounding the fault segments in Fig. 19. The heat flow data and heat flow anomaly signature $\{c_i\}$ used in the computation of the posteriors following Eq. (23) is also shown in Fig. 19.

probability to anomalies of similar or larger frictional power. If one were to use the 10 % tail quantile as an upper bound on the fault's frictional power, the upper bound would be 111 MW corresponding to 0.77 "heat flow units" of Lachenbruch and Sass (1980). The best estimate of Lachenbruch and Sass (1980), 10 % of the reference anomaly corresponding to 14.6 MW, is within the bulk of the posterior mass. Under the premise of a stochastic interpretation of the heat flow data, the best estimate of Lachenbruch and Sass (1980) cannot reliably be distinguished from larger bounds within a factor of 8.

In the Carrizo section, panel (b) of Fig. 21, the existence or non-existence of a finite heat flow anomaly is less clear. The PDF has a finite mode between 50 and 100 MW that could indicate the existence of a heat flow anomaly, but the curve is not well-separated from zero. The origin of this peak becomes clear in Fig. 19, panel (b). On one hand there is a high heat flow cluster just next to the north end of the fault segment compared to the north-eastern part of the region, which could support the existence of an anomaly. On the other hand, the cluster south of the fault trace consists of similarly or exceedingly high heat flow, adding the ambiguity that the northern cluster could be due to the generally large range of heat flow from north to

south. Hence, the existence of an anomaly is an option but equally plausible is the overlay of an independent regional trend, for instance the Coast Ranges-Great Valley transition that Williams et al. (2004) have modeled near Parkfield. More geophysical analysis would have to be performed to disentangle the posterior for, say, $P_H < 175$ MW (the 10 % tail quantile in panel (b) of Fig. 21). Only the reference anomaly of Lachenbruch and Sass (1980) seems to be a large enough effect that it could be considered unlikely as-is, being close to the 10 % tail quantile.

As before, the results for the Creeping and North Coast sections are of limited significance due to the data and geologic situation. In the Creeping section, panel (c) of Fig. 21, there is one dominant mode at about 200 MW but there is a long tail, with a few percent probability, up to about 1500 MW. The origin of this tail becomes clear when looking at the logarithm of the PDF (inset): the selection of data points from the cluster of four data points close to the fault trace leads to three separated peaks in the with modes at roughly 200 MW, 500 MW, and 1500 MW. While the mass associated to the two higher peaks is small, this example highlights that the identification of the frictional power $P_H$ from very small data sets is aggravated by the depency on individual data points. In this region, our data filtering has a profound effect (we retain 8 data points of the NGHF while d'Alessio et al. (2006), for instance, use 49 data points) so that a detailed quality analysis of the data set is warranted for further refinement of the analysis.

In the North Coast section, the posterior has a peak at roughly 300 MW. This is likely due to the generally higher heat flow close to the fault compared to the north-east end of the region. The peak is broad, however, and a 90 % credibility interval surrounding the peak is not much smaller than 90 % of $P_H$ domain (e.g. 325 MW from about 10 MW to 425 MW versus 450 MW). The reference anomaly is a bit below to the posterior's median.

The fairly sharp separation into two regions parallel and north to the fault, one with high and one with low heat flow, has already been discussed by Lachenbruch and Sass (1980). They have proposed that the increased heat flow within 100 km of the fault trace might be due a combination of an interaction with the subducting plate and frictional heating within the fault systems surrounding the SAF. This scenario is plausible within the data and highlights that the North Coast region is an example of a region where further geothermal modeling is required before REHEATFUNQ can separate the heat flow anomaly from the regional background heat flow.

In this section, we have used the expression given in Eq. (25) to model all heat flow anomalies. This allowed us to infer the uncertainty inherent to the anomaly quantification due to stochasticity in the heat flow data. What we have not captured in this simple analysis is the uncertainty in the heat flow anomaly itself. For instance in case of the creeping section, a localized creeping asperity and particular rock composition might cause an anisotropic heat flow anomaly (Brune, 2002; d'Alessio et al., 2006).

## 6 Conclusions

This study presented the REHEATFUNQ model for regional aggregate heat flow distributions and the quantification of heat flow anomalies. The REHEATFUNQ model is a new approach to the analysis of regional heat flow by aggregating the data into a single heat flow distribution that is agnostic to the spatial component of the data. Heat flow data is interpreted as the result of

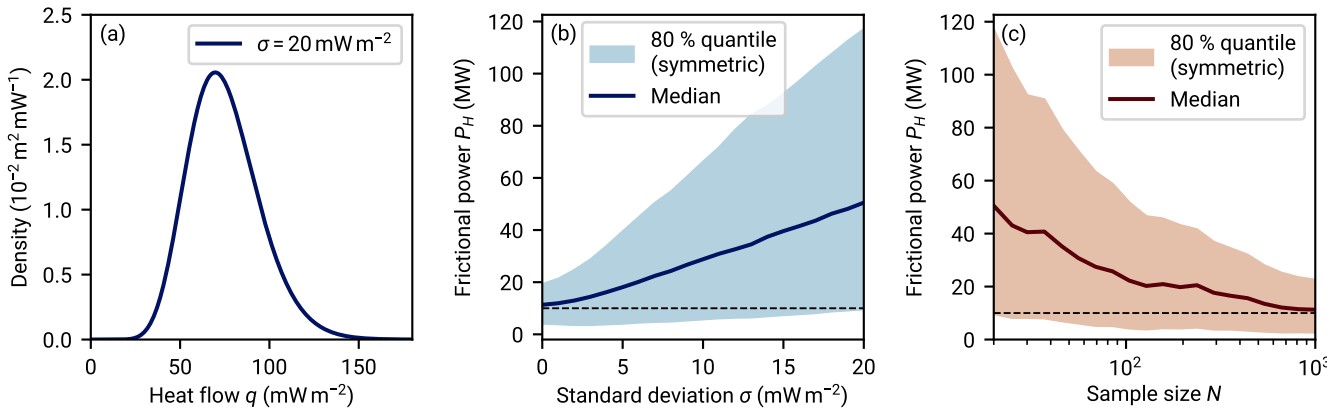

**Figure 22.** Anomaly quantification uncertainty reduction by decreasing heat flow data variance and increasing sample sizes. Panel (a) shows the synthetic regional aggregate heat flow distribution we start from. The gamma distribution has mean $75\,\mathrm{mW\,m^{-2}}$ and standard deviation $20\,\mathrm{mW\,m^{-2}}$, which is similar to the posterior predictive distributions in the Mojave section (panel (b) of Fig. 20). Panels (b) and (c) show the results of quantifying a fault-generated heat flow anomaly superposed on synthetic gamma-distributed regional aggregate heat flow distributions (the fault configuration is the same of Fig. 13 with a depth $d = 14\,\mathrm{km}$). In panel (b), the standard deviation of the distribution in (a) is reduced while keeping the mean and the sample size $N = 20$ constant. This could, for instance, be achieved by removing spatial trends through modeling. In panel (c), the standard deviation is kept constant while the sample size is increased. In both panel (b) and (c), the minimum distance criterion is not enforced.

a stochastic process characteristic to the region. As a result, REHEATFUNQ treats the variability of heat flow measurements

on short spatial scales by design, and uses Bayesian inference both to estimate the regional aggregate heat flow distribution and to quantify the frictional power that generates a potential heat flow anomaly superposed on the regional heat flow. Thereby, REHEATFUNQ can quantify the uncertainty of estimating fault-generated heat flow anomalies from heat flow measurements.

REHEATFUNQ is an empirical model and uses the gamma distribution to model regional aggregate heat flow. Our goodness-of-fit analysis shows that the gamma distribution is not a perfect model. Yet, it is optimal in the sense that among other similarly

simple probability distributions, none is clearly favorable to the gamma distribution. Furthermore, we have tested how resilient the heat flow anomaly quantification of REHEATFUNQ is to real world data-inspired extreme deviations from the gamma distribution hypothesis. Our results show that REHEATFUNQ is successful in determining upper bounds on the strength of fault-generated heat flow anomalies under these conditions, and that it can quantify fairly well the strength of an anomaly if the heat flow data set and the power of the anomaly are sufficiently large. We have found no indication that these results depend

significantly on the region size up to a circumradius of $260\,\mathrm{km}$.

In this article, we focused on the analysis of the heat flow anomaly generated by a vertical fault in a homogeneous half space (Lachenbruch and Sass, 1980), which is implemented in REHEATFUNQ. Other heat flow anomalies, obtained for instance from numerical methods, can easily be used by providing scale coefficients at the data locations. This might be especially important for complex fault geometries, in case of inhomogeneous thermal conductivity, or in the presence of convection.

An application of the REHEATFUNQ model to the San Andreas fault in California highlights that a stochastic interpretation of the heat flow data can significantly relax the upper bound for a fault-generated heat flow anomaly derived by Lachenbruch and Sass (1980). Their best estimate has an amplitude that is low enough such that random fluctuations might have hidden the anomaly. The stochastic approach underlying REHEATFUNQ is hence worth attention and REHEATFUNQ can be a valuable tool for the analysis of regional heat flow values.

The reduction of regional heat flow variability through correction of modeled geothermal patterns and a standardized heat flow data processing, including for instance the removal of topographic effects, may be the most promising path forward to reduce the uncertainties of the heat flow anomaly quantification. While Fig. 22 indicates that a reduction of heat flow variability leads to a proportional reduction of anomaly quantification uncertainty, the uncertainty reduction with increased sample size scales sub-linearly and requires orders of magnitude more (expensive) heat flow data than available today. Steps that can lead to

such a reduction of heat flow variability are the unified correction for topographic effects as performed by Fulton et al. (2004) and detrending based on geothermal modeling (e.g. Cacace et al., 2013). An important aspect might also be the determination of boundaries between regions of (nearly) uniform heat flow characteristics. Our empirical analysis has shown that the mixture of two aggregate heat flow distributions, possibly resulting from the boundary between two such regions, is sufficient to explain the empirical mismatch of the gamma distribution model. Finally, the effort of International Heat Flow Commission to update

the Global Heat Flow Database (Fuchs et al., 2021) might lead to a reduction of heat flow variability.

*Code and data availability.* The REHEATFUNQ model code is available from the GFZ Data Services repository (Ziebarth, 2023) and at Zenodo (Ziebarth, 2024). It can be installed as a Python package or built as a Docker image. It comes with a set of Jupyter notebooks (Pérez and Granger, 2007; Kluyver et al., 2016; Granger and Pérez, 2021) that reproduce the analysis described in this article. These notebooks should make it easy to apply REHEATFUNQ to other areas. Documentation for REHEATFUNQ can be found at https://mjziebarth.github.

io/REHEATFUNQ/ or built from source using sphinx (https://www.sphinx-doc.org/).

    The NGHF data set can downloaded as supplement to the article of Lucazeau (2019). The shoreline data used in this manuscript are from the Global Self-consistent, Hierarchical, High-resolution Shoreline Database (GSHHS, Wessel and Smith, 1996). They can be downloaded from https://www.soest.hawaii.edu/pwessel/gshhg/. The UCERF3 model is available from Milner (2014).

    The REHEATFUNQ model builds upon free software. For numerical computations, REHEATFUNQ builds on the scientific Python stack

of NumPy (van der Walt et al., 2011) and SciPy (Virtanen et al., 2020). The computationally intensive number crunching is written in C++, interfaced via Cython (Behnel et al., 2011), and makes use of the boost math library, GNU MP (Granlund and the GMP development team, 2020), and Eigen (Guennebaud et al., 2010). Spatial computations are performed with the help of GeographicLib (Karney, 2022), PROJ (PROJ contributors, 2022), PyProj (Snow et al., 2022), and Geopandas (Jordahl et al., 2022; The pandas development team, 2022; GDAL/OGR contributors, 2022). Visualizations are created using Matplotlib and FlotteKarte (Hunter, 2007; Ziebarth, 2022b; Crameri,

2021; van der Velden, 2020; Thyng et al., 2016). A number of other numerical software developments are used less prominently (Pedregosa et al., 2011; Ziebarth, 2022a; Giezeman and Wesselink, 2022; Badros et al., 2001; Wang et al., 2013; Fousse et al., 2007)

All compiled software should be readily available within Linux distributions contemporary to this article, and all Python packages are either available on the Python packaging index (https://pypi.org) or are automatically fetched from their repositories. Furthermore, the archived version at GFZ Data Services (Ziebarth, 2023) contains a snapshot of all relevant software packages to build the model.

## Appendix A: Workflow Cheat Sheet

Here we provide a short synopsis of a typical workflow that uses the REHEATFUNQ Python package (`reheatfunq`) to analyze regional aggregate heat flow and quantify the power of an underground heat source with known surface heat flow pattern. Figure 8 accompanies this synopsis and illustrates important steps.

The workflow assumes that a region of interest (ROI) and a set of heat flow data available therein are given. Furthermore, the subject of the analysis is an underground heat source (the *unknown heat source*) whose surface heat flow pattern $q_a(\boldsymbol{x})$ (the *heat flow anomaly*) can be computed but whose power $P_H$ is unknown and should be inferred.

1. Filter the heat flow data. For instance, this might include the removal of any known unreliable measurements or known outliers.

2. Choose a minimum distance parameter $d_{\min}$. This parameter aims to reduce the impact of spatial data clusters by considering heat flow data as potentially dependent if they are located closer than $d_{\min}$. In this article, we use $d_{\min} = 20\,\mathrm{km}$ (see section 3.2).

3. Remove any known spatial heat flow pattern from the heat flow data. This might include topographic effects (e.g. Fulton et al., 2004) but also the surface heat flow generated by a known underground heat source.

4. Compute the coefficients $\{c_i\}$ that specify how much surface heat flow is generated at the heat flow data locations $\{\boldsymbol{x}_i\}$ by the unknown heat source per power $P_H$. This might be done by computing the surface heat flow anomaly $q_a(\boldsymbol{x}_i)$ at the locations of the heat flow measurements $\{q_i\}$ for an arbitrary power $P_H$, and dividing the result by that power ($c_i = q_a(\boldsymbol{x}_i)/P_H$). Any external method can be used to solve the conduction-advection equation for the problem at hand (heat source distribution and boundary conditions) as long as $q_a(\boldsymbol{x})$ can be evaluated at the heat flow data locations and $P_H$ is known. The `AnomalyNearestNeighbor` class can be used to provide the coefficients $c_i$ to the REHEATFUNQ model. The REHEATFUNQ package provides furthermore the solution (A23*b*) of Lachenbruch and Sass (1980) for an infinitely long, surface-rupturing straight strike-slip fault in a homogeneous half space (`AnomalyLS1980`). Figure 8 (a) shows how this anomaly may look compared to some artificial heat flow data.

5. Evaluate the posterior of the unknown heat source's power $P_H$. Use the `HeatFlowAnomalyPosterior` class to compute the posterior PDF, CDF, complementary CDF ('tail distribution'), and tail quantiles of the power $P_H$. Figure 8 (c) shows how the posterior PDF looks like for the example in Fig. 8 (a): the heat flow anomaly was successfully assessed but due to the scatter in the data, the result is uncertain.

Code examples can be found in the documentation shipped with the REHEATFUNQ source code (Ziebarth, 2023) and online at https://mjziebarth.github.io/REHEATFUNQ/. All analysis performed in this article can be reproduced using the Jupyter notebooks (Kluyver et al., 2016) provided with the source code.

## Appendix B: Random global $R$-disk coverings

In the analyses, we frequently make use of *random global $R$-disk coverings* (RGRDCs). The RGRDC is an algorithm to generate a set of many regional aggregate heat flow distributions from the NGHF data set by means of random sampling. The algorithm allows us to capture the typical variability of heat flow within the global data base. Control over the radius $R$ allows us to investigate the global data set over various spatial scales.

In the RGRDCs, disks are randomly distributed across Earth's surface to asymptotically capture the detail of the inhomogeneous data set. Within each disk, data from the filtered NGHF is selected and forms one regional aggregate heat flow distribution. In detail, the algorithm proceeds as follows:

1. Filter the NGHF according to general criteria. All remaining points are unmarked.

2. Draw 100,000 random points $\{p_i\}$ from a spherical uniform distribution.

3. Perform the following for each of them:

    (a) Collect the set $\{x_i\}$ of NGHF data points within a distance $R$ of $p_i$. If any of them are marked discard $p_i$ and continue with $p_{i+1}$.

    (b) Ensure a minimum distance $d_{\min}$ between the $\{x_i\}$ to prevent biases from spatial measurement clusters (see section 3.2). From violating data point pairs, keep one data point at random until the criterion is fulfilled for all remaining $\{x_i\}$.

    (c) if the number of remaining $\{x_i\}$ is larger than 10, keep the disk $p_i$. The remaining $\{x_i\}$ are the corresponding *regional heat flow sample* and are marked.

4. Repeat from step 1 $M$ times to obtain $M$ coverings.

Fig. 1 (a) shows the location of the $R$-disks ($R = 80\,\mathrm{km}$) that have been determined for later use in section 3.3.2.

## B1 Synthetic random global $R$-disk coverings

To understand how well our gamma distribution model of regional heat flow can describe the data from the RGRDCs obtained from the NGHF data base, we generate synthetic coverings from the gamma distribution model and compare statistics of the NGHF coverings with those of the synthetic coverings. To be able to do so, the synthetic coverings need to replicate the sample size structure and the heat flow distribution of the NGHF data coverings. Furthermore, the same data filtering needs to be applied. To this end, we use the following algorithm to generate $M$ synthetic RGRDCs:

1. Generate a set $\mathcal{X} = \{X_1, ..., X_n\}$ of *random global R-disk coverings* $X_i = \{Q_1^i, ..., Q_{m(i)}^i\}$ from the NGHF data base (see Appendix B).

2. For each heat flow distribution $Q_j^i = \{q_1, ..., q_{l(i,j)}\}$, compute the gamma distribution maximum likelihood parameters $(k_{ij}, \theta_{ij})$.

3. Repeat $M$ times:

    (a) Select a random NGHF covering $X_i$.

    (b) For each heat flow distribution $Q_j^i$ within $X_i$, proceed as follows:

        i. generate a heat flow sample of at least $l(i,j)$ data points. Draw these heat flow values from a gamma distributions with parameters $(k_{ij}, \theta_{ij})$.

        ii. For each heat flow value from the sample, draw a random standard deviation $\sigma$ from the relative error distribution (Fig. 9). From a central normal distribution with that $\sigma$, draw a random relative error. Distort the corresponding heat flow value by this relative error.

        iii. Round all heat flow values from the sample to integers.

        iv. Remove all negative heat flow values and all those larger than $250\,\mathrm{mW\,m^{-2}}$.

        v. If less than $l(i,j)$ data points remain, repeat. Otherwise the first $l(i,j)$ accepted heat flow values will form the $j$'th heat flow distribution within the current synthetic covering.

    (c) The generated synthetic covering will match the sample sizes of $X_i$ and have, within the bounds imposed by the gamma distribution and the additional effects, a similar heat flow distribution.

## Appendix C:  Points-of-Interest Measurement Model

This section considers a toy model that aims to mimic a data acquisition process in which few explorative measurements are joined by many dependent measurements that focus on the vicinity of particularly interesting previous measurements. We call these previous measurements "points of interest" (POI). One might imagine that they represent a geothermal or oil field field in which, after its discovery, many data points scatter in close proximity.

The POI model consists of an algorithm that distributes a desired number of sampling points $\{x_k\}$ on a rectangular spatial domain. Synthetic heat flow measurements are generated for these points by querying a pre-generated spatial heat flow field $q(x)$ on the spatial domain. This heat flow field $q(x)$ is represented by a raster filled with values $q_{ij}$ which are in turn drawn from a gamma distribution. Hence, if the points $x_k$ were uniformly distributed, the resulting heat flow sample would follow a gamma distribution (up to discretization effects of the finite raster). With the POI spatial sampling, this is not the case.

The POI sampling algorithm, illustrated in Fig. C1 (a), proceeds as follows:

1. Generate a rasterized heat flow field $q(x)$ by drawing i.i.d. random values from a gamma distribution for each point of the raster.

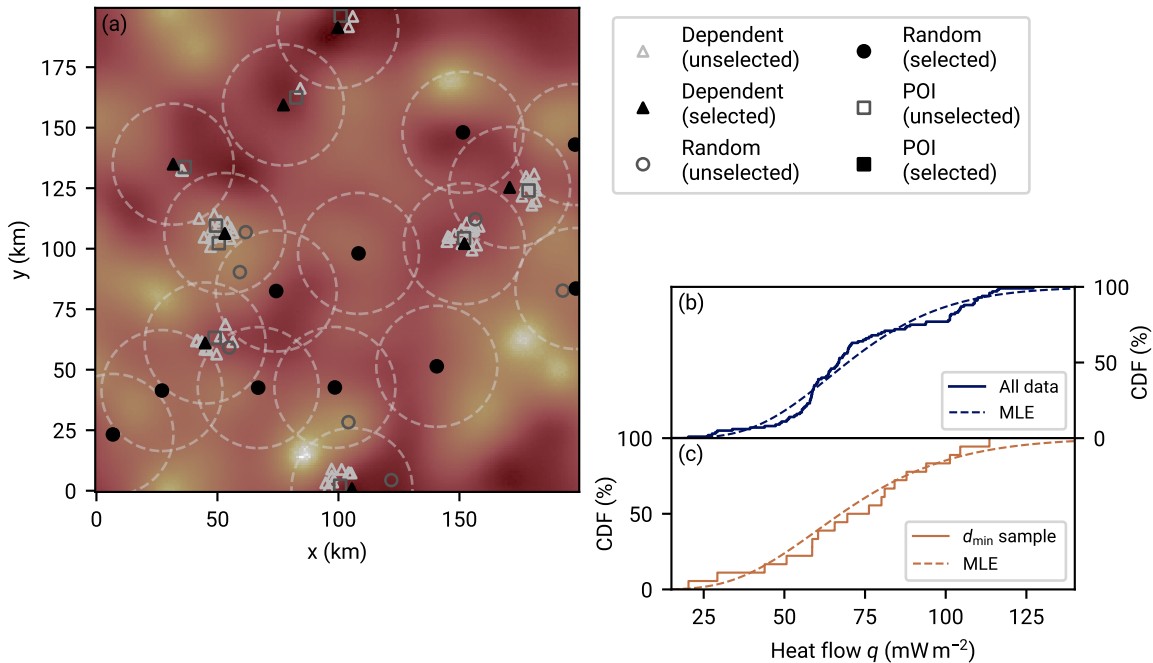

**Figure C1.** The point-of-interest (POI) sampling algorithm to generate heat flow data sets with spatial clustering described in section C. Panel (a): the algorithm generated 100 random sampling points. Each point had a 20 % chance to become a POI (squares) and be randomly distributed over the square. At 70 % probability, a new point would be dependent and generated within an 8 km square surrounding a previous POI (triangles). At 10 %, the point would be randomly distributed but not of interest (circles). The coloring of the markers and the dashed circles surrounding some points illustrate the $d_{min}$ sampling: only the filled black markers are used for the analysis in this particular sample, ignoring all data points within the $d_{min}$ disks surrounding them. Panels (b) and (c) show the impact that the POI point generation and the $d_{min}$ sampling have on the aggregate heat flow distribution. Panel (b) shows the empirical cumulative distribution function (eCDF) of the full data set as well as the CDF of the corresponding gamma distribution maximum likelihood estimate (MLE). Panel (c) shows the same for only the selected data points of panel (a). Deviations of the eCDF from the CDF in panel (a) due to the clustered sample points (steep slopes in the eCDF) are successfully removed in panel (c).

2. Iteratively flip random pixel pairs of the heat flow raster $q_{ij}$ if the flip reduces the variance in the local neighborhoods of the both pixels. This smoothens the field while retaining its aggregate gamma distribution.

3. Generate a first POI from a uniform distribution across the spatial domain.

4. Sequentially for each of the remaining $N - 1$ requested points, choose one of the following actions:

    (a) With probability $P_{\text{POI}}$ generate a new POI

    (b) With probability $P_{\text{f}}$ generate a follow up point. Choose one of the existing POIs at random and place the follow up point within a square of side length $R$ of the selected POI

    (c) With remaining probability, generate a non-POI point uniformly in the spatial domain.

5. For each of the generated points, determine the measured heat flow value from the heat flow field $q_(\boldsymbol{x})$.

The general idea of this clustered point sampling is reflected in the assumptions made in the formulation of the REHEATFUNQ likelihood in section 3.3.1. In the POI sampling, the dependent sampling points that attach to a POI lead to correlations in the aggregate heat flow distribution (see Fig. C1 b) due to the spatial correlation of the underlying heat flow field. One point of the cluster—the POI—is spatially independent and leads to a random gamma-distributed heat flow measurement. Assumption II of section 3.3.1 takes this into account by using only one data point of a cluster identified by the $d_{\text{min}}$-disk at the same time. Assumption III then acknowledges that each point of the cluster is equally likely to be the independent, and the summation over all possible choices for independent points (latent parameter $j$) ensures that no data point is left behind.

In Fig. C1 (c), we can indeed observe that the enforcement of $d_{\text{min}}$ results in more gamma-like samples compared to the full data set (panel b). We have furthermore observed from Monte-Carlo simulations of the POI sampling that the likelihood approach in Eq. (5) leads to more accurate recovery of the true regional aggregate heat flow distribution (Fig. C2) and reduces bias in the estimation of heat flow anomalies (Fig. C3) when compared to using all heat flow data in the presence of spatial clustering.

## Appendix D: Expressions for numerical quadrature

### D1 Gamma conjugate prior

The normalization constant $\Phi$, Eq. (9), requires one numerical quadrature for the evaluation of the $\alpha$ integral. To this end, we first compute the location of the integrand's maximum $\alpha_{\text{max}}$ using an approximation based on Stirling's formula,

$$\alpha_0 = \exp\left(\frac{\ln p - \nu s + \nu \ln \nu}{n - \nu}\right), \tag{D1}$$

followed by Newton-Raphson refinement. Integration is then performed in the intervals $[\alpha_{\text{min}}, \alpha_{\text{max}}]$ and $[\alpha_{\text{max}}, \infty[$ using tanh-sinh and exp-sinh quadrature respectively (Takahasi and Mori, 1974; Tanaka et al., 2009). To prevent the integrand from

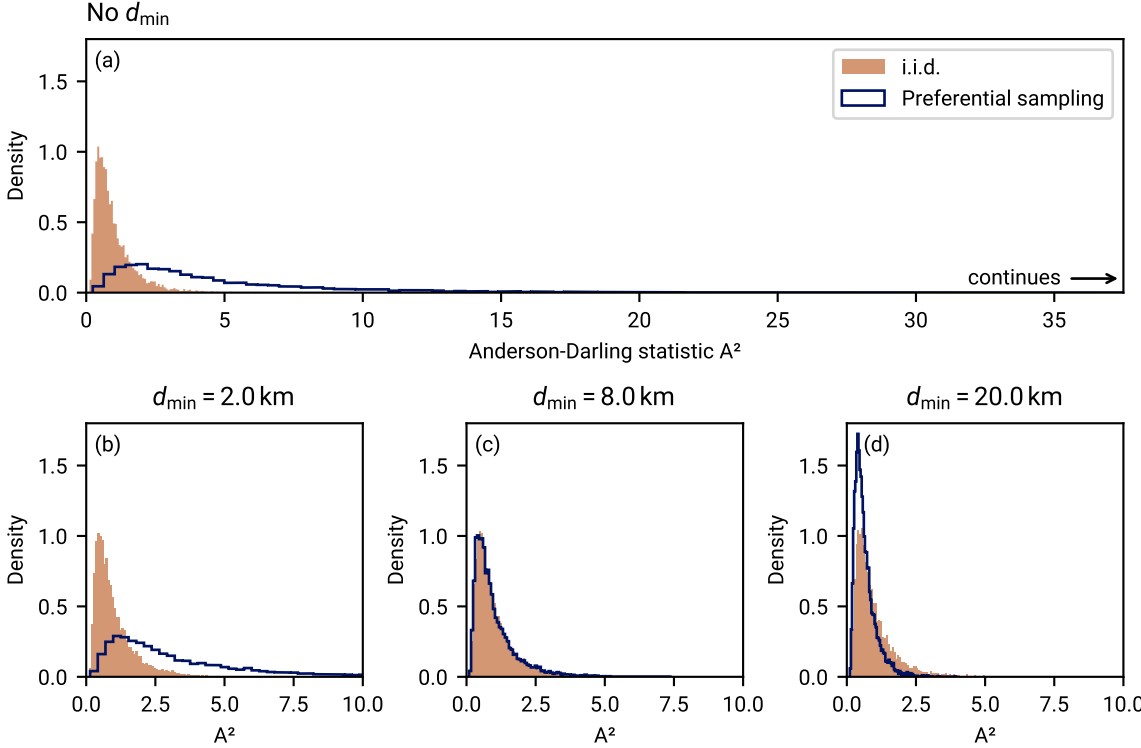

**Figure C2.** Improvement of the gamma distribution likeness of heat flow samples generated by the point-of-interest (POI) sampling method (Appendix C) when using the minimum distance sampling. Each panel shows the distribution of the Anderson-Darling statistic $A^2$ (Stephens, 1986) for the gamma distribution, a measure of how well the sample matches the gamma distribution (less being a better match). The filled histograms labeled 'i.i.d.' show the distribution of $A^2$ that follows from independent and identically distributed gamma random variables with $\alpha = 10$. The histograms marked by the dark blue line show the distribution of $A^2$ generated from the POI sampling method with a gamma landscape generated for $\alpha = 10$ (shown in Fig. C1), and with $P_{\text{POI}} = 20\,\%$, $P_f = 70\,\%$, and $R = 8\,\text{km}$. The minimum distance criterion has then been applied with the $d_{\text{min}}$ specified in the panel titles. The good match between both distributions in panel (c) shows that an accurately chosen $d_{\text{min}}$ can counter the spatial clustering effect of the POI sampling model. If $d_{\text{min}}$ is chosen too small, the clustering is not effectively countered and large $A^2$ compared to the i.i.d. histograms indicate significant departures from the gamma distribution. If $d_{t}extmin$ is chosen too large, the heat flow data generated from the POI model show less difference to the gamma distribution CDF than the actual i.i.d. gamma random variable. This is due to the minimum distance criterion removing *too many* clusters, that is, also those clusters that naturally appear for uniform random variables.

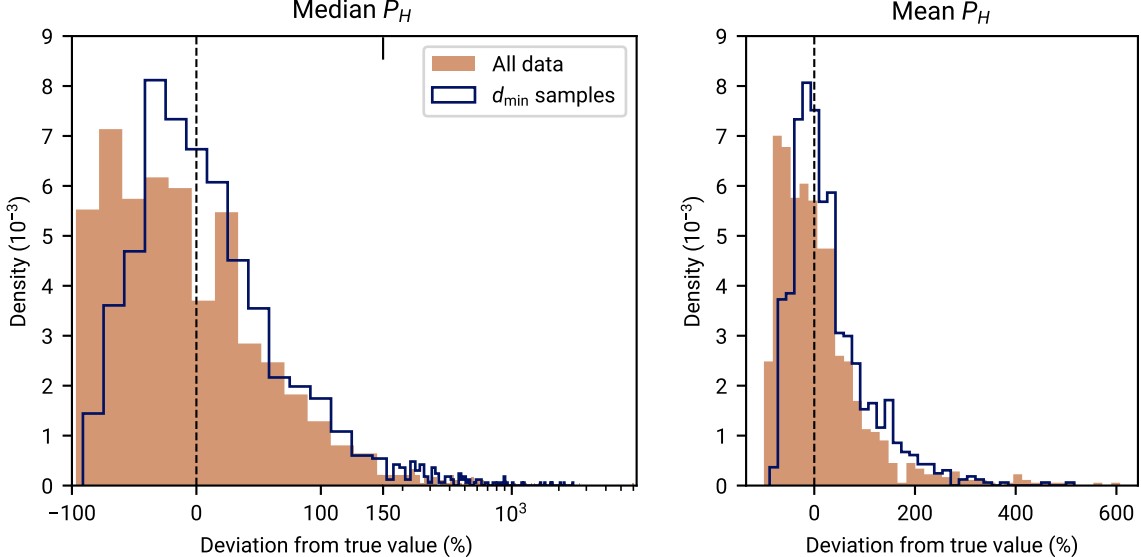

**Figure C3.** Improvement of the accuracy of the posterior $P_H$ estimates for point-of-interest (POI) generated data as the minimum distance sampling is enforced. Both panels analyze data generated from 1000 POI sampling runs with $P_{POI} = 12\%$, $P_f = 80\%$, $R = 5\,\text{km}$, and the gamma-distributed field in Fig. C1. The sample size is $N = 50$ and $d_{min}$ is 5 km. Panel (a) shows how the relative deviation of the $P_H$ posterior's median from the true $P_H$ is distributed across the 1000 POI runs. For each run, the $P_H$ posterior is evaluated on all data ($d_{min} = 0$) and with minimum distance enforced ($d_{min} = 5\,\text{km}$). Using the minimum distance criterion improves the accuracy of the median estimator. Panel (b) shows the same for the posterior mean $P_H$. Similarly, the use of the minimum distance criterion reduces bias.

overflowing we use a transform of the type

$$f(\alpha) = \exp\left(\ln f(\alpha) - \ln f(\alpha_{\max})\right) \tag{D2}$$

where $f$ is the integrand of Eq. (9) and $\ln f$ is expressed through the logarithmic versions of its constituents. This rescaling cancels in normalization.

    The piecewise integrals involved in evaluating the posterior predictive CDF at a set of points $\{q_i\}$ for a given $\Phi$ are computed
using adaptive Gauss-Kronrod quadrature (Laurie, 1997; Kronrod, 1965; Gonnet, 2012).

### D1.1   Kullback-Leibler divergence

The Kullback-Leibler divergence $K$ is given in Eq. (14). For the gamma conjugate prior with variables $\boldsymbol{x} = (\alpha, \beta)$, the expression reads

$$K\Big(\phi(\alpha,\beta), \phi_{\text{ref}}(\alpha,\beta)\Big) = \int\limits_{\alpha_{\min}}^{\infty} \mathrm{d}\alpha \int\limits_{0}^{\infty} \mathrm{d}\beta\, \frac{1}{\Phi} \frac{\beta^{\nu\alpha-1} p^{\alpha-1} e^{-s\beta}}{\Gamma(\alpha)^n} \ln\left(\frac{\Phi_{\text{ref}}}{\Phi} \frac{\beta^{\nu\alpha-1} p^{\alpha-1} e^{-s\beta}\Gamma(\alpha)^{n_{\text{ref}}}}{\beta^{\nu_{\text{ref}}\alpha-1} p_{\text{ref}}^{\alpha-1} e^{-s_{\text{ref}}\beta}\Gamma(\alpha)^{n}}\right) \tag{D3}$$

where $(p, s, n, \nu)$ are the parameters of $\phi$ with normalization $\Phi$ and $(p_{\text{ref}}, s_{\text{ref}}, n_{\text{ref}}, \nu_{\text{ref}})$ belong to the reference model $\phi_{\text{ref}}$ with normalization $\Phi_{\text{ref}}$, and $\Gamma(\alpha)$ is the gamma function. After a bit of algebra, the integrals can be converted to the following expression (with $K$ being shortened notation in the following):

$$K = \ln\left(\frac{\Phi_{\text{ref}}}{\Phi}\right) + \frac{1}{\Phi}\int\limits_{\alpha_{\text{min}}}^{\infty} d\alpha \frac{p^{\alpha-1}}{\Gamma(\alpha)^n}\left[\left((\alpha-1)\ln\frac{p}{p_{\text{ref}}} - \Delta n \ln\Gamma(\alpha)\right)\int\limits_{0}^{\infty} d\beta\, \beta^{\nu\alpha-1}e^{-s\beta}\right.$$

$$- \Delta s \int\limits_{0}^{\infty} d\beta\, \beta^{\nu\alpha}e^{-s\beta}$$

$$\left. + \alpha\Delta\nu \int\limits_{0}^{\infty} d\beta\, \ln(\beta)\beta^{\nu\alpha-1}e^{-s\beta}\right], \tag{D4}$$

where we have used the abbreviations

$$\Delta s = s - s_{\text{ref}}, \quad \Delta n = n - n_{\text{ref}}, \quad \text{and} \quad \Delta\nu = \nu - \nu_{\text{ref}}. \tag{D5}$$

The three $\beta$ integrals can be evaluated analytically with a little help from SymPy (Meurer et al., 2017):

$$\int\limits_{0}^{\infty} d\beta\, \beta^{\nu\alpha-1}e^{-s\beta} = \frac{\Gamma(\nu\alpha)}{s^{\alpha\nu}} \tag{D6}$$

$$\int\limits_{0}^{\infty} d\beta\, \beta^{\nu\alpha}e^{-s\beta} = \frac{\Gamma(\nu\alpha+1)}{s^{\alpha\nu+1}} \tag{D7}$$

$$\int\limits_{0}^{\infty} d\beta\, \ln(\beta)\beta^{\nu\alpha-1}e^{-s\beta} = \frac{\left(\psi(\alpha\nu) - \ln s\right)\Gamma(\nu\alpha+1)}{\alpha\nu s^{\nu\alpha}}, \tag{D8}$$

where $\psi(x)$ is the digamma function. After some more algebra, the expression used in REHEATFUNQ to estimate the Kullback-Leibler divergence from the reference model $(p_{\text{ref}}, s_{\text{ref}}, n_{\text{ref}}, \nu_{\text{ref}})$ to the model $(p, s, n, \nu)$ is

$$K = \ln\left(\frac{\Phi_{\text{ref}}}{\Phi}\right) + \frac{1}{\Phi}\int\limits_{\alpha_{\text{min}}}^{\infty} d\alpha \frac{p^{\alpha-1}\Gamma(\nu\alpha)}{s^{\nu\alpha}\Gamma(\alpha)^n}\left[(\alpha-1)\ln\frac{p}{p_{\text{ref}}} - \Delta n\ln\Gamma(\alpha) - \frac{\nu\alpha\Delta s}{s} + \alpha\Delta\nu\left(\psi(\nu\alpha) - \ln s\right)\right]. \tag{D9}$$

The integral (D9) is solved in REHEATFUNQ using the tanh-sinh integration routine implemented in the Boost C++ library (Takahasi and Mori, 1973). The bracketed part of the integrand in Eq. (D9) typically has a change in sign, which we have observed to lead to a condition number ($L_1/|Q|$, where $L_1$ is the integral of the absolute integrand and $Q$ the integral of the signed integrand) of ~$10^3$. This is indicative of a precision loss of three digits (Agrawal et al.) but we have not observed further numerical difficulties that could lead to great loss of usefulness.

## D2  $P_H$ marginal posterior normalization

Evaluating the integrand of the marginal posterior, Eq. (23),

$$\gamma\left(\{\Delta_i\}_j; \alpha, \beta\right)\phi(\alpha, \beta) \sim \frac{\beta^{\nu'\alpha-1}\left(p\prod_{i\in\mathcal{I}_j}\Delta_i\right)^{\alpha-1}\exp\left(-\left(s+\sum_{i\in\mathcal{I}_j}\Delta_i\right)\beta\right)}{\Gamma(\alpha)^{n+N_j}}, \tag{D10}$$

$$\nu' = \nu + N_j, \qquad \Delta_i = q_i - P_H c_i$$

where $N_j$ is the number of data points indexed in $\mathcal{I}_j$, leads to the modified hyperparameter update rule

$$\nu \to \nu'_j = \nu + N_j, \quad n \to n'_j = n + N_j, \quad s \to s'_j = s + \sum_{i\in\mathcal{I}_j}(q_i - P_H c_i), \tag{D11}$$

$$p \to p'_j = p\prod_{i\in\mathcal{I}_j}(q_i - P_H c_i)$$

given a sample $\{q_i\}_j$ of $N_j$ regional heat flow measurements. Since $P_H$ occurs both in $s$ and $p$ and nonlinearly in the latter, it breaks the conjugacy of the prior, i.e. the posterior does not have the same functional shape. Additionally the normalization constant changes:

$$\mathcal{F} = \sum_{j=1}^m\int_0^{P_H^{\mathrm{m}}}\mathrm{d}P_H\int_0^\infty\mathrm{d}\beta\int_0^\infty\mathrm{d}\alpha\,\frac{\beta^{\nu'_j\alpha-1}(p'_j)^{\alpha-1}\exp\left(-s'_j\beta\right)}{\Gamma(\alpha)^{n'_j}}$$

$$= \sum_{j=1}^m\int_0^{P_H^{\mathrm{m}}}\mathrm{d}P_H\int_0^\infty\mathrm{d}\alpha\,\frac{\Gamma(\nu'_j\alpha)p^{\alpha-1}}{\Gamma(\alpha)^{n'_j}}\frac{\prod_{i\in\mathcal{I}_j}(q_i - P_H c_i)^{\alpha-1}}{\left(s+\sum_{i\in\mathcal{I}j}(q_i - P_H c_i)\right)^{\nu'_j\alpha}}$$

$$= \sum_{j=1}^m\Psi_j \tag{D12}$$

For numerical evaluation we use the transform

$$z = \frac{P_H}{P_H^{\mathrm{m},j}} \tag{D13}$$

and express these integrals by

$$\Psi_j = P_H^{\mathrm{m},j}\int_0^1\mathrm{d}z\int_0^\infty\mathrm{d}\alpha\,I_j(\alpha, z) \tag{D14}$$

with the integrand

$$I_j(\alpha, z) = \frac{\Gamma(\nu'_j\alpha)(\tilde{p}_j)^{\alpha-1}}{\Gamma(\alpha)^{n'_j}(\tilde{s}_j)^{\nu'_j\alpha}}\frac{\prod_{i\in\mathcal{I}_j}\left(1-\kappa_i^{(j)}z\right)^{\alpha-1}}{\left(1-\omega_j z\right)^{\nu'_j\alpha}} \tag{D15}$$

and the parameters

$$\tilde{s}_j = s + \sum_{i \in \mathcal{I}_j} q_i, \qquad B_j = \sum_{i \in \mathcal{I}_j} c_i, \qquad \omega_j = \frac{B_j P_H^{\mathrm{m},j}}{\tilde{s}_j},$$

$$\tilde{p}_j = p \prod_{i \in \mathcal{I}_j} q_i, \qquad \kappa_i^{(j)} = \frac{P_H^{\mathrm{m},j} c_i}{q_i}, \qquad P_H^{\mathrm{m},j} = \min_{i \in \mathcal{I}_j} \left\{ \frac{q_i}{c_i} \right\}.$$

To avoid overflow, we evaluate the integrand $I_j(\alpha, z)$ as an exponentiated sum of logarithms. The full posterior is then, with parameters updated as described above:

$$\psi(P_H, j, \alpha, \beta) = \frac{\beta^{\nu'_j \alpha - 1} \left(p'_j\right)^{\alpha - 1} \exp\left(-s'_j \beta\right)}{\mathcal{F} \Gamma(\alpha)^{n'_j}} \tag{D16}$$

The marginal posterior density of $P_H$ then reads

$$f\left(P_H \,|\, p, s, n, \nu, \{(q_i, c_i)\}\right) = \frac{1}{\mathcal{F}} \sum_{j=1}^{m} \int_0^\infty \mathrm{d}\alpha \, I_j\left(\alpha, \frac{P_H}{P_H^{\mathrm{m},j}}\right). \tag{D17}$$

### D2.1  Rescaling integrands

Before computing the integrals in $\Psi$ using numerical integration, it is helpful to determine the maximum of the integrand and normalize relatively. This can reduce the chance of overflow in the integration routine and, by splitting the integration interval in $\alpha$ at the maximum, stabilize the numerical integration routine (we use $\tanh$-sinh and $\exp$-sinh quadrature from the boost software package with a Gauss-Kronrod fallback if an error occurs). REHEATFUNQ furthermore computes the global maximum of the integrand in $\alpha$ using a combined Newton-Raphson and TOMS 748 method (Alefeld et al., 1995) to get an initial estimate of the global norm scale. This is then used to rescale the integrand:

$$\ln \Psi_j = \ln P_H^{\mathrm{m},j} + \ln \left( \int_0^1 \mathrm{d}z \int_0^\infty \mathrm{d}\alpha \, I_j(\alpha, z) \right)$$

$$= \ln P_H^{\mathrm{m},j} + \ln I_{\max}^{(j)} + \ln \left( \int_0^1 \mathrm{d}z \int_0^\infty \mathrm{d}\alpha \, \exp\left( \ln I_j(\alpha, z) - \ln I_{\max}^{(j)} \right) \right) \tag{D18}$$

Here $I_j(\alpha, z)$ denotes the integrand in Eq. (D14). The nominators occurring in the resulting PDF and CDF are then equally rescaled.

### D2.2  Large $P_H$ ($z \to 1$)

In this section, we leave out the indices $j$ for brevity. All variables defined in the previous sections are to be considered $j$-indexed as before. Products $\prod_i$ are to be considered as products over $i \in \mathcal{I}_j$. Where required by dependency on $j$-indexed variables of previous sections, variables introduced in this section are to be considered $j$-indexed (this applies to most variables).

For $z$ approaching 1, the double integral of Eq. (D14) can become unstable due to the product of the $1 - \kappa_i z$ approaching zero. This is caused by the largest $\kappa_i$ at index $i = i_{\max}$ which is one by definition. Hence, it is helpful to use a Taylor expansion in $y = 1 - z$ to explicitly compute the integral for $z$ above a suitable threshold $1 - y_{\mathrm{m}}$ close to one. With this change of variables, the high-$z$ part of the integral $\Psi$ becomes

$$\int_{1-y_{\mathrm{m}}}^{1} \mathrm{d}z \int_{0}^{\infty} \mathrm{d}\alpha \, I(\alpha, z) = \int_{0}^{y_{\mathrm{m}}} \mathrm{d}y \int_{0}^{\infty} \mathrm{d}\alpha \, \frac{\Gamma(\nu'\alpha)(\tilde{p})^{\alpha-1}}{\Gamma(\alpha)^{n'} (\tilde{s})^{\nu'\alpha}} y^{\alpha-1} g(\alpha, y) \tag{D19}$$

where we have defined $g$ through

$$\frac{\prod_i (1 - \kappa_i z)^{\alpha-1}}{(1 - \omega z)^{\nu'\alpha}} = y^{\alpha-1} \underbrace{\frac{\prod_{i \neq i_{\max}} (1 - \kappa_i + \kappa_i y)^{\alpha-1}}{(1 - \omega + \omega y)^{\nu'\alpha}}}_{g(\alpha, y)} \tag{D20}$$

We now aim to expand $g(\alpha, y)$ into powers of $y$ which will allow us to compute analytically the $y$ integrals in Eq. (D19). Then, approximate the integral by retaining only a finite order of the expanded polynomial. For this, we first expand the product $i \neq i_{\max}$ into the first four polynomial coefficients:

$$\prod_i (1 - \kappa_i z) = y \prod_{i \neq i_{\max}} (1 - \kappa_i + \kappa_i y)$$
$$= y(h_0 + h_1 y + h_2 y^2 + h_3 y^3 + \mathcal{O}(y^4)) \tag{D21}$$

where $h_0$ to $h_2$ are the expansion coefficients of the second product. After some algebra, this leads to the following approximation of the integral for $z$ close to one:

$$\int_{1-y_{\mathrm{m}}}^{1} \mathrm{d}z \int_{0}^{\infty} \mathrm{d}\alpha \, \frac{\Gamma(\nu'\alpha)(\tilde{p})^{\alpha-1}}{\Gamma(\alpha)^{n'} (\tilde{s})^{\nu'\alpha}} \frac{\prod_i (1 - \kappa_i z)^{\alpha-1}}{(1 - \omega z)^{\nu'\alpha}} \approx \int_{0}^{\infty} \mathrm{d}\alpha \, \frac{\Gamma(\nu'\alpha)(h_0\tilde{p})^{\alpha-1}}{\Gamma(\alpha)^{n'} (\tilde{s}(1-\omega))^{\nu'\alpha}} \sum_{k=0}^{3} \frac{y_{\mathrm{m}}^{\alpha+k}}{\alpha+k} C_k(\alpha) \tag{D22}$$

$$C_0(\alpha) = 1 \tag{D23}$$

$$C_1(\alpha) = \frac{(\alpha-1)h_1}{h_0} - \frac{\nu'\alpha\omega}{1-\omega} \tag{D24}$$

$$C_2(\alpha) = \frac{1}{2} \left( \frac{(\alpha-1)(\alpha-2)h_1^2}{h_0^2} + \frac{2(\alpha-1)h_2}{h_0} - \frac{2\nu'\alpha\omega(\alpha-1)h_1}{h_0(1-\omega)} + \frac{\nu'\alpha(\nu'\alpha-1)\omega^2}{(1-\omega)^2} \right) \tag{D25}$$

$$C_3(\alpha) = \Bigg[ \quad \alpha^3 \left( \nu^3\omega^3 + 3h_1 h_0 \nu^2 \omega^2 (\omega-1) + 3\frac{h_1^2}{h_0^2} \nu\omega(\omega(\omega-2)+1) \right.$$
$$\left. + \frac{h_1^3}{h_0^3} (\omega(\omega^2 - 3\omega + 3) - 1) \right)$$
$$+ 3\alpha^2 \left( \nu^2\omega^3 + \frac{h_1}{h_0} \nu\omega^2 (\nu-1)(1-\omega) + 2\frac{h_2}{h_0} \nu\omega(\omega-1)^2 \right.$$

$$+3\frac{h_1^2}{h_0^2}\nu\omega\left(\omega(2-\omega)-1\right)+2\left(\frac{h_1h_2}{h_0^2}-\frac{h_1^3}{h_0^3}\right)\left(\omega(\omega^2-3\omega+3)-1\right)\Bigg)$$

$$+\quad\alpha\left(2\nu\omega^3+3\nu\omega(\omega(\frac{h_1}{h_0}(1-\omega)+2\frac{h_2}{h_0}(2-\omega))-2\frac{h_2}{h_0})+6\frac{h_1^2}{h_0^2}\nu\omega(\omega^2-2\omega+1)\right.$$

$$\left.+\left(6\frac{h_3}{h_0}-18\frac{h_1h_2}{h_0^2}+11\frac{h_1^3}{h_0^3}\right)(\omega^3-3\omega^2+3\omega-1)\right)\Bigg]$$

$$\times\frac{1}{\omega^3-3\omega^2+3\omega-1}+6\left(2\frac{h_1h_2}{h_0^2}-\frac{h_3}{h_0}-\frac{h_1^3}{h_0^3}\right) \tag{D26}$$

### D2.3 Asymptotics of $\Gamma(\nu\alpha)/(n\Gamma(\alpha))$

By far the most expensive operation when numerically integrating $I_j(\alpha,z)$ is to evaluate the two $\ln\Gamma$ functions:

$$I_j(\alpha,z)=\exp\left(\ln\Gamma(\nu_j'\alpha)-n_j'\ln\Gamma(\alpha)+\dots\right). \tag{D27}$$

Furthermore, the difference between these two dominant terms in the exponent of $I$ can lead to a catastrophic loss of precision for large $\alpha$, leading to numerical difficulty in evaluating $I_j$. For these two reasons, we used SymPy (Meurer et al., 2017) express the difference of the two $\ln\Gamma$ functions as an asymptotic series expansion:

$$\Delta\ln\Gamma\left(\alpha\,|\,n,\nu\right)=\ln\Gamma(\nu\alpha)-n\ln\Gamma(\alpha)$$

$$=\alpha\Big((n-\nu)(1-\ln\alpha)+\nu\ln\nu\Big)$$

$$+\frac{1}{2}\Big((n-1)(\ln\alpha-\ln 2\pi)-\ln\nu\Big)$$

$$+\frac{1}{12\alpha}\left(\frac{1}{\nu}-n+\frac{1}{30\alpha^2}\left[n-\frac{1}{\nu^3}+\frac{2}{7\alpha^2}\left(\frac{1}{\nu^5}-n+\frac{3}{4\alpha^2}\left[n-\frac{1}{\nu^7}\right]\right)\right]\right)$$

$$+\mathcal{O}\left(\frac{1}{\alpha^9}\right) \tag{D28}$$

To estimate the error of that series expansion, we use the leading order error term

$$\Delta(\alpha)=\left|\frac{n-\frac{1}{\nu^9}}{1188\alpha^9}\right|. \tag{D29}$$

If $\Delta(\alpha)$ is less than machine $\epsilon$ compared to the value obtained from Eq. (D28), we use the expansions, while otherwise the $\ln\Gamma$ functions are explicitly evaluated.

The expansion Eq. (D28) is computed for $\alpha\to\infty$ and becomes increasingly imprecise at small $\alpha$. To avoid having to explicitly compute $\ln\Gamma$ functions also for small $\alpha$, we use the argument shift technique described by Johansson (2023). The argument shift is based on the recurrence relation of the $\Gamma$ function and allows to express the $\Gamma$ function through arguments shifted by integer values $M$ (index $j$ omitted for brevity):

$$(x)_M=x(x+1)\cdots(x+M-1)=\frac{\Gamma(x+M)}{\Gamma(x)}. \tag{D30}$$

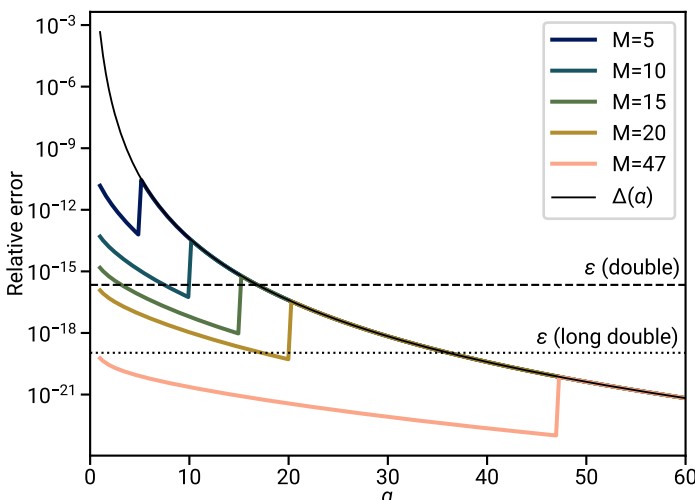

**Figure D1.** Relative error of the series expansion Eq. (D28) with argument shift $M$, Eq. (D31), as a function of $\alpha$. The argument shift is applied for $a < M$. The series expansion is then compared to the actual function, $\ln\Gamma(\nu\alpha) - n\ln\Gamma(\alpha)$. The functions have been evaluated with 100 digits precision using MPMath (The mpmath development team, 2023) for parameters $n = \nu = 1.1$. The situation is similar for some other combinations of $n$ and $\nu$ in the range $0 < \nu \leq n < 2$.

Applied to Eq. (D28) we find

$$\Delta\ln\Gamma\big(\alpha\,|\,n,\nu\big) = n\ln(\alpha)_M - \ln(\nu\alpha)_M + \Delta\ln\Gamma\left(\alpha + M\,\Big|\,n, \frac{\nu\alpha + M}{\alpha + M}\right). \tag{D31}$$

To compute $\ln(\alpha)_M$ and $\ln(\nu\alpha)_M$, we iteratively multiply the sequence $(x)_M$ within the dynamic range of the floating point type. Whenever overflow impends (say at index $o$), an intermediate logarithm is computed and the remaining product sequence $(x + o)_{M-o}$ is evaluated separately. Finally, all intermediary and the final logarithm(s) are summed.

Using high precision evaluation of Eq. (D31) and the actual difference $\ln\Gamma(\nu\alpha) - n\ln\Gamma(\alpha)$, we have found that an argument shift $M = 47$, applied for $a < 47$, is a compromise that leads to relative errors below `long double` precision for most of the $\alpha$ range given some common parameter combinations of $n$ and $\nu$ (see Fig. D1).

### D3  Interpolating the marginal $P_H$ density $f(P_H)$

The PDF $f(P_H)$ is the base for all uses of the marginal $P_H$ posterior. Due to the required $\alpha$ integration, but in particular due to the latent dimension $j$, $f(P_H)$ is an expensive bottleneck of the REHEATFUNQ model. To reduce the required computation time once the normalization has been computed, we provide a barycentric Lagrange interpolant (Berrut and Trefethen, 2004) that can be (and by default is) used for all $f(P_H)$ evaluations, including evaluations of the CDF, tail distribution, and tail quan-

tiles. The interpolant uses Chebyshev points of the second kind, leading to simple and stable formulae (Berrut and Trefethen, 2004).

Since we are interested in a wide dynamic range in the tail of $f(P_H)$ to be able to determine a wide range of tail quantiles, we interpolate the logarithm of the PDF,

$$f(P_H) \approx \exp(\mathfrak{B}_{\text{tot}}(P_H)), \tag{D32}$$

where $\mathfrak{B}$ is the interpolant. If $f(P_H)$ is interpolated directly, the tail may be obscured by oscillations due to the PDF's bulk, which require unwieldy amounts of samples to achieve the desired accuracy.

A further challenge to overcome when interpolating $f(P_H)$ appears in the tail of $\ln f(P_H)$, which diverges to $-\infty$ as $P_H \to P_H^{\mathrm{m}}$ because the PDF vanishes at $P_H = P_H^{\mathrm{m}}$. Even when limiting the interpolation interval to $[0, P_H(1-\epsilon)]$ (where $\epsilon$ is the machine precision), the result of this steep descent at the endpoint of the interpolant's support leads to large oscillations. To handle this difficulty, we split the $P_H$ support into two intervals at $0.9 P_H^{\mathrm{m}}$. For the first interval, we use a standard barycentric Lagrange interpolant $\mathfrak{B}_0(P_H)$. For the second interval, we resolve the endpoint difficulty via a coordinate transform

$$t = \ln(P_H^{\mathrm{m}} - P_H) \tag{D33}$$

which transforms the divergence for $P_H \to P_H^{\mathrm{m}}$ into a slope for as $t$ approaches its lower limit $t_{\min}$. With $P_H$ given in double precision, discernible up to precision $\epsilon$, the smallest $t$ that can be discerned from $P_H^{\mathrm{m}}$ is

$$t_{\min} = \ln\left(P_H^{\mathrm{m}}(1-\epsilon)\right). \tag{D34}$$

The support for the second interpolant $\mathfrak{B}_1(t)$ is therefore $[t_{\min}, t_{\max}]$ with $t_{\max} = \ln\left(0.1 P_H^{\mathrm{m}}\right)$. Finally, the combined interpolant is

$$\mathfrak{B}_{\text{tot}}(P_H) = \begin{cases} \mathfrak{B}_0(P_H) & : P_H \leq 0.9 P_H^{\mathrm{m}} \\ \mathfrak{B}_1\left(\ln\left(P_H^{\mathrm{m}} - P_H\right)\right) & : 0.9 P_H^{\mathrm{m}} < P_H < P_H^{\mathrm{m}} \\ -\infty & : \text{otherwise.} \end{cases} \tag{D35}$$

Both $\mathfrak{B}_0$ and $\mathfrak{B}_1$ use Chebyshev points of the second kind (Berrut and Trefethen, 2004)

$$x_s = \cos\left(\frac{(2s+1)\pi}{2S+2}\right) \quad \text{with} \quad s = 0, \ldots, S \tag{D36}$$

scaled to their respective support. For this point scheme, we implemented the following adaptive refinemenet strategy:

1. Start with $\ln f$ evaluated at $S+1$ Chebyshev points of the second kind.

2. Evaluate $\ln f$ at $2S+1$ Chebyshev points of the second kind. All previous points can be reused and $\ln F$ needs to be evaluated only at the Chebyshev points of even index $s$. These new points are located in the intervals spanned by the odd $S+1$ starting points.

3. For each of the newly evaluated points, compute the difference of $\ln F$ with the barycentric Lagrange interpolation using the $S+1$ starting points. Obtain the maximum absolute deviation between interpolant and $\ln F$ over the new points.

4. If the maximum absolute deviation is greater than a set tolerance, set $S \to S' = 2S+1$ and repeat from step 1. Otherwise exit the refinement.

This allows us to refine the interpolant's approximation of the PDF up to a desired relative precision.

## D4   Cumulative functions of the marginal $P_H$ posterior: adaptive Simpson's rule

For the cumulative and related functions of the marginal $P_H$ posterior—the CDF, the tail distribution, and the tail quantiles—

1250 we need to be able to evaluate the $z$-integral of $\Psi$, Eq. (D14), for parts of the interval $[0, 1]$. For this purpose, we divide the full interval into a binary tree of sub-intervals in which we can evaluate the integral to sufficient precision.

    In each subinterval, we evaluate $f$ (by default using its barycentric Lagrange interpolant) in three points: the center and the endpoints (hence adjacent subintervals share function evaluations). The total mass of a subinterval can then be evaluated using Simpson's rule (Mysovskikh, 2006)

$$1255 \quad \int_{z_l}^{z_r} \mathrm{d}z \, f(z) \approx \frac{z_r - z_l}{6} \left( f(z_l) + 4f(z_c) + f(z_r) \right) \tag{D37}$$

Furthermore, we can evaluate the left-aligned quadratic polynomial defined by these three function evaluations:

$$\mathfrak{f}(\delta z) = \mathcal{C}_0 + \delta z \left( \mathcal{C}_1 + \delta z \mathcal{C}_2 \right) \tag{D38}$$

where   $\delta z = z - z_l, \qquad \mathcal{C}_0 = f(z_l),$

$$\mathcal{C}_1 = \frac{1}{z_r - z_l} \left( -3f(z_l) + 4f(z_c) - f(z_r) \right)$$

$$1260 \qquad \mathcal{C}_2 = \frac{2}{(z_r - z_l)^2} \left( f(z_l) - 2f(z_c) + f(z_r) \right)$$

This polynomial can readily be integrated to any point $\delta z$ within the interval:

$$\mathfrak{F}(\delta z) = \delta z \left( \mathcal{C}_0 + \delta z \left( \frac{\mathcal{C}_1}{2} + \delta z \frac{\mathcal{C}_2}{3} \right) \right). \tag{D39}$$

These tools at hand, the adaptive Simpson's and polynomial quadrature rule implemented for the cumulative distributions is as follows: start with the root element over the full interval $[0, 1]$ in the TODO list. Iterate the following until the TODO list is

1265 empty:

    1. Choose and remove an item from the TODO list with integrated polynomial $\mathfrak{F}_0$ spanning an interval $[z_l, z_r]$.

    2. Evaluate $f$ at the two centers of the subintervals $[z_l, z_c]$ and $[z_c, z_r]$. Using the central nodes of the subintervals, use Simpson's rule to compute the integrals over the subintervals ($I_l$ and $I_r$).

    3. If $I_l$ and $I_r$ are within a prescribed tolerance of the estimates obtained by evaluating $\mathfrak{F}_0$, accept the chosen interval $[z_l, z_r]$.
Otherwise split the interval at $z_c$ and add the two subintervals to the TODO list.

Over the resulting tree of subsequent subintervals, we sum the integrals in forward and backward direction to obtain, at the start and end of each subinterval, an estimate of the CDF and tail distribution.

Evaluating the CDF and tail distribution at a point $z$ then amounts to a binary search to find the interval that contains the coordinate $z$, computing the corresponding $\delta z$, evaluating $\mathfrak{F}(\delta z)$, and adding to or subtracting from the corresponding value at the subinterval's boundary.

To compute tail quantiles $t$ of the marginal posterior in $P_H$, Eq. (24), we use the TOMS 748 method (Alefeld et al., 1995) to find the root $z_t$ of the expression $\mathfrak{T}(z) - t$, where $\mathfrak{T}(z)$ is the tail distribution evaluated by the subinterval tree. The solution $z_t$ is then scaled back to $P_H$ coordinates.

## Appendix E: Neighbor density on a disk with uniform point density

In this section, we derive the neighbor density of points drawn from a disk of radius $R$ with uniform point density. This neighbor density is the probability density of the distance $d$ between to points which are both drawn from a uniform point density on the disk. In other words, this distribution describes the following: if we draw two random points from the uniform density on the disk, what distance $d$ do we expect between the two points?

Suppose that we have a points $p_0$ drawn randomly from the uniform distribution on the disk. Without loss of generality, we can rotate the disk as indicated in Fig. E1 (a) so that $p_0$ is at distance $x$ from the center of the disk. For the random point $p_0$ drawn from the disk, $x$ follows the distribution

$$p(x) = \frac{2x}{R^2} \, . \tag{E1}$$

The orange circle wedge shows the set of points within the disk that are located at distance $d$ from $p_0$. For the configuration shown in panel (a) of Fig. E1, the wedge intersects the disk's border at the red dot. This dot can be parameterized by the angle $\alpha$, measured counterclockwise from the line that connects $p_0$ with the disk's center. The angle can be computed by the law of cosines:

$$\alpha(d, x, R) = \arccos\left(\frac{x^2 + d^2 - R^2}{2xd}\right) \, . \tag{E2}$$

The sketch Fig. E1 (a) equips us to compute the density of points at distance $d$ from $p_0$ within the set of all points in the disk. Conceptually, we grow the light gray circle from panel (a) from a point ($d = 0$) up to the maximum size that intersects with the disk ($d = R + x$, indicated in panel (b)). For each $d$, the density is the length of the orange wedge, that is, $2\alpha d$, divided by the disk's area. If $d$ is too large ($d > R + x$), the density is zero, and if $d \leq R - x$, the length is the full circle perimeter (see panel (c) of Fig. E1). The density of $d$ conditional on $x$ is hence

$$p(d \,|\, x) \sim \begin{cases} 0 & : d > R + x \\ 2\alpha(d, x, R)d & : R - x < d \leq R + x \\ 2\pi d & : d \leq R - x \end{cases} \, . \tag{E3}$$

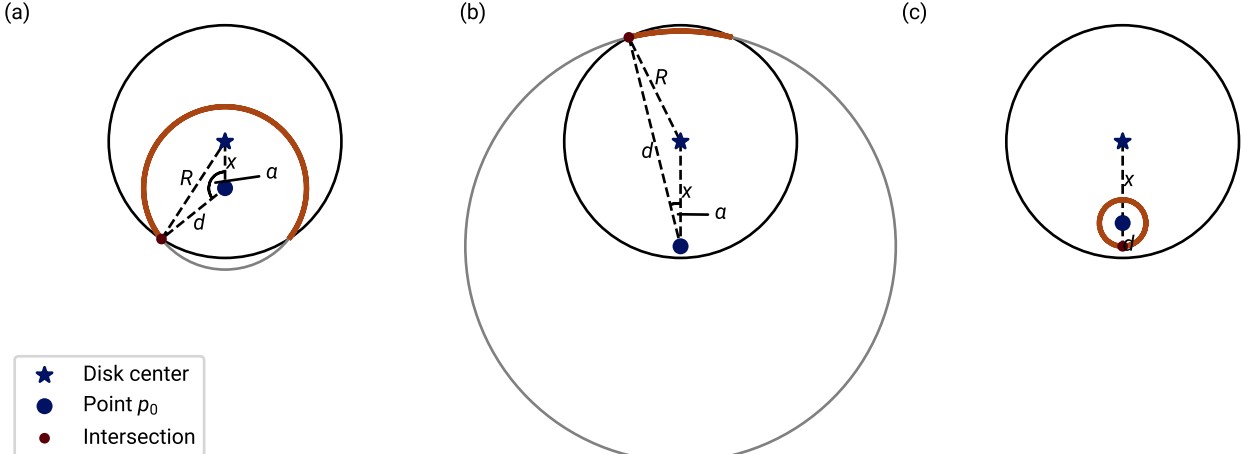

**Figure E1.** Geometry for the computation of the neighbor density by distance on a disk of radius $R$ with uniform point density. This sketch illustrates the sets of points at a distance $d$ from a test point on the disk. (a) For a point that is located at distance $x$ from the disk center, the set of points at distance $d$ within the disk is a circular segment that spans an angular range of $2\alpha$. This angle $\alpha$ is determined from the trigonometry of the parameters $x$, $d$, and $R$. (b) As the distance $d$ approaches its upper limit $2R$, the circular segment converges to a point that is antipodal to a point on the disks perimeter ($x = R$). (c) If $x + d \leq R$, the set of points at distance $d$ from the test point is a full circle of radius $d$.

To obtain the distance distribution within the population of pairs $(p_0, p_1)$, this density needs to be averaged over the density $p(x)$, Eq. (E1). We hence find the density

$$f(d) = \frac{1}{F} \int_0^R \mathrm{d}x\, p(d\,|\,x) p(x) d \tag{E4}$$

with $p(x)$ given in Eq. (E1), $p(d\,|\,x)$ given in Eq. (E3), and with the normalization constant

$$F = \int_0^{2R} \mathrm{d}y \int_0^R \mathrm{d}x\, p(d\,|\,x) p(x) y\,. \tag{E5}$$

The resulting density is shown by dashed lines in Fig. 3. As $d_{\min}$ increases in that Fig., the density Eq. (E4) is adjusted by setting the lower $y$ integration bound for $F$ in Eq. (E5) to $d_{\min}$.

## Appendix F: Artificial surface heat flow with gamma-like aggregate distribution

This section describes how the artificial heat flow field in Fig. 2 is generated. The generation follows a two step procedure.

First, a target surface heat flow distribution is generated on the $x$ interval shown in Fig. 2 (a). This target heat flow distribution is determined through thirteen equidistantly distributed points on the $x$ range (the *control points*). On the two boundary points, the heat flow is set to $42\,\mathrm{mW\,m^{-2}}$ (Lucazeau, 2019, an arbitrary value within the range of heat flow encountered within regional samples of the NGHF;). The eleven remaining points are allowed to vary freely between $35\,\mathrm{mW\,m^{-2}}$ and $50\,\mathrm{mW\,m^{-2}}$. Between these thirteen points, the heat flow field is interpolated using SciPy's smoothing cubic spline with smoothing parameter $s=2$ (Dierckx, 1975; Virtanen et al., 2020). The interpolated heat flow field is evaluated at 573 equidistant points on the $x$ interval, and a cost function is constructed by computing the Anderson-Darling statistic for a maximum likelihood estimate of the gamma distribution on these 573 sample values. This cost is minimized by optimizing the heat flow values at the eleven control points using the SciPy implementation of the limited-memory bound-constrained BFGS optimizer (Byrd et al., 1996; Virtanen et al., 2020).

Once this target heat flow distribution with gamma-like aggregate distribution has been created, it is used as an optimization target for the heat flow generated by underground heat sources. Below the $x$ extent shown in Fig. 2, $x \in [-50\,\mathrm{km}, 75\,\mathrm{km}]$, a 200 km wide and 80 km deep grid is created with $x \in [-80\,\mathrm{km}, 120\,\mathrm{km}]$ and 201×151 cells. The heat generation in each cell is allowed to vary between $0\,\mathrm{W\,m^{-3}}$ and $8\,\mathrm{W\,m^{-3}}$, below which most rock samples are found (Jaupart and Mareschal, 2005, Figure 1). The material is assumed homogeneous with a thermal conductivity of $2.5\,\mathrm{W\,m^{-1}\,K^{-1}}$ (Lachenbruch and Sass, 1980, as used in ). At the surface, a temperature of $0\,\mathrm{K}$ is assumed. At 80 km depth, a boundary temperature of $318\,\mathrm{K}$ is enforced, causing a surface heat flow of $10\,\mathrm{mW\,m^{-2}}$ that is superposed by any heat sources in the grid. The heat equation is solved on the grid using the finite difference method. Starting from a uniform heat generation of $0.8\,\mathrm{W\,m^{-3}}$ in each grid cell, the constrained trust-region optimization algorithm of SciPy (Conn et al., 2000; Virtanen et al., 2020) is used to minimize the squared deviations of the solved surface heat flow from the target heat flow evaluated at the corresponding surface points.

The code to generate Fig. 2 is part of the REHEATFUNQ model (Ziebarth, 2023, `A9-Simple-Heat-Conduction.ipynb`).

## Appendix G: A note on nonlinear heat transport

This is a technical note on the applicability of REHEATFUNQ for nonlinear heat transport, brought to our attention by the anonymous reviewer 1. A central equation of connecting the Bayesian methods with the physical model of heat conduction, ultimately leading to the $P_H$ posterior, is Eq. (18):

$$q_a(\boldsymbol{x}_i) = P_H c(\boldsymbol{x}_i). \tag{G1}$$

This equation assumes linearity of the heat conduction in heat power $P_H$—which is the case for heat conduction if $P_H$ is sufficiently small but may break down as other means of heat transport set in as $P_H$ increases. In such a case of non-linear heat transport, the anomaly may in general be of the non-decomposable form

$$q_a(\boldsymbol{x}_i) = \mathfrak{q}(P_H, \boldsymbol{x}_i) \tag{G2}$$

REHEATFUNQ is not able to handle such a non-decomposable non-linearities in heat transport. The linear decomposition of $q_a(\boldsymbol{x}_i)$ into a constant coefficient $c_i$ and a variable but global magnitude parameter (here $P_H$) is required. However, there is

one kind of non-linear dependence of the heat transport onto the power $P_H$ that fulfills this requirement: a factorization into a function of $P_H$ and a function of $\boldsymbol{x}$,

$$q_a(\boldsymbol{x}_i) = \mathfrak{p}(P_H)\mathfrak{c}(\boldsymbol{x}_i) \tag{G3}$$

Then, one could simply provide $\{\mathfrak{c}(\boldsymbol{x}_i)\}$ instead of $\{c_i\}$ to REHEATFUNQ. The posterior PDF $f$ and CDF $F$ would then be evaluated in $\mathfrak{p}$ instead of the heat power $P_H$. This result would then have to be manually transformed to $P_H$ by inverting the function $\mathfrak{p}(P_H)$.

We do not know whether there are cases of non-linear heat transport that factorize according to Eq. (G3) and show no spatial non-linearity, and hence whether this note is relevant. We leave this open for further study.

*Author contributions.* **Malte J. Ziebarth:** Conceptualization, Data curation, Formal analysis, Methodology, Software, Visualization, Writing – original draft preparation. **Sebastian von Specht:** Conceptualization, Formal analysis, Methodology, Software, Writing – Review & Editing.

*Competing interests.* The authors declare that they have no conflict of interest.

*Acknowledgements.* MJZ would like to kindly acknowledge helpful comments and support by John G. Anderson, Fabrice Cotton, and Oliver Heidbach on parts of this manuscript before it was split off a preceding research effort. MJZ would like to acknowledge funding from the Initiative and Networking Fund of the Helmholtz Association through the project Advanced Earth System Modelling Capacity (ESM). The authors would like to warmly thank the two anonymous reviewers for the valuable comments that helped us to improve our manuscript.

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
