# Peer review of "REHEATFUNQ 2.0.1: A model for regional aggregate heat flow distributions and anomaly quantification"

_EGUsphere, 2023_

## Author Response (AR1)

**Response to RC1**

Dear Reviewer #1,

thank you kindly for your extensive and valuable feedback. We have now addressed all points you mentioned, some of which required extensive work on the code base. Hence we would like to offer our apologies for the late reply.

We would like to respond to your comments in two steps. First, we would like to answer to a small subset of the comments in this interactive discussion; this is the first section of this document. The remaining majority of the comments led to straightforward improvements of our manuscript; we list them in the second part of this manuscript.

**Discussion**

- *The paper addresses many interesting aspects that influence the results of the heat flow determination and highlight the importance to consider the processes not in a deterministic manner but in a probabilistic to get an estimate of the variability of the heat flow. However, at times it is difficult to extract the key messages and consequences. It would be beneficial to move some of the more detailed technical explanations to the appendix and better highlight the key findings and consequences of the individual sections.*

    ⇒ Are there any specific parts which "more detailed technical explanations" refer to? We have improved the highlighting of the key findings in several places as the result of multiple others of your comments (and those of RC2).

- *Additionally, it might be useful to point out possible sources for the variability observed for the heat flow data points close to each other. So whether these are uncertainties caused by the measurements themselves or if they are physical processes yielding these differences.*

    ⇒ Four possible sources (variability of radiogenic heat production, displacement in faulting, topographic effects, and measurement errors) are mentioned in lines 149 to 159.

- *p. 7 l. 139-140: "If a region is characterized by uniform heat flow and sufficient data has been collected, a statistical analysis will yield a precise result". I would avoid the word "precise" this can be easily mistaken as a metric of how well the uncertainty quantification worked. A better formulation might be "yield a result with a small variability". Furthermore, it would be good to clarify this statement. The uniform heat flow is only visible when in addition no significant measurement errors are observable.*

    ⇒ We would like to kindly thank you for your suggestion but we have decided to remain with "precise". We find that in the context of observational error, "accuracy" and "precision" are quite well-known terms, and "precision" quantifies exactly what we want to express (and which you have also noted): small variance of the results with repeated measurements.

Nevertheless, we have clarified the meaning of "uniform". We use uniform in the sense of "identically distribution", that is, we imply that "uniform heat flow" is drawn from the same distribution throughout a region.

The issue of "significant measurement errors" of your last sentence is hence resolved. Within our underlying physical model (Equation 3), this distribution includes any potential measurement error. Hence, we do not need to measure constant heat flow throughout the region for the region to be "characterized by uniform heat flow" in the sense we have now clarified.

- *p. 8 l. 185-186: It is explained that clustered points have high discrepancies and will yield less accurate deterministic distributions. This statement should be further clarified. If neighboring points show very different flow values would that not also be an indication of potential measurement errors?*

  ⇒ The answer to the question is yes (strong differences between close heat flow measurements could be both due to strong material contrasts in terms of heat production, or due to measurement errors q_f), and REHEATFUNQ aims to capture this variability by means of the latent parameter described in sections "3.2.4 Handling spatial data clusters" and "3.4 Handling spatial data clusters". The paragraph l. 183-187 was written deliberately focused on the effect that spatial measurement clustering has on the regional aggregate heat flow distribution if at least a part of it is "random" due to the sampling of a spatially variable heat flow field (Fig. 4).

  ⇒ We have now added an early explanation that REHEATFUNQ aims to mitigate the clustering effect while still capturing the variability that is contained in a cluster.

- *p. 13 l. 292: Add a remark on how does this final cost relates to the desired cost?*

  ⇒ We neither assume nor define a desired cost. Our only criterion is that the cost should be minimal, hence implying that the "fit" of the prior to the distributions observed in the global data is maximal. At least locally, our final cost is minimal. To the degree that our sampling employed in the SHGO method is dense enough, the cost is also globally minimal.

- *p. 14 l. 299-303: Sensitivity is derived from a sensitivity analysis and has a specific mathematical meaning. Therefore, it would be important to not use the world in context of the uncertainty quantification and talk instead of uncertainties and variability.*

  ⇒ We would like to politely state our disagreement. First, uncertainty and sensitivity are two interwoven aspects of the same phenomenon: uncertain estimates. Second, in cited lines, we use sensitivity in the sense of "strength of the dependence onto input variation" which corresponds to its semantics in both sensitivity analysis and common speech. More specifically

- l.299-300, "and the parameters' sensitivities for different α and β." refers to the gamma distribution parameters' sensitivity to the (shape of the) distribution itself, and how this sensitivity changes as the distribution parameters change.

- l. 302-303 specifically explains the interwoven nature of uncertainty and sensitivity: "However, the sensitivity of the overall distribution relative to the distribution parameters—and consequently the uncertainty of the distribution estimates—changes with the distribution parameters."

- *p. 14 l. 304-306: So the uncertainty quantification has been performed with both influential and non-influential parameters? Has it been considered to perform a sensitivity analysis prior to the uncertainty quantification to determine the non-influential parameters and keep them fixed during the uncertainty quantification? This aspect is important because non-influential parameters can significantly degrade the robustness of the uncertainty quantification. Was the robustness of the analysis investigated?*

  ⇒ We have not performed uncertainty quantification. Lines 299-306 refer to general properties of the gamma distribution's parameters α and β if one were to compute point estimates and then quantify their uncertainty. Since REHEATFUNQ is a Bayesian approach, no point estimates of α and β are employed—instead, α and β are always integrated over all possible values—and uncertainty quantification therefore does not apply.

- *p. 23 l. 475: Why was a 5 % rejection rate chosen?*

  ⇒ The 5 % rejection rate was initially chosen ad-hoc due to the availability of critical statistics for the gamma and other distributions (e.g. 1 %, 2.5 %, 5 %, 10 % given by Stephens, 1986).
  If a lower rejection rate would be chosen, this rejection rate could not be resolved from the small number of disks that cover the global data set at small R (due to the minimum sample size criterion). For instance, one would need to have at least around 100 valid disks to be able to quantify a 1 % rejection rate, whereas around 20 disks suffice to quantify a 5 % rejection rate. In case of the NGHF data set we used, the graphs labeled "Inverse average number of disks in RGRDC" in Fig. 11 indicate by radius the number of valid disks that can be fit onto Earth surface (say, approximately R >= 120 km would be sufficient for 1 % rejection rate). Hence, reducing the rejection rate prevents the analysis of smaller disks—or, in other words, decreases the spatial resolution. The 5 % rejection rate we chose is close to the minimum that can be quantified at 80 km.
  If a higher rejection rate would be chosen, the goodness-of-fit tests are generally more powerful at the expense of a higher Type-I error. This is something we would like to prevent.
  All in all, the choice of the rejection rate has considerable leeway—yet 5 % seems to work well for the purpose of and the data set used in this study.

- *Section 4.3.1: Are general rules available for choosing the prior?*

⇒ There are the following general rules:
1) If the sample size is small, and the sample distribution is "typical" (i.e. similar to what we have found in the NGHF data set, see Fig. 14), the informed prior with optimized parameters is beneficial
2) if the sample's variance is low (e.g. after correcting for numerous known effects that cause spatial variation), the uninformed prior may be beneficial
3) If the sample size is large, both choices should be relatively similar.

⇒ Added a paragraph describing these general rules.

- *p. 33 l. 669: "To conclude, the test yield posterior support for our gamma model choice." Has this analysis also been done for the other distributions that showed in Figure 13 similar performance as the gamma distribution? That would be important to know in order to judge whether also other prior distributions yield similar posterior results.*

    ⇒ We have not performed this analysis for other distributions. Considering the amount of algebraic and numerical developments that went into stabilizing the REHEATFUNQ code to its current stage (that is, eliminating numerical difficulties in various parts of the parameter space), repeating this process to a comparable for other distributions would likely add another manuscript worth of workload onto the present manuscript. We hence consider this work out of scope for the present manuscript.
    That said, we have no reason to believe that, given a careful method development, a REHEATFUNQ model based on another similar-fitting PDF, such as the log-logistic or normal distribution, would perform significantly worse or better than our development based on the gamma distribution. The differences between a good fit of these three distributions among each other are decidedly less than the difference between the gamma distribution and the mixture distributions used in section 4.3.2. The phrase "To conclude, the tests yield posterior support for our gamma model choice" is meant to support (or validate) our specific model without indicating preference among the models tested in Fig. 13.

- *p. 34 l. 702: Can the geophysical modeling of the heat generation and transport that can help in detrending the data, be done in REHEATFUNQ or does this need to be done externally?*

    ⇒ No, this needs to be done externally (as mentioned also section "3.5 Heat Conduction").

- *p. 39 Figure 20: In panels d,f, and g the posterior pdf does not match with the data. Further explanations of what can be done in this case are required.*

    ⇒ These differences are due to the distance selection criterion. In the Carrizo, Creeping, and North Coast segment regions, there are visible effects of clustering (e.g. at geothermal areas or clustered around the San Andreas fault in the Carrizo segment area. Here, the minimum distance criterion works as intended: it enforces a more uniform sampling of the area, which does not overly weight small regions with clustered measurements, while keeping all that information within the latent dimension (exlusive sampling of the clusters). These effects are already described in detail in section 5.1 with

references to the maps Fig. 19. The remedy in this case, to bring the histograms closer to the posterior predictive, is to provide more uniform heat flow measurements.

⇒ We have added to the caption of Figure 20 a reference to the explanation in section 5.1, and we have mentioned that this section discusses the mismatch between data and posterior.

**Changes**

*General Remarks:*

- *For several equations not all variables are introduced (e.g., Eq. 5,7,11). Please check all equations and introduce all used variables.*

  ⇒ We have checked the equations and added introduction of variables (some of them redundantly to improve local readability)

- The figures are not displayed in the order they are mentioned and are partly placed far away from the corresponding text. So, it would be important to revise this.

  ⇒ We have referenced Figure 1 earlier and moved Figure 2. The cross reference to later Fig. 18 of the Southern California example is difficult to resolve; we have left it as is.

- *A main contribution of this paper is the characterization of the variability of the regional heat flow. For this Bayesian inference has been used for the heat flow anomaly strength. However, it is in the current form difficult to understand the individual steps performed for the entire analysis (so not only the anomaly) and to find out which parameters have been used since they are distributed over the different text sections and are partly missing. Therefore, it would help to have a designated section describing the workflow and presenting the workflow also in a schematic figure. Furthermore, it would be helpful to present the used parameters in form of a table.*

  ⇒ Added a section "Workflow Cheat Sheet" that lists the steps of a typical use of the REHEATFUNQ model directly after the introduction.

- *In the Supplement, a couple of additional Figures are presented. It would be good to reference these figures (so far only S1 is referenced) also in the main manuscript and provide additional explanations in the Supplement to better understand how these figures fit into the context of the manuscript.*

  ⇒ Added explicit Fig S2 to S4 references in section 4.3.2 where they were previously referenced implicitly.

*Further Remarks:*

- *p. 5 l. 104-105: What is the influence of d_min on the variability? This is mentioned later on but a brief explanation here and a reference to the detailed explanation would be helpful.*

  ⇒ Added a note that d_min aims to prevent biases due to spatial clustering of measurements, and a reference to the later following explanation

- *p. 7 l. 136: It is mentioned that REHEATFUNQ uses a black-box approach and prior to that also the importance of the physics is highlighted. So, it would be good to provide a small justification for using a black-box approach.*

    ⇒ We have detailed the connection between the paragraph of line 136 and the one before. We now emphasize that REHEATFUNQ aims to capture those principally known physical processes which yet cannot be modeled due to insufficient data. That is, REHEATFUNQ captures the known physics which is acting from unknown sources.

- *p. 8 l. 154: Please clarify in numbers what "sufficiently deep" means.*

    ⇒ Clarified as a range depending on the thermal gradient and the topographic variability

- *p. 8 l. 159-160: Please clarify shortly how the separation is performed.*

    ⇒ The separation of $q_a$ from the remaining heat flow signal is based on the superposition of $q_a$ and $q_u$ in equations (3) and (4), and on treating $q_u$ as a (gamma-distributed) random variable. "Separation" in this case means inferring the additive component $q_a$ based on the model for $q_u$ and on the knowledge about the pattern of $q_a$.
    Added a better explanation of this approach in l. 175 and rewrote the paragraph starting in l. 159.

- *p. 8 eq. 3: It would be good to provide a reference to the later section, where it is described how $q_a$ is determined.*

    ⇒ The anomalous or modeled heat flow $q_a$ is an input to the REHETAFUNQ model. Therefore, section 3.5, the later section which simplifies the computation of $q_a$ for one particular heat source, is not general enough to fulfill the role of "describ[ing] how $q_a$ is determined". We have rewritten the paragraphs preceding eq. (3) to better convey the message that a user of REHEATFUNQ's anomaly quantification would already have a $q_a$ in mind—which is determined by the problem that the researcher investigates. Nevertheless, we have referred to section 3.5 as an example of how $q_a$ is expressed in terms of $P_H$, and how the required coefficients $c_i$ can be computed (added more detail for that as well).

- *p. 8 eq. 4, line 175: Throughout the paper, the assumption is made that the magnitude of $q_a$ is small. It would be good to clarify if the software would work also for large magnitudes of $q_a$ and how this nonlinear function could be incorporated.*

    ⇒ We clarified that the software does not work if the heat source that generates the anomaly $q_a$ itself drives nonlinear convection. However, we note that this is not the main use case of REHEATFUNQ: If the anomaly heat source is that strong, the resulting anomaly will likely have a good signal-to-noise ratio against the background heat flow, and one will likely be able to discern the signal without REHEATFUNQ.
    (We also noted one technical way to apply REHEATFUNQ if the nonlinearity does not lead to a nonlinear signal in space, and can be handled by a global transformation of

variables. We cannot assess how likely such a case might be, though, so our preferred answer is that REHEATFUNQ does not work in these cases).

- *p. 8 l. 183: The number for the Figure is missing.*

    ⇒ Added

- *p. 9 Figure 4: Panel a: Please explain in detail how the heat flow was computed. According to the text a probability distribution has been used. So, is this one possible realization?*

    ⇒ The heat flow was computed by optimizing underground heat sources (not shown) such that the resulting surface heat flow field, sampled uniformly in space, has an aggregate heat flow distribution that is close to a gamma distribution (measured by the Anderson-Darling statistic). In a sense, this is one realization of a random process, but not as simple as one realization of a probability distribution. The wording in the caption has been clarified to concisely explain the generation of the heat flow field. A detailed explanation of the two-step process is given in a new section of the Appendix.

- *p. 11 eq. 9: It would be good to provide an explanation of why these equations are used.*

    ⇒ Gave a short explanation of Bayesian updating and its benefits by reducing the number of dimensions of numerical quadrature.

- *p. 13 l. 278: Which global optimization method has been used?*

    ⇒ This paragraph was clarified to mention that these are general considerations independent of the specific method. The paragraph was split at the point where algorithm, SHGO, is mentioned (l. 285) to clarify.

- *p. 13 l. 286: Please provide further details about the SHGO method.*

    ⇒ Added a short explanation of how the SHGO method works. Furthermore, gave more detail to the advantage of the SHGO method (yielding exactly one local optimization starting point per local minimum) if the parameter space has been sufficiently sampled.

- *p. 23 l. 480: What is the consequence of the regional aggregate heat flow not following a gamma distribution? This is later explained but it would be important to provide here already a brief explanation and a reference to the later section 4.2.3 for the detailed explanation.*

    ⇒ Added a brief explanation of the potential problem of using an imperfect model, and refer to sections 4.2.2, 4.2.3, 4.3.2

- *p. 26 l. 534: The process has been repeated 100 times. Has a convergence test been performed for this?*

    ⇒ Added Figures S9, S10, and S11 in the supplement, showing the convergence of the plots in Fig. 13 of the manuscript.

- *p. 30 l. 609: How many samples have been used for the Monte-Carlo simulations? Where convergence test performed?*

⇒ Added Figures S7 and S8 in the supplement that shows the convergence analysis for Fig. 15 in the manuscript.

- *p. 32 l. 640: See the previous comment about the convergence test.*

    ⇒ Same simulations as previous point.

- *p. 35 l. 724-725: What impacts would an uncertain heat transport have on the determination of the heat flow over REHEATFUNQ?*

    ⇒ Uncertain heat transport leads to further uncertainty and we expect a further diffusion of the prior if the heat flow anomaly is variable within a certain range. Since the heat transport enters the REHEATFUNQ analysis only through the coefficients $c_i$, uncertain heat transport can be introduced to our model by varying these coefficients according to the uncertainty of the heat flow model, and provide weights for each point in the model range. In this way, one would condense all uncertainty of the heat transport into marginal distributions of the $c_i$.

    ⇒ We have implemented an option to provide uncertain heat transport via a weighted list of sets of functions $f_i$ to evaluate the coefficients at the data locations (the structure is $\{(w_i, f_i(x,y)) : i=1,\ldots,n\}$). This is effectively a sample of the marginal "distribution" of the heat flow distribution. Internally, this set of weighted functions is combined, randomly sampled, with the set of data selections from the minimum distance criterion to yield a combined latent parameter dimension. This dimension samples the combined posterior of the distance criterion and the effect of uncertain heat transport.

- *Supplementary Material: Figures S2 to S4 are missing the axis labels.*

    ⇒ Added missing labels

**Response to RC2**

Dear Reviewer #2,

thank you kindly for your helpful and valuable feedback that led us to improve our manuscript.

We would like to respond to your comments in two steps. First, we would like to answer to your high-level comments in this interactive discussion; this is the first section of this document. The remaining text corrections led to straightforward improvements of our manuscript; we list them in the second part of this manuscript.

**Discussion**

*This contribution presents a model tackling the problem that surface heat flow measurements do not dissociate the background crustal heat flow from the heat flow induced by anomalies such as fault friction, taken as the feature of focus in this model. A method is devised to separate from heat flow measurements, the background heat flow and the strength of a fault frictional heat anomaly. This Bayesian inference method seems adequate and is interesting to pursue. I like the random global R-disk coverings to reduce local variability but it could be specified in simpler words why it is used. The heat flow is assumed to follow a gamma distribution and the message is hard to distillate because it is emphasised the gamma distribution does not fit the data, but that it is the best that can be used. The model was tested on 4 different regions around the San Andreas fault to revisit previous results about fault strength.*

- *The paper is dense and very technical. With my limited understanding of the technical content, I had a hard time following the authors and I believe the paper is not easily accessible to every geoscientist reader. In that sense, interpretations should be supplemented by a more physical interpretation so that all geoscientists are able to understand and use the model and/or cite the results. Those interpretations could be used as summary of sections that are quite dense to help with readability.*

  - Thank you kindly for this feedback. Indeed, our paper tries to manage a difficult balance. On one hand, it is a reference of all technical details of the development of the REHEATFUNQ model, as well as a synthesis of all supporting tests and analyses that we have performed. On the other hand, it aims to make REHEATFUNQ accessible for a wide number of geoscientists—where as you have pointed out correctly our draft has a high potential to obscure the practically relevant details through the large amount of technical description. We have made a number of changes to the organization of the manuscript to make it more accessible to a wide geoscientific audience:

    - We have added a new one-page section "2. Workflow cheat sheet" directly after the introduction. This page gives a short summary of the steps involved in a typical usage scenario of REHEATFUNQ, lists Python classes involved in those steps, and links to the documentation where code examples are listed.

- - Added the "Points-of-Interest Measurement" toy model to the appendix to better illustrate the potential impact of spatial data clustering, and to illustrate the mitigation using the $d_{min}$-sampling. The model is built on simple assumptions on the nature of human heat flow data acquisition.

- *I was confused by the introduction of the random global R-disk coverings this early as part the Data section 2.2 when more of its details is explained in the model section 3.2.4 and is also a mini section 3.4. I would like to see only one section about it for readability and I would rather see it in the model section 3.2.4.*

  - The original reason why the RGRDCs are mentioned at multiple places in the manuscript is that different variants of the same idea are used at different places (when comparing different distributions and when fitting the conjugate prior, we use coverings of the real world NGHF data. For the synthetic validation tests, we use random computer-generated data that mimic the RGRDCs of the NGHF). We have now consolidated the different sections into the appendix since the algorithms are mostly a technical detail—the underlying idea can be conveyed using just Figure 1.

- *I would like the authors to find a more positive and logical spin in the validation section about the choice of gamma distribution. Furthermore it should be clearly highlighted in 4.2.3 why the gamma distribution is the best distribution and if it is not, why it was chosen regardless. I do not understand why Weibull is not a good distribution for example. It should be also highlighted if no other distribution is better, what could be an alternate solution that would produce a lower rejection rate and why is the solution not considered in the model (maybe it cannot be modelled?).*

  - As we have noted in the section, the results of the comparison is not conclusive (arguably with the exception of rejecting the Fréchet distribution). No distribution is unanimously selected in the majority of regional aggregate heat flow distributions, and furthermore the positive evidence (Fig. 13a) is not significant in the vast majority of cases that a distribution is selected by the BIC. In particular, this concerns also the Weibull distribution: as we have noted, it has the highest BIC selection rate but the vast majority of selections have a $\Delta BIC<2$, which is not significant (e.g. Kass & Raftery, 1995, p. 777) and may well be due to random fluctuations. Concluding, with the data we have at hand there is no significant preference between many of the distributions investigated, including the gamma distribution. Therefore, the choice of distribution is a modeling decision.

    - A further note on the Weibull distribution: the possible left-skewness of the Weibull distribution may help the Weibull distribution to acquire a significant amount of BIC selections on small sample sizes by pure chance. We have tested the BIC selection rates for some random data of sample size N=10 drawn from a gamma distribution and found that the Weibull distribution was selected by the BIC criterion over the gamma distribution in about one third of the cases.

  - From a modeling point of view, the gamma distribution is the only one that combines all of the following criteria: (1) it is defined on a positive support, (2) it has a conjugate prior

(important for enabling the costly computations), (3) it is right-skewed, like the global heat flow distribution, for all parameter combinations

  ○ We have highlighted a possible alternate solution, leading to a lower selection rate, in section 4.2.2 (now 5.1.2) and Fig. 12 (now Fig. 11). The high rejection rate can be explained by the bimodal mixture distribution of two gamma distributions. Vice versa, one may be able to achieve a lower rejection rate by (1) deriving a REHEATFUNQ model for a gamma mixture distribution or (2) by considering spatial dependence of distribution parameters. These two avenues are possible improvements of our model in future works. For now, we have shown in section 4.3.2 (now 5.2.2) that the REHEATFUNQ method can handle bimodal regional aggregate heat flow distributions, and it does so by yielding results of higher uncertainty.

  ○ We have improved the conclusion of section 4.2.3 based on these arguments.

**Changes**

*Text correction:*

- *Section 2 Data: can the term "data" be specified?*

  ○ We have specified that this section is about heat flow data, and we note furthermore that it describes the database we have used (Lucazeau, 2019) and the kind of filtering we applied to this database.

- *line 117: leads us to chose a minimum distance -> choose*

  ○ Corrected.

- *Line 183: illustrated in Figure ? Highlights*

  ○ Added the Figure number.

- *Figure 4 refers to CDF which has not been introduced yet*

  ○ Added the abbreviation to the caption.

- *Line 241: a prior -> a priori*

  ○ Corrected.

- *Line 327: we can compute evaluate posteriors -> ?*

  ○ Corrected to "evaluate".

- *Can section 3.5 about heat conduction be part of 3.3 on the anomaly detection? I feel it should not be an independent section*

  ○ Done.

- *Rejection rate needs to be explained more in details*

  - Added more explanation in lines 475+ (of the old manuscript)

- *Section 4.3 synthetic data: can the term "data" be specified?*

  - Added a short description to the first sentence of that section: "compute-generated samples $\{(x,y,q)_i\}$ of surface heat flow."

- *Line 793: the results are less clear -> I have trouble seeing what makes a result clear or not…*

  - Specified the sentence: the "results" refer to the existence or non-existence of a finite heat flow anomaly.

---

## Author Response (AR2)

**Response to Report #1 by Anonymous Referee #2**

Dear Anonymous Referee #2,

thank you kindly for your second review of our resubmitted manuscript. The comments have helped us polish our manuscript.

We now follow with a point-by-point response to your report. All line numbers in this response refer to the marked-changes file "`Ziebarth_von_Specht_REHEATFUNQ_diff.pdf`" of the last review round.

**Discussion**

*It is good that the authors put more of the technical details in appendix, it clears out the main content of the text.*

- *I took the opportunity of a refreshed look at the manuscript, to take again the viewpoint of a random reader. And while the idea of the section "Workflow Cheat Sheet" is valuable, it is not where it should be. I was confused by all variables and abbreviations that had not been introduced at this point. Hence, I suggest putting the section and its figure in appendix and referring in relevant part of the manuscript to which phase of the workflow in the appendix is being detailed (because I noticed that this section had not been referred anywhere in the manuscript).*

  - We have moved the section to the appendix. To fulfill the role that the early cheat sheet was intended to fulfill (i.e. to give geoscientist readers from different backgrounds a quick overview about which parts of the manuscript may be relevant for them), we have appended an outline paragraph to the introduction.

    - We have referred to steps of the workflow in the manuscript where appropriate.

    - We have slightly reordered independent points of the listed workflow

    - We have moved the figure to the new section "3.4.1 Providing heat transport solutions", which is the location that it is first referenced in the revised manuscript.

- *RGRDC needs to be introduced in the text since the abbreviation is now explained in the appendix otherwise, I suggest putting it in legend of Figure 2 (heat flow data).*

    - Introduced in the legend of Figure 2.

- *Section 3.2 is two lines. It needs to be absorbed in another section.*

    - These two lines plus the (definition) equation were absorbed to section 3.

- *Appendix C needs to be placed back in the text.*

    - We worked Appendix C into section "4.3.1 A combined gamma model".

- *Line 449: "optimziation"*

- ○ Corrected.

- *I believe paragraph of line 638 should not exist anymore, because the section was removed.*

  - ○ Thank you for drawing our attention to this paragraph, and, in fact, the whole introductory paragraphs l. 634–649 to section 5. For some reason, all section references therein to the preceding section 4 were hand-written (other than all remaining section references in the manuscript).

    - ▪ We have removed the paragraph of line 638. The appendix describing the RGRDCs is cited where needed.

    - ▪ We have replaced the fixed section numbers with latex label references.

- *Line 714 should not mention results obtained in future sections.*

  - ○ The paragraph l. 710–716 answers to the following comment

    > *"p. 23 l. 480: What is the consequence of the regional aggregate heat flow not following a gamma distribution? This is later explained but it would be important to provide here already a brief explanation and a reference [...]."*

    of RC1 of the previous review round. Anticipating the result of the analysis performed in the later section 5.2.2 is required to answer the question "What is the consequence […]?" (lines 710-714 discuss only what *could be* the consequences).
    We agree, though, that an anticipation of the results of section 5.2.2 is not required for the manuscript's arc and have hence

    - ▪ removed the last sentence of that paragraph.

- *Paragraph of line 720 needs to come at line 710.*

  - ○ It is true that the paragraph of line 720 is not an optimal end of section 5.1.1. This paragraph discusses ancillary results of Figure 10 – the section 5.1.1 can be improved by ending with a discussion of its most important finding: the rejection rate is elevated in a way that cannot be described by data uncertainty alone. However, moving the paragraph of line 720 to line 710 is not optimal. All text beginning from "A striking observation" (l. 700) up until line 719 is a continuous discussion of the issue of the elevated rejection rate. Inserting paragraph l. 720 at line 710 breaks this arc and distracts with the ancillary results. In our opinion, the following edit better addresses your comment:

    - ▪ insert paragraph of line 720 into the paragraph of line 700 (start the results listing with the ancillary results)

    - ▪ break the paragraph of line 700 at "A striking observation" (continue and end the section with the discussion of the main result)

- *References to sections and figures should be consistent (Sec. and Fig. or Section and Figure?) and need to be checked, as well as the ones to appendix. I would advise introducing LaTeX*

*labels to point to sections, some seem to be done manually at the moment which could explain why mistakes were left out.*

- We kept "section" homogeneous.

- We have homogenized the labeling of "Figure" and "Equation" to be abbreviated in-sentence and written out at sentence beginnings. Figure and Equation are capitalized.

- We have performed a dedicated reading of the manuscript focusing only on inter-manuscript references, and updated some references that have undetectedly pointed to wrong markers after previous revisions.